# Task-Specific Exploration in Meta-Reinforcement Learning via Task Reconstruction

**Radu Stoican**[*]                                   *radu.stoican@manchester.ac.uk*
*Manchester Centre for Robotics and AI*
*The University of Manchester*
*Manchester, United Kingdom*

**Angelo Cangelosi**                            *angelo.cangelosi@manchester.ac.uk*
*Manchester Centre for Robotics and AI*
*The University of Manchester*
*Manchester, United Kingdom*

**Christian Goerick**                          *christian_goerick@honda-ri.com*
*Honda Research Institute USA, Inc.*
*San Jose, California, USA*

**Thomas H. Weisswange**                     *thomas.weisswange@honda-ri.de*
*Honda Research Institute Europe GmbH*
*Offenbach am Main, Germany*

**Reviewed on OpenReview:** *https://openreview.net/forum?id=VRRapVcaJH*

## Abstract

Reinforcement learning trains policies specialized for a single task. Meta-reinforcement learning (meta-RL) improves upon this by leveraging prior experience to train policies for few-shot adaptation to new tasks. However, existing meta-RL approaches often struggle to explore and learn tasks effectively. We introduce a novel meta-RL algorithm that learns to learn task-specific exploration policies for sample-efficient few-shot adaptation. We achieve this through task reconstruction, an original method for learning to identify and collect small but informative datasets from tasks. To leverage these datasets, we also propose learning a meta-reward that encourages policies to learn to adapt. Empirical evaluations demonstrate that our algorithm achieves higher returns than existing meta-RL methods. Additionally, we show that even with full task information, adaptation is more challenging than previously assumed. However, policies trained with our meta-reward adapt to new tasks successfully.

## 1 Introduction

Research in reinforcement learning (RL) has led to many impressive achievements and powerful methods in the past decade (Sutton, 2018). However, real-world usage is still not widespread, as the algorithms used to solve RL problems often suffer from low sample efficiency and poor generalization to new tasks (Hospedales et al., 2021; Beck et al., 2023b). Meta-reinforcement learning (meta-RL) is a re-emerging approach that tackles these issues. It follows the meta-learning approach of "learning to learn" (Schmidhuber, 1987; Thrun & Pratt, 1998), making it well-suited for few-shot adaptation settings. In few-shot adaptation, agents aim to become optimal in any new task from a given distribution of tasks, after collecting only a few episodes.

---

[*]Corresponding author.

Instead of directly learning to solve tasks from scratch, meta-RL finds RL algorithms that quickly learn the optimal policy. While meta-RL agents are general, the algorithms they produce are task-specific. These meta-learned algorithms contain prior knowledge of the task distribution. By leveraging this prior, they quickly learn the optimal method of exploring or solving a task. Therefore, for any task in the distribution, meta-learned algorithms are expected to be more sample-efficient than and outperform standard RL algorithms.

Meta-RL has already been successfully applied to ad hoc teamwork (Zintgraf et al., 2021a; He et al., 2023; Mirsky et al., 2022), robotics (Nagabandi et al., 2018; Zhao et al., 2022), human-robot interaction (Gao et al., 2019; Ballou et al., 2023), multi-agent RL (Yang et al., 2022; Xu et al., 2022a), and sim-to-real transfer (Arndt et al., 2020). Despite this, meta-RL has not yet fully addressed the challenges it aims to overcome (Stoican et al., 2023). While recent years have seen several improvements, meta-RL still struggles with adapting to complex tasks. A significant issue is the difficulty of designing strategies for few-shot exploration in complex, dynamic RL environments. The advantage of meta-RL is that these strategies do not have to be manually crafted, but can be learned from data for each particular task distribution. Often, meta-RL agents interact with a task for only a few episodes before being evaluated on that same task. These agents can be seen as exploring the task by using an implicit or explicit meta-learned exploration strategy. However, implicit exploration may be too sample-inefficient for few-shot adaptation (e.g., leading to agents that do not use their task distribution priors effectively during exploration). Recent approaches have shown that explicitly optimizing for few-shot exploration leads to more powerful adaptive agents (Beck et al., 2023b). Moreover, the difficulties of few-shot adaptation extend beyond task exploration. Our empirical results complement those of Beukman et al. (2024) in showing that optimizing a policy that leverages learned priors to solve new tasks can be non-trivial.

To get a clear high-level picture of in-context meta-RL, we consider separating it into three main parts, as shown in Fig. 1. The first phase is task exploration. After a task is sampled from the task distribution, a policy $\pi^{\text{explore}}$ collects a "few" episodes of data from it. Optionally, this policy may be guided by an encoder $f_u$. The goal of $\pi^{\text{explore}}$ is to collect data that contains useful information about the task. This information is then extracted by a second encoder $g$ during the task learning stage, and encoded into a task context vector. Finally, in the task-solving phase, a policy $\pi$ attempts to optimally solve the task at hand. This final stage is similar to standard RL, except that $\pi$ is conditioned on the task context.

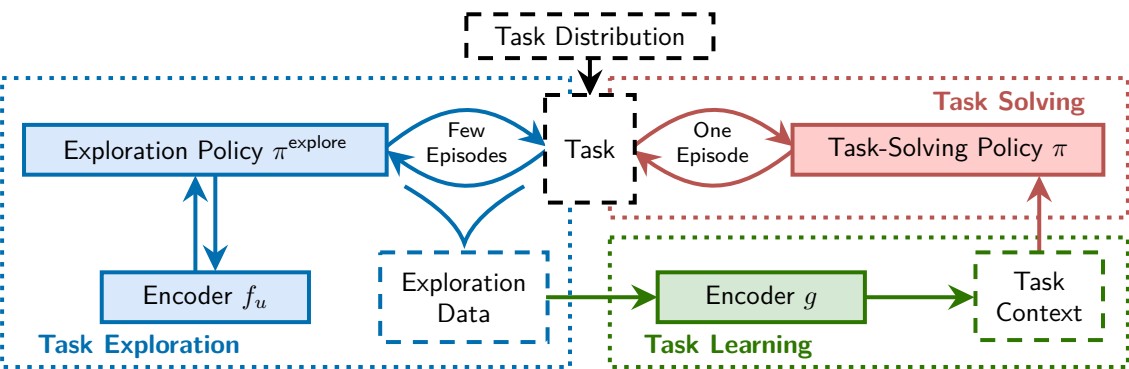

Figure 1: A high-level overview of in-context meta-RL.

There are several ways of learning to reinforcement learn. The approach previously described, and used in this work, is called in-context meta-RL. In-context policies $\pi^{\text{explore}}$ and $\pi$ use task context information provided by $f_u$ and $g$ to explore and solve tasks, respectively. Note that this three-phase paradigm is not always cleanly separated in practice, e.g., encoders $f_u$ and $g$ may share information or not be separate encoders at all, a single policy may be used for both task exploration and solving, etc. Nevertheless, this view helps form an intuitive understanding of meta-RL. Besides aiding intuition, we also use Fig. 1 throughout this paper to clarify which of the three components (i.e., task exploration, learning, or solving) we are discussing.

We introduce **La**tent **S**pace **E**xploration via Task **R**econstruction (**LaSER**), a novel meta-RL algorithm for few-shot adaptation.[1] The primary objective of LaSER is to learn task-specific exploration strategies. The core idea behind our method is to identify small datasets that capture rich task-specific information with minimal redundancy. We achieve this by introducing a novel approach of meta-learning by linearly reconstructing these small datasets into much larger datasets collected from the same task. We refer to this process as *task reconstruction*. It serves as a measure of how effective a given small dataset is for few-shot adaptation. Following this measure, LaSER meta-learns a latent space for directly identifying such small datasets, without having to first collect a larger one. This allows training a sample-efficient few-shot exploration policy that can collect task-specific data online. We then use the context vectors encoded from this data to train a policy that solves tasks. However, we show that previous approaches to training such a policy fail, even when using optimal task contexts provided by an oracle. Moreover, this failure persists even when the task distribution is replaced by a predefined set of tasks. As a result, LaSER's secondary objective is to learn to use task contexts effectively. To achieve this, we propose an augmented RL objective that encourages the task policy to continue exploring until contexts are effectively exploited. Overall, LaSER's optimization regime and architecture leverage data- and compute-heavy training to achieve rapid test-time adaptation requiring a low number of environment interactions.

We summarize our main contributions as follows.

1. We formalize an important assumption about the data required for few-shot adaptation. This forms the basis of our task reconstruction method. We then show how an encoder built from these ideas can learn representations that facilitate exploration.

2. We leverage these representations to design an intrinsic reward for training a task-specific, efficient few-shot exploration policy.

3. We propose using our aforementioned augmented objective to improve the adaptability of task-solving policies through directed exploration of the task context space. This simple change allows meta-RL policies to be trained with standard RL algorithms.

4. We introduce a hybrid encoder architecture composed of a unidirectional and a bidirectional model. The former is used for online task exploration, while the latter offers richer task contexts.

We formalize the meta-RL problem and introduce the necessary background in Sec. 2. In Sec. 3, we address the objectives described above and introduce LaSER, our novel meta-RL algorithm. We position our work within the meta-RL literature in Sec. 4. In Sec. 5, we provide empirical results by evaluating LaSER against existing meta-RL algorithms. We first assess LaSER's performance and adaptability to new tasks after a short exploration phase (i.e., four episodes). We then analyze each individual component, i.e., exploration policy, encoder, and task policy. Sec. 6 then provides a high-level discussion of the effect and applicability of our work within the broader meta-RL context. Finally, we conclude and provide directions for future research in Sec. 7.

## 2 Preliminaries

**Meta-Reinforcement Learning.** Meta-RL extends the standard RL problem. Instead of learning from a single task, a meta-RL agent trains on a distribution of tasks. Usually, the agent's objective is to adapt to new tasks. We consider a probability distribution $p(\mathcal{M})$ over tasks and define each task $\mathcal{M}_i \sim p(\mathcal{M})$ as a Markov decision process (MDP) $(\mathcal{S}, \mathcal{A}, T_i, R_i, \gamma, H)$. All MDPs share the set of states $\mathcal{S}$, set of actions $\mathcal{A}$, discount factor $\gamma$, and horizon $H$. However, the transition function, given as the probability $T_i(s' \mid s, a, \mathcal{M}_i)$ of transitioning from state $s$ to state $s'$ by taking action $a$ in task $\mathcal{M}_i$, is task-dependent. Similarly, the reward function $R_i$ is task-dependent. An episode $\tau = \{s_t, a_t, r_t\}_{t=1}^{H}$ is a sequence of states, actions, and rewards, such that at each time-step $t$ a tuple $(s_t, a_t, r_t)$ is collected. For a distribution over episodes $P_{\mathcal{M}_i}^{\pi}(\tau)$, we denote $\tau \sim P_{\mathcal{M}_i}^{\pi}$ to be an episode collected by following a policy $\pi(a \mid s)$ in task $\mathcal{M}_i$. The return of an episode $\tau$ is the accumulated discounted reward $G(\tau) = \sum_{t=1}^{H} \gamma^t r_t$.

---

[1]Our code is publicly available at https://github.com/RStoican/LaSER.

**In-Context Meta-RL** A common approach to solving meta-RL problems is in-context meta-RL, where a pretrained policy acts as an adaptable RL algorithm (Duan et al., 2016; Laskin et al., 2023; Moeini et al., 2025). Formally, given a context vector $c_i$ that describes a task $\mathcal{M}_i$, the goal is to train a policy $\pi$ to optimally solve $\mathcal{M}_i$ by taking in-context actions $a \sim \pi(\cdot \mid s, c_i)$, conditioned on both the state $s$ and the context $c_i$. The meta-RL agent can therefore learn new policies from $c_i$, even post-training. A standard approach is to concatenate $s$ and $c_i$, then optimize the in-context policy $\pi(a \mid s, c_i)$ using standard RL algorithms. However, this simple method may not generalize well across tasks in complex task distributions (Beukman et al., 2024). To address this, we propose a novel meta-RL optimization algorithm that explicitly uses $c_i$ to train in-context policies.

**Meta-RL for Few-Shot Adaptation.** In few-shot adaptation settings, in-context meta-RL attempts to optimize $\pi$ to solve previously unseen tasks $\mathcal{M}_i \sim p(\mathcal{M})$ in a sample-efficient manner. However, because the meta-RL agent lacks knowledge of the task-specific dynamics $T_i$ and $R_i$, it must first collect data from $\mathcal{M}_i$. The few-shot constraint limits this to $K$ episodes per task. We call this a $K$-shot adaptation problem. Following Beck et al. (2023b), we refer to a sequence of $K$ episodes collected in $\mathcal{M}_i$ as a meta-episode $\mathcal{D}_i^{(K)} = \{\tau_k\}_{k=1}^{K}$. We can then compute a latent context vector $c_i = g(\mathcal{D}_i^{(K)})$ using an encoder $g$. Importantly, $c_i$ should capture information about $T_i$ and $R_i$, to allow generalization to new tasks. To collect such meta-episodes, we follow Liu et al. (2021) and define a separate exploration policy $\pi^{\text{explore}}$, decoupled from the task-solving policy $\pi$. For a given task, its goal is to collect an informative meta-episode, without necessarily achieving a high return. We define $\pi^{\text{explore}}(a \mid s, \Gamma_i)$ to be an in-context policy, where $\Gamma_i$ is a task representation encoded specifically for exploration. Although we can choose $\Gamma_i$ to be the same as $c_i$, it is less restrictive to use a new representation. Given this setup, we generalize Liu et al. (2021) and consider agents that maximize the meta-RL objective

$$\mathcal{J}(\pi, \pi^{\text{explore}}, g) = \mathbb{E}_{\mathcal{M}_i \sim p(\mathcal{M}), \mathcal{D}_i^{(K)} \sim P_{\mathcal{M}_i}^{\pi^{\text{explore}}}} \left[ V_i^{\pi}\Big(g(\mathcal{D}_i^{(K)})\Big) \right], \tag{1}$$
$$\text{where } V_i^{\pi}(c_i) = \mathbb{E}_{\tau \sim P_{\mathcal{M}_i}^{\pi(c_i)}} [G(\tau)]$$

is the expected return in task $\mathcal{M}_i$ for the in-context policy $\pi$ conditioned on $c_i$.

**Training Setup.** Meta-training is the process of training a meta-RL agent. The agent seeks a $K$-shot algorithm to optimize policies for each task drawn from $p(\mathcal{M})$. The agent's ability to generalize is evaluated during meta-testing by solving new tasks from $p(\mathcal{M})$. For each meta-testing task, the agent is first allowed to collect and learn from $K$ episodes. Then, it is evaluated according to Eq. 1. We refer to collecting $K$ episodes from $P_{\mathcal{M}_i}^{\pi^{\text{explore}}}$ as task exploration. This type of exploration is meta-learned to be task-specific. Besides task exploration, the agent still performs the standard RL exploration. To achieve their objectives, $\pi$ and $\pi^{\text{explore}}$ have to explore their environment during meta-training. This is commonly referred to as meta-exploration (Beck et al., 2023b). During meta-testing, there is no meta-exploration.

**Data Representation.** States, actions, and rewards are represented as vectors or scalars. A timestep $(s_t, a_t, r_t)$ is then a $d$-dimensional vector, where $d$ is the sum of the dimensions of $s_t$, $a_t$, and $r_t$. We extend this to episodes. An episode $\tau \in \mathbb{R}^{H'}$ is a vector of $H$ concatenated timesteps, where $H' = Hd$. We can further extend this to meta-episodes. For a meta-episode $\mathcal{D}^{(K)} \in \mathbb{R}^{H' \times K}$ composed of $K$ episodes, the $k$-th column $\mathcal{D}_{:,k}^{(K)}$ corresponds to the $k$-th episode. Episodes and meta-episodes can therefore be represented as tensors instead of sequences, when convenient.

## 3 Methods

In this section, we present our approach to in-context meta-RL and use the ideas we introduce to design the main components of our proposed algorithm, LaSER. We structure this section based on the three-stage meta-RL paradigm in Fig. 1. Sec. 3.1 corresponds to the task-solving phase, while Secs. 3.2 and 3.3 focus on the task exploration part. Sec. 3.4 continues task exploration but also discusses task learning, with Sec. 3.5 corresponding to the combination of these three stages into a complete meta-RL algorithm.

Specifically, in Sec. 3.1, we introduce a novel approach for in-context RL optimization. Given a context vector $c_i$ for task $\mathcal{M}_i$, we propose augmenting the standard RL objective with a term that encourages

meta-exploration whenever $\pi$ can achieve better returns by improving its understanding of $\boldsymbol{c}_i$. Sec. 3.2 then discusses the training of an exploration policy $\pi^{\text{explore}}$ for collecting the data required to compute $\boldsymbol{c}_i$. It assumes a given encoder $f_u$ for guiding $\pi^{\text{explore}}$, with $f_u$ modeled as a unidirectional transformer (Vaswani et al., 2017) of size $d_{\text{model}}$. We propose a novel method of training $f_u$ for few-shot task-specific exploration in Sec. 3.3. We first introduce an important assumption on the data collected for few-shot adaptation, then show how a practical learning objective, i.e., task reconstruction, can be built from it.

Next, we describe our proposed architecture for encoders $f_u$ and $g$ in Sec. 3.4. We model $g$ as a transformer, as their ability to process each timestep in the context of an entire meta-episode has been shown to be advantageous in meta-RL settings (Melo, 2022; Shala et al., 2024). Finally, in Sec. 3.5, we introduce our meta-training and meta-testing algorithms. From this point forward, we simplify notation and drop the task subscript $i$ whenever it is clear that we are referring to task-specific data or representations.

## 3.1 Task Solving

As discussed in Sec. 2, optimizing an in-context policy $\pi(a_t \mid s_t, \boldsymbol{c})$ to adapt to new tasks is non-trivial (Beukman et al., 2024). We hypothesize that this difficulty arises because $\pi$ fails to leverage the task-specific information in context $\boldsymbol{c}$ when solving $\mathcal{M}_i$. While standard RL requires exploring the state space to understand a single task, meta-RL demands an additional meta-exploration objective: the agent must explore to understand the relationship between $s_t$ and $\boldsymbol{c}$. To encourage this directed meta-exploration of the state-context space, we augment rewards with a novel term that encapsulates meta-RL performance. This "meta-reward" allows the in-context policy to be optimized using standard RL algorithms.

Following this intuition, we propose a method that encourages $\pi$ to meta-explore more if $\boldsymbol{c}$ is not properly used. Let $V(s_t, \boldsymbol{c})$ and $V(s_t)$ be the approximated value functions at state $s_t$, with and without considering $\boldsymbol{c}$, respectively. We assume that, for an optimal policy, $V(s_t) \leq V(s_t, \boldsymbol{c})$ should hold at any timestep $t$, i.e., information from $\boldsymbol{c}$ should never decrease performance. Whenever $\pi$ violates this inequality during meta-training for some state $s_t$, the agent should meta-explore more from $s_t$. Our solution adopts the standard approach of encouraging exploration by maximizing entropy (Williams & Peng, 1991), with the important distinction that this maximization is guided by the context $\boldsymbol{c}$. We therefore propose the meta-reward

$$r_t^+ = r_t + \beta\, w(s_t, \boldsymbol{c})\, S[\pi](s_t, \boldsymbol{c}), \tag{2}$$

$$\text{where } w(s_t, \boldsymbol{c}) = \max\left(0, \tanh\left(V(s_t) - V(s_t, \boldsymbol{c}) - \zeta\right)\right) \tag{3}$$

is a non-negative dynamic weight on the entropy $S$ of $\pi(\cdot \mid s_t, \boldsymbol{c})$, $r_t$ is the environment reward, and $\beta$ is a constant. Note that $w(s_t, \boldsymbol{c})$ is non-zero only when $V(s_t) - V(s_t, \boldsymbol{c}) \geq \zeta$, for a given threshold $\zeta \in (-\infty, 0]$.

Intuitively, $w(s_t, \boldsymbol{c}) > 0$ means that the agent is disregarding information provided by $\boldsymbol{c}$ and should meta-explore in state $s_t$. When $w(s_t, \boldsymbol{c}) = 0$, i.e., $V(s_t) \leq V(s_t, \boldsymbol{c}) + \zeta$, we recover the original RL objective, with $r_t^+ = r_t$. By simply replacing $r_t$ with $r_t^+$, $\pi$ can be optimized in-context using standard RL algorithms. In practice, we use the proximal policy optimization (Schulman et al., 2017, PPO) algorithm, stabilized via proximal feature optimization (Moalla et al., 2024, PFO) (see full details in Appendix A.1).

This subsection introduced a novel meta-exploration method for meta-training task policies. However, it does not address task exploration. We tackle this distinct challenge in the next subsection.

## 3.2 Task Exploration Policy

While the approach in Sec. 3.1 addresses how to use a task context $\boldsymbol{c}$ effectively, it assumes such a context is already available. We now formally discuss how to obtain it via task exploration. The objective of the exploration policy $\pi^{\text{explore}}$ is to collect a single informative meta-episode $\mathcal{D}^{(K)}$ for a task $\mathcal{M}_i \sim p(\mathcal{M})$. Specifically, $\pi^{\text{explore}}$ should gather information about the task-specific dynamics of $\mathcal{M}_i$ while avoiding irrelevant or redundant exploration.

To quantify informativeness, we rely on a latent representation $\boldsymbol{\Gamma} = f_u(\mathcal{D}^{(K)}) \in \mathbb{R}^{(H d_{\text{model}}) \times K}$ computed by encoder $f_u$, with $\boldsymbol{\Gamma}_{:,k}$ denoting the representation of the $k$-th episode $\mathcal{D}_{:,k}^{(K)}$. We optimize $f_u$ (detailed in Sec. 3.3) such that the cosine similarity $S_C(\boldsymbol{\Gamma}_{:,k}, \boldsymbol{\Gamma}_{:,k'})$ is inversely proportional to how useful it is to

collect episode $\mathcal{D}^{(K)}_{:,k'}$, after having already collected $\mathcal{D}^{(K)}_{:,k}$, for any $k, k' \in [K], k < k'$.[2] Therefore, an informative meta-episode contains a set of diverse, orthogonal episodes with an average pairwise similarity of approximately zero.

We leverage this property of $\mathbf{\Gamma}$ to optimize $\pi^{\text{explore}}$. Let the average causal similarity score $\boldsymbol{d}_k = \frac{1}{K} \sum_{k'=1}^{k-1} S_C(\mathbf{\Gamma}_{:,k}, \mathbf{\Gamma}_{:,k'})$ measure how similar the current $k$-th episode is to the history of previously collected episodes $\{1, \ldots, k-1\}$, while hiding similarities to future episodes. To encourage task exploration, at the end of the $k$-th episode, the agent receives the task-dependent intrinsic reward

$$\tilde{r}^k = \exp\left(-\frac{1}{\sigma} \boldsymbol{d}_k^2\right), \tag{4}$$

where $\sigma$ is a constant, and a reward of zero at each non-terminal timestep. Intuitively, this reward is maximized at $\boldsymbol{d}_k = 0$, encouraging the agent to minimize the similarity between episode $k$ and the previous $(k-1)$ episodes. In practice, we train $\pi^{\text{explore}}$ to maximize this intrinsic reward using PPO. A more detailed overview of how we create the average similarity scores is given in Appendix A.2.

For $\pi^{\text{explore}}$ to collect a useful episode, it must have knowledge of the episodes it has already collected. To create an exploration-focused task context, we denote $\mathcal{D}^{(:k,:t)}$ to be an incomplete meta-episode, containing $(k-1)$ complete episodes, and the first $t$ timesteps of the $k$-th episode, for $k \in [K], t \in [H]$. We then condition the in-context policy $\pi^{\text{explore}}$ on the history $\mathbf{\Gamma}^{(:k,:t)} = f_u(\mathcal{D}^{(:k,:t)})$ and explore episode $k$ by taking action $a^k_{t+1} \sim \pi^{\text{explore}}(\cdot \mid s^k_{t+1}, \mathbf{\Gamma}^{(:k,:t)})$. The overall task exploration process is visualized in Fig. 2.

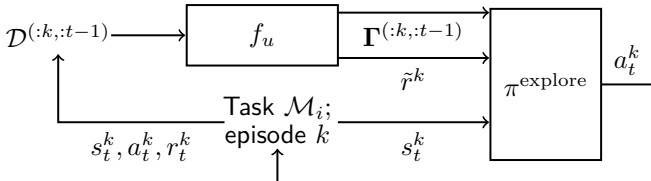

Figure 2: Task exploration. Policy $\pi^{\text{explore}}$ explores a task $\mathcal{M}_i$ by collecting $K$ episodes. At each timestep $t$ of each episode $k$, $\pi^{\text{explore}}$ is conditioned on the current state $s^k_t$ and the latent representation $\mathbf{\Gamma}^{(:k,:t-1)}$ of previous timesteps and episodes. Its objective is to take actions that maximize the intrinsic exploration rewards $\tilde{r}^k$. Both $\mathbf{\Gamma}^{(:k,:t-1)}$ and $\tilde{r}^k$ are computed by the encoder $f_u$ from the data collected.

### 3.3 Learning to Explore by Reconstructing Tasks

The exploration policy described in the previous section relies heavily on the assumption that the latent space $\mathbf{\Gamma}$ produced by $f_u$ encodes the informativeness of collected data in its geometry. In this section, we formally introduce a training method for $f_u$ that ensures this assumption holds.

Throughout this section, our focus is centered on encoders $g$ and $f_u$, which enable task solving and task exploration, respectively. Let $\mathbf{B} \in \mathbb{R}^{Q \times H' \times K}$ be a tensor containing $Q$ meta-episodes collected from task $\mathcal{M}_i$. Each slice $\mathbf{B}_j$ (shorthand notation for $\mathbf{B}_{j,:,:}$), for $j \in [Q]$, corresponds to a distinct meta-episode $\mathcal{D}^{(K)}$, as defined in Section 2. While $\mathcal{D}^{(K)}$ was used to represent a single independent meta-episode, we adopt this change in notation to emphasize that each $\mathbf{B}_j$ is a component of the batch tensor $\mathbf{B}$.

Recall that, given $\mathbf{B}_j$, we use $\boldsymbol{c} = g(\mathbf{B}_j)$ for task-solving and $\mathbf{\Gamma} = f_u(\mathbf{B}_j)$ for task exploration. Encoders $g$ and $f_u$ are optimized based on a key assumption about the data used for few-shot adaptation. To formalize this, we first define the $K$-shot adaptation objective:

**Definition 1** ($K$-Shot Adaptation). A policy $\pi$ is $K$-shot adaptable in a distribution $p(\mathcal{M})$ over MDPs if, for any $\mathcal{M}_i \sim p(\mathcal{M})$, $\pi$ becomes optimal in $\mathcal{M}_i$ after at most $K$ episodes collected from $\mathcal{M}_i$.

Finding these $K$ episodes directly can be difficult. Instead, we propose the following intuition: if a policy is $K$-shot adaptable in a task $\mathcal{M}_i$ when using the $QK$ episodes in $\mathbf{B}$, then Definition 1 implies that $\mathbf{B}$ contains

---

[2]With a slight abuse of notation, we use $[N]$ to mean the sequence $[1, 2, \ldots, N]$ for any integer $N$.

the $K$ episodes required for $K$-shot adaptation. More precisely, there exists a meta-episode $\mathbf{B}_j$ composed of these $K$ episodes. We create a practical algorithm from this idea by making the following assumption on the relationship between $\mathbf{B}$ and $\mathbf{B}_j$ (see Appendix A.3 for a more formal discussion):

**Assumption 1** (Linear Task Reconstruction). *Let $\mathbf{B} \in \mathbb{R}^{Q \times H' \times K}$ contain $QK$ episodes collected from an MDP $\mathcal{M}_i$ such that $\mathbf{B}$ is sufficient for $K$-shot adaptation. Furthermore, let $\mathbf{B}_j \in \mathbb{R}^{H' \times K}$ be the slice of $\mathbf{B}$ that contains these $K$ necessary episodes. Then, we assume there exists a tensor $\mathbf{C} \in \mathbb{R}^{Q \times K \times K}$ such that*

$$\mathbf{B}_j \mathbf{C} \approx \mathbf{B}, \tag{5}$$

*where the operation denotes the batch matrix multiplication $\mathbf{B}_j \mathbf{C}_{q,:,:} \approx \mathbf{B}_{q,:,:}$ for all $q \in [Q]$.*

One reason for keeping this assumption linear, besides simplicity, is to create an information bottleneck for task reconstruction. While an expressive enough non-linear reconstruction could reconstruct $\mathbf{B}$ from any (non-informative) meta-episode, our linear transformation ensures that successful reconstruction relies on the content of $\mathbf{B}_j$.[3]

We leverage Assumption 1 to measure how effective a given meta-episode $\mathbf{B}_j$ is for $K$-shot adaptation, quantifying the potential of computing a task context vector $\boldsymbol{c}$ from $\mathbf{B}_j$ instead of $\mathbf{B}$. We refer to the process of probing the linear relationship between $\mathbf{B}_j$ and $\mathbf{B}$ by searching for coefficients $\mathbf{C}$ as *task-reconstruction*. To find a suitable tensor $\mathbf{C}$, we define the task-reconstruction loss

$$\mathcal{L}_{\text{t-rec}} = \mathbb{E}_{\mathcal{M}_i \sim p(\mathcal{M}), \, \mathbf{B} \sim P_{\mathcal{M}_i}^{\pi^{\text{explore}}}, \, j \sim U(Q)} \left[ \text{MSE} \left( \mathbf{B}_j \mathbf{C}, \, \mathbf{B} \right) \right], \tag{6}$$

where $U(Q)$ is a uniform distribution over the integers $\{1, 2, \ldots, Q\}$, MSE is the mean squared error, and $\mathbf{C}$ is computed from $\mathbf{B}$ and $\mathbf{B}_j$ using encoder $g$ (see Sec. 3.4). By minimizing this loss, the agent learns to find $\mathbf{C}$, which is then used to assess the quality of $\mathbf{B}_j$.

Recall that our ultimate goal is to train a function $f_u$ that can reliably guide $\pi^{\text{explore}}$ to collect an informative meta-episode $\mathbf{B}_j$ during online task exploration. While the linear map $\mathbf{C}$ assesses the quality of $\mathbf{B}_j$, a critical limitation is that $\mathbf{C}$ cannot be computed during online exploration, where the agent is limited to collecting a single meta-episode. Specifically, computing $\mathbf{C}$ is equivalent to finding an approximate solution to the system of linear equations $\mathbf{B}_j \mathbf{C} = \mathbf{B}$, which requires access to the full batch $\mathbf{B}$.

Our solution is to treat $\mathbf{C}$ strictly as an offline "ground truth" metric that supervises the learning of $f_u$. To this end, we aim to optimize $f_u$ to create a latent exploration space that acts as an online proxy for the information contained in $\mathbf{C}$. The first step is to define a target $\delta_j$ that quantifies the effectiveness of the linear reconstruction in Eq. 5 for a given meta-episode $\mathbf{B}_j$. We design $\delta_j$ such that $\delta_j \approx 0$ implies $\mathbf{B}_j$ is a good approximation of $\mathbf{B}$:

$$\delta_j = 1 - \exp\left( -\xi \overline{\left( \mathbf{B}_j \mathbf{C} - \mathbf{B} \right)^2} \right), \tag{7}$$

where $\xi$ is a constant and the $\overline{(\cdot)}$ operator denotes the element-wise mean of a tensor.

We use this target to shape the latent exploration space $\boldsymbol{\Gamma} = f_u(\mathbf{B}_j)$. By optimizing $f_u$ such that the cosine distance between episode representations $\boldsymbol{\Gamma}_{:,k}$ and $\boldsymbol{\Gamma}_{:,k'}$ is approximately $\delta_j$ (for $k, k' \in [K]$), we encode reconstruction quality directly into the latent angles. Therefore, collecting a meta-episode where all vectors in $\{\boldsymbol{\Gamma}_{:,k}\}_{k=1}^{K}$ are approximately orthogonal becomes equivalent to searching for a $\mathbf{B}_j$ that closely approximates $\mathbf{B}$. This property is important because it enables $K$-shot adaptation, whereas the alternative approach of collecting $Q$ meta-episodes is impractical in online settings.

---

[3]Regardless, Eq. 5 and our entire framework can naturally extend to non-linear settings by replacing the linear map $\mathbf{C}$ with a parameterized non-linear function. This change should not compromise learning, provided that the expressivity of the non-linear transformation is carefully tuned. However, we find that the linear approximation is sufficient for effective task reconstruction in the domains evaluated in this work, while also being simpler and computationally cheaper. For the sake of completeness, we provide a formal definition of the non-linear extension in Appendix C and an ablation study in Appendix G.1.

To train $f_u$ such that the reconstruction score $\delta_j$ is explicitly encoded into the angles between episode representations, we introduce the loss

$$\mathcal{L}_{\text{contr}} = \mathbb{E}_{\mathcal{M}_i \sim p(\mathcal{M}), \, \mathbf{B} \sim P_{\mathcal{M}_i}^{\pi^{\text{explore}}}, \, j \sim U(Q)} \left[ \text{MSE} \left( S_C(\mathbf{\Gamma}^{\mathsf{T}}, \mathbf{\Gamma}), \, \boldsymbol{A}(\delta_j) \right) \right], \tag{8}$$

$$\text{where } \mathbf{\Gamma} = f_u(\mathbf{B}_j),$$

and $\boldsymbol{A}(\delta_j) \in \mathbb{R}^{K \times K}$ is the matrix with $\delta_j$ on the off-diagonals and ones on the diagonal. Eq. 8 is a form of contrastive learning for RL (Eysenbach et al., 2022; Erraqabi et al., 2022), where $\mathbf{\Gamma}_{:,k}$ and $\mathbf{\Gamma}_{:,k'}$ are pushed apart or pulled together according to the soft similarity signal $\delta_j$. Since this contrastive objective operates over cosine similarity, pushing dissimilar representations corresponds to maximizing the angle between them, while pulling corresponds to aligning similar representations by minimizing the angle.

Through this training regime, the encoder $f_u$ is optimized to be used as described in Sec. 3.2. Specifically, it can train an exploration policy $\pi^{\text{explore}}$ to find an informative meta-episode $\mathbf{B}_j$ (denoted as $\mathcal{D}^{(K)}$ during online task exploration).

## 3.4  Architecture and Optimization

Having defined the learning objectives for task exploration and reconstruction, we are now in a position to describe the architecture and optimization of the encoder $g$ used to obtain the latent task context vector $\boldsymbol{c}$.

To accommodate both sequential online task exploration and offline batch learning, we model $g$ as a dual-transformer architecture.[4] Its first component is $f_u$: the online encoder that separates the informative meta-episodes from the non-informative ones collected during task exploration. Since $f_u$ is constrained to encode meta-episodes timestep by timestep, as they are being collected, we model it as a unidirectional transformer with parameters $\omega_u$.

To create more informative contexts once a set of data has been collected, we leverage the second component of $g$: a bidirectional transformer $f_b$ of size $d_{\text{model}}$ with parameters $\omega_b$. Given a tensor $\mathbf{B}$ of meta-episodes, $f_b$ computes two latent representations offline: $(\boldsymbol{z}, \mathbf{Z}) = f_b(\mathbf{B}; \omega_b)$. On one hand, vector $\boldsymbol{z} \in \mathbb{R}^{d_{\text{model}}}$ captures the shared task structure across distribution $p(\mathcal{M})$. To achieve this, $\boldsymbol{z}$ is computed from a batch of $\mathbf{B}$-like tensors collected from different tasks (see Appendix A.4.1). On the other hand, tensor $\mathbf{Z} \in \mathbb{R}^{Q \times (HK) \times d_{\text{model}}}$ is a task-specific representation of $\mathcal{M}_i$, with $\mathbf{Z}_j \in \mathbb{R}^{(HK) \times d_{\text{model}}}$ encoding meta-episode $\mathbf{B}_j$, for any $j \in [Q]$. While $\mathbf{Z}$ and $\mathbf{\Gamma}$ appear to have similar roles, we note that $\mathbf{\Gamma}$ is optimized to learn representations that are only useful for task exploration. It may therefore fail to capture structure that is important for solving tasks, but irrelevant for exploration.

For a given meta-episode $\mathbf{B}_j$, we compute the final context vector $\boldsymbol{c} \in \mathbb{R}^{(HKd_{\text{model}})}$ as a function of $\boldsymbol{z}$, $\mathbf{Z}_j$, and $\mathbf{\Gamma}$.[5] Specifically, for a function $h_{\omega_h}$ with parameters $\omega_h$, we encode $\mathbf{B}_j$ into

$$\boldsymbol{c} = g_\omega(\mathbf{B}_j) = h_{\omega_h}(\boldsymbol{z}, \mathbf{Z}_j, \mathbf{\Gamma}), \tag{9}$$

where $\mathbf{\Gamma} = f_u(\mathbf{B}_j; \omega_u)$, and $g_\omega$ has the concatenated parameters $\omega = \omega_u \oplus \omega_b \oplus \omega_h$. In practice, $h$ consists of two attention layers. Our encoder architecture is shown in Fig. 3.

To train $g$, we attach two pre-training heads (shown in Fig. 4). The task-reconstruction head $h_{\text{t-rec}}$, parameterized by $\omega_{\text{t-rec}}$, computes the coefficient tensor $\mathbf{C} = h_{\text{t-rec}}(g_\omega(\mathbf{B}), g_\omega(\mathbf{B}_j); \omega_{\text{t-rec}})$ required for the low-rank approximation in Eq. 6. The reconstruction head $h_{\text{rec}}$, with parameters $\omega_{\text{rec}}$, is used during the optimization of $g_\omega$ to reconstruct data from corrupted inputs. Specifically, $h_{\text{rec}}$ projects the context $\boldsymbol{c}$ computed from masked meta-episodes back to the input space, attempting to predict the masked timesteps, which is a

---

[4]Due to overlapping terminology, LaSER can be seen as performing in-context learning in two senses. From the meta-RL perspective, LaSER uses in-context policies conditioned on contextual information. From a different perspective, these context vectors are computed using transformers, which, due to their self-attention mechanism, are considered to be in-context learners.

[5]The reader might note that the $(HKd_{\text{model}})$-dimensional context vector $\boldsymbol{c}$ does not necessarily reduce the size of the input $\mathbf{B}_j$. A high-dimensional $\boldsymbol{c}$ is beneficial since preserving the structure of $\mathbf{B}_j$ (i.e., the $HK$ timesteps) makes it simpler to train transformers $f_u$ and $f_b$. Despite this, $\boldsymbol{c}$ is still a useful representation of $\mathbf{B}_j$. Each of the $HK$ output timesteps has been computed from its corresponding input timestep and the rest of the input sequence. This enables dimensionality reduction later on. Specifically, we use the task policy $\pi$ to reduce $\boldsymbol{c}$ to a $d_{\text{model}}$-dimensional vector.

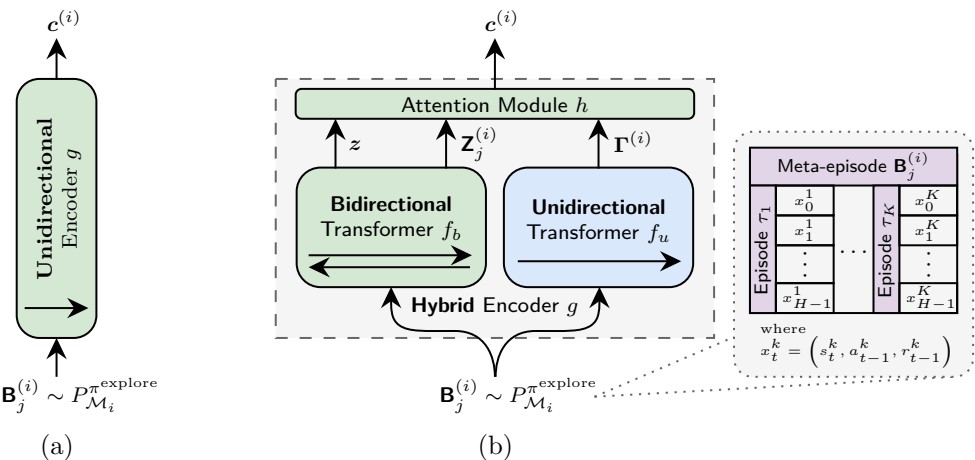

Figure 3: Two types of meta-RL encoders. (a) Unidirectional encoder, common in meta-RL. It processes task exploration data online, step by step, as it is being collected. (b) LaSER's encoder $g$. We enhance standard architectures by adding a bidirectional encoder. Encoder $g$ can be used online for task exploration or offline to compute context $c^{(i)}$. We use $\cdot^{(i)}$ to differentiate between meta-episodes and representations belonging to different tasks. Note that we define timesteps $x_t^k$ to contain the current state $s_t^k$, but the previous action $a_{t-1}^k$ and reward $r_{t-1}^k$, with $x_1^k = (s_1^k, \_, \_)$. This simplifies the training of unidirectional encoders.

common approach for pre-training bidirectional transformers. Note that both $h_{\text{t-rec}}$ and $h_{\text{rec}}$ are only used during meta-training. For environment interactions, these heads are replaced by policies $\pi^{\text{explore}}$ or $\pi$.

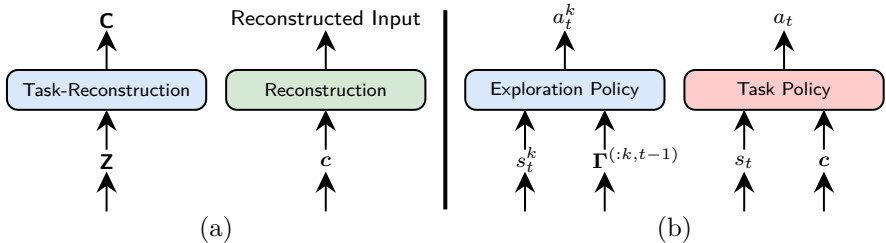

Figure 4: LaSER's pre-training and policy heads. (a) Task-reconstruction head $h_{\text{t-rec}}$ and reconstruction head $h_{\text{rec}}$, used for pre-training encoder $g$ for task exploration and task learning, respectively. (b) Policies $\pi^{\text{explore}}$ and $\pi$, used for task exploration and task solving, respectively.

We train $g_\omega$ using masked self-supervision (Devlin et al., 2019; Lewis et al., 2019). The encoder learns useful representations by adding noise to the input, reconstructing the original input, and optimizing a reconstruction loss $\mathcal{L}_{\text{rec}}$. We extend this idea and optimize the entire architecture using the loss

$$\mathcal{L}_{\text{LaSER}}(\omega, \omega_{\text{rec}}, \omega_{\text{t-rec}}) = c_{\text{rec}}\mathcal{L}_{\text{rec}}(\omega, \omega_{\text{rec}}) + c_{\text{t-rec}}\mathcal{L}_{\text{t-rec}}(\omega, \omega_{\text{t-rec}}) + c_{\text{contr}}\mathcal{L}_{\text{contr}}(\omega) + c_{\mathcal{R}}\mathcal{R}(\omega), \qquad (10)$$

where $c_{\text{rec}}, c_{\text{t-rec}}, c_{\text{contr}}, c_{\mathcal{R}}$ are weighting coefficients, and $\mathcal{R}$ is a regularization term. Following Piratla et al. (2020), $\mathcal{R}$ enforces a soft orthogonality constraint between the global representation $z$ and the task-specific representation $\mathbf{Z}_j$. Intuitively, $\mathbf{Z}$ is encouraged to avoid capturing the global information already contained in $z$, and instead focus on task-specific structure. Full details regarding our architecture, composed objective, and optimization approach can be found in Appendix A.4, with specific implementation details provided in Appendix D.

### 3.5 Meta-Training and Meta-Testing

The LaSER meta-training algorithm contains two training phases, as shown in Alg. 1. We first train the exploration policy $\pi_\phi^{\text{explore}}$, parameterized by $\phi$, and the encoder $g_\omega$, parameterized by $\omega$, for $N_{\text{explore}}$ iterations. Note that these components are trained together because they are interdependent. We alternate between optimizing one while keeping the other fixed. In the second stage, these two components are fixed and only the task policy $\pi_\theta$, parameterized by $\theta$, is optimized for $N_{\text{task}}$ iterations.

---

**Algorithm 1** LaSER Meta-Training

    **Input** $p(\mathcal{M})$, task distribution
    **Output** $\pi_\theta$, task policy; $\pi_\phi^{\text{explore}}$, exploration policy; $g_\omega$, encoder; $\hat{z}$, shared component
1: $\theta, \phi, \omega, \omega_{\text{rec}}, \omega_{\text{t-rec}} \leftarrow$ initialize randomly
2: **for** $n \in [N_{\text{explore}}]$ **do**
3:      $\omega, \omega_{\text{rec}}, \omega_{\text{t-rec}} \leftarrow \text{train\_encoder}(p(\mathcal{M}), \pi_\phi^{\text{explore}}, g_\omega, \omega_{\text{rec}}, \omega_{\text{t-rec}})$         $\triangleright$ Alg. 3
4:      $\phi \leftarrow \text{train\_exploration\_policy}(p(\mathcal{M}), \pi_\phi^{\text{explore}}, g_\omega)$         $\triangleright$ Alg. 4
5: **end for**
6: $\mathcal{B} \leftarrow \left\{ \mathbf{B} \sim P_{\mathcal{M}_i}^{\pi_\phi^{\text{explore}}} \middle| \mathcal{M}_i \sim p(\mathcal{M}) \right\}$         $\triangleright$ Collect a batch of data from multiple tasks
7: $\hat{z}, \_ \leftarrow f_b(\mathcal{B}; \omega_b)$
8: **for** $n \in [N_{\text{task}}]$ **do**
9:      $\theta \leftarrow \text{train\_task\_policy}(p(\mathcal{M}), \pi_\theta, \pi_\phi^{\text{explore}}, g_\omega, \hat{z})$         $\triangleright$ Alg. 5
10: **end for**
11: **return** $\pi_\theta, \pi_\phi^{\text{explore}}, g_\omega, \hat{z}$

---

Recall that $z$ captures global information, so it requires data from a batch of tasks, which is not available in the online phase. Therefore, we compute a fixed $\hat{z}$ from the pre-training data and use it to train the task policy and perform meta-testing. With a slight abuse of notation, we then use $g_\omega(\mathcal{D}^{(K)}, \hat{z}) = h_{\omega_h}(\hat{z}, \mathbf{Z}_j, \mathbf{\Gamma})$ to denote $\mathbf{Z}_j$ and $\mathbf{\Gamma}$ are implicitly computed from $\mathcal{D}^{(K)}$, while $\hat{z}$ is given separately. Alg. 2 shows the meta-testing phase where, for any $\mathcal{M}_i \sim p(\mathcal{M})$, the meta-trained agent first collects $K$ episodes and computes the latent context $c$. Then, it uses $c$ to find the optimal policy for $\mathcal{M}_i$.

---

**Algorithm 2** LaSER Meta-Testing

    **Input** $p(\mathcal{M})$, task distribution;
         $\pi_\theta$, task policy; $\pi_\phi^{\text{explore}}$, exploration policy; $g_\omega$, encoder;
         $\hat{z}$, shared component
1: **for** $\mathcal{M}_i \sim p(\mathcal{M})$ **do**
2:      $\mathcal{D}^{(K)} \sim P_{\mathcal{M}_i}^{\pi_\phi^{\text{explore}}}$         $\triangleright$ Sample exploration meta-episode
3:      $c \leftarrow g_\omega(\mathcal{D}^{(K)}, \hat{z})$
4:      $\tau \sim P_{\mathcal{M}_i}^{\pi_\theta(a|s,c)}$         $\triangleright$ Sample exploitation episode
5:      Measure return in $\tau$
6: **end for**

---

## 4 Related Work

**Meta-RL.** The earlier successes of modern meta-RL start with MAML (Finn et al., 2017) and RL$^2$ (Duan et al., 2016; Wang et al., 2016a). MAML, together with other MAML-inspired algorithms (Sung et al., 2017; Li et al., 2017; Gupta et al., 2018; Zintgraf et al., 2019), are gradient-based methods for meta-training policies that adapt to new tasks by taking a small number of task-specific gradient steps. Gradient-based approaches usually implement an explicit dual-loop algorithm that follows the standard meta-learning paradigm of adapting to tasks in an inner loop, while meta-learning adaptation strategies in an outer loop. On the other side of the spectrum, RL$^2$ is an in-context meta-RL approach, meta-training a policy to behave like an RL

algorithm that learns from collected task contexts (Laskin et al., 2023; Moeini et al., 2025). This learning is usually represented by a forward pass through the meta-trained policy, with no task-specific weight updates. Recent meta-RL techniques, including ours, follow this paradigm of first identifying task dynamics and then adapting a task-agnostic policy to them. Generally, in-context learning may lead to higher meta-testing sample-efficiency than gradient-based methods, which is ideal in few-shot adaptation (Beck et al., 2023b).

**In-Context Policies.** To learn task-specific policies, $RL^2$ uses a recurrent architecture that implicitly conditions the policy on task history. Later works make this conditioning more explicit (Rakelly et al., 2019; Zhou et al., 2019; Zintgraf et al., 2019). Specifically, they show that in-context policies can be trained using standard RL optimization by simply augmenting the input state with task contexts. Similar ideas have been explored in the closely related area of unsupervised representation learning for RL (Igl et al., 2018; Papoudakis et al., 2021; Botteghi et al., 2025). This approach has since become standard in meta-RL, with subsequent research focusing more on learning informative task contexts than on novel architectures or optimization algorithms for in-context policies. Recently, Beukman et al. (2024) observed that this simple method may struggle when there is high variation across the optimal task-specific policies, which can arise in complex task distributions. As an alternative, Beck et al. (2023a) and Beukman et al. (2024) propose that meta-training hypernetworks (Ha et al., 2017) may lead to better generalization. Therefore, they introduce hypernetworks that take the task context as input and generate the weights of a context-dependent policy. While this architectural change may better leverage task contexts, their approach is still limited by the use of standard RL optimization, which constrains task space exploration. Additionally, this method inherits issues related to hypernetworks, such as slow and unstable training (Ortiz et al., 2023; Chauhan et al., 2024; Beukman et al., 2024).

In contrast, our proposed meta-reward is not an adaptation of standard learning methods to meta-RL, but is explicitly designed for the meta-RL setting. Moreover, it is architecture-agnostic, introduces little overhead, and allows in-context policies to be optimized through standard RL. This may simplify solving meta-RL problems, as practitioners can rely on stable and well-understood RL algorithms.

**Exploration in Meta-RL.** A considerable body of literature also focuses on exploration in meta-RL. As opposed to standard RL, exploration strategies for few-shot meta-RL can be learned from interactions with the environment and then applied to new tasks. Since identifying and solving RL tasks requires exploration, all meta-RL algorithms learn to explore, at least implicitly. However, several works have shown the benefits of explicitly learning to explore. Rakelly et al. (2019) use posterior sampling to explore. Zintgraf et al. (2021b) consider Bayes-optimal policies, which optimally trade off exploration and exploitation, and meta-learn approximations of such policies. They later extend their work in Zintgraf et al. (2021c) by encouraging the agent to explore novel hyper-states during meta-training. Some other approaches make exploration more efficient by structuring the exploration space through contrastive learning (Fu et al., 2021; He et al., 2024; Yu et al., 2024) or information gain (Liu et al., 2021; Jiang et al., 2021; Zhang et al., 2021). Similarly, Chu et al. (2024) construct an exploration space through task clustering, enabling exploration via a divide-and-conquer strategy of first identifying a task's cluster, then the task itself. Gradient-based meta-RL can also explicitly learn to explore. Gupta et al. (2018) explore by adding structured, meta-learned noise to the policy. Similarly, Stadie et al. (2018) enhance MAML and $RL^2$ by adding an exploration term to the meta-RL objective. Gurumurthy et al. (2020) add self-supervised objectives to lower variance during exploration. Finally, several of the works discussed improve exploration even further by decoupling the exploration and task-solving policies (Zhou et al., 2019; Gurumurthy et al., 2020; Liu et al., 2021; Fu et al., 2021; Chu et al., 2024). Norman & Clune (2024) train such a decoupled policy to meta-learn exploration strategies that know when to sacrifice environmental rewards during exploration to maximize returns during exploitation.

LaSER also collects data using a decoupled exploration policy and then meta-learns a structured exploration space. However, it uses a novel objective that encourages the collection of a single meta-episode, which serves as a low-rank linear representation of a larger dataset drawn from the same task. An important distinction to numerous previous works is that task exploration depends only on the structure of the data, while being agnostic to the RL objective of the task policy. This may be advantageous for out-of-distribution adaptation, where meta-test tasks may have goals that differ from those seen in meta-train tasks. In such scenarios, data collected for maximizing meta-training return might not always be relevant during meta-testing.

Norman & Clune (2024) is an example of recent work that, similarly to us, attempts to meta-learn structured task exploration. A critical difference, however, is their limitation in collecting multiple exploration episodes effectively. While LaSER explicitly optimizes a policy that avoids redundancy within a meta-episode, "First-Explore does not actively explore to enable future exploration" (Norman & Clune, 2024). Consequently, LaSER is primarily designed for settings with an exploration budget of $K > 1$, where information from early episodes enables more effective or diverse exploration in subsequent episodes.

**Meta-Learning Contexts.** Once task data has been collected, a straightforward way to obtain informative contexts is through recurrent neural networks (Duan et al., 2016; Wang et al., 2016a). More sophisticated methods include meta-learning latent representations of value functions (Rakelly et al., 2019) or MDP dynamics (Zhou et al., 2019; Zintgraf et al., 2021b;c), model-based meta-RL (Clavera et al., 2018; Nagabandi et al., 2018), hybrid techniques that combine in-context and gradient-based methods (Imagawa et al., 2022), using permutation variant and invariant sequence models (Beck et al., 2024), or enhancing tasks by incorporating language instructions (Bing et al., 2023). Attention mechanisms (Bahdanau et al., 2015) and transformers (Vaswani et al., 2017) have also been adopted by the meta-RL community. These architectures' success in in-context learning, long-sequence modeling, and efficient parallelizable training aligns well with the needs of in-context meta-RL. Earlier research focused solely on meta-learning through attention (Mishra et al., 2018), whereas more recent work used transformer architectures (Melo, 2022; Xu et al., 2022b; Lee et al., 2023; Shala et al., 2024; Grigsby et al., 2024). Moving beyond architecture, Zhou et al. (2025) emphasize the importance of measuring the reliability of task contexts to avoid ambiguous or out-of-distribution contexts. The scope of their uncertainty estimator shares similarities with our goals of measuring data quality through task reconstruction. However, while their approach is built for offline meta-RL for one-shot adaptation, we tackle the online few-shot adaptation scenario.

A limitation of previous works that attempt to meta-learn task dynamics is that only unidirectional encoders are used. This constraint arises naturally since task exploration and task learning are coupled. Specifically, task data must be encoded online while it is being collected, in order to guide the exploration policy at the next timestep. An obvious limitation is that only interactions between the current and past timesteps are considered. LaSER improves this design by using an additional bidirectional encoder that also considers future timesteps, which could possess useful information and lead to richer representations (Devlin et al., 2019; Banino et al., 2022). While the aforementioned constraint cannot be avoided during task exploration, it need not restrict the computation of the final task context once all exploration data has been collected.

Another critical distinction between LaSER and prior work lies in the structural assumptions. Many meta-RL algorithms rely on the implicit assumption that task dynamics can be effectively captured simply by encoding interaction histories into a latent variable via the agent's learning mechanism (e.g., recurrence, variational inference, attention). However, without explicit structural constraints, it is unclear whether these unstructured representations can generalize to the entire task distribution or apply to new tasks. Additionally, these representations often optimize for state or reward prediction, without encoding explicit signals to aid task exploration. In LaSER, we instead make an explicit low-rank linear reconstruction assumption (Assumption 1). The assumption that a small set of basis episodes is sufficient to linearly reconstruct any data collected from a specific task leads to representations that provide a more directed exploration signal. Specifically, LaSER's exploration objective is reduced to finding these basis episodes, leading to a policy that can safely ignore non-basis episodes.

## 5 Experiments

In this section, we present empirical results for LaSER. We first introduce the environment and algorithms we use in Sec. 5.1. In Sec. 5.2, we compare LaSER with other types of meta-RL algorithms on multiple benchmarks. Next, we perform two ablation studies by analyzing individual stages of the meta-RL pipeline (Fig. 1). In Sec. 5.3, we evaluate our novel approach to meta-training in-context task policies. This corresponds to LaSER's task-solving phase, which we perform using ground-truth contexts instead of task exploration and learning. Finally, we assess LaSER's task exploration and task learning stages in Sec. 5.4 by visualizing the latent task contexts computed during meta-testing.

## 5.1 Experimental Setup

### 5.1.1 Environments

We evaluate LaSER on the meta-RL benchmarks MEWA (Stoican et al., 2023), Meta-World (Yu et al., 2020; McLean et al., 2025), and MuJoCo HopperMass (Rothfuss et al., 2019; Zhou et al., 2019; Nakhaeinezhadfard et al., 2025). We briefly discuss these environments here, focusing especially on MEWA. We then provide more details in Appendix E, including the publicly available code used to implement each benchmark.

MEWA provides a distribution of tasks that share the same central idea: certain states, called critical states, can lead to mistakes. These mistakes affect the final return negatively. The probability of a mistake happening depends on both the type of critical state and the task's dynamics. For each task, meta-RL agents must find the optimal policy for avoiding high-risk mistakes while minimizing delays. Importantly, Stoican et al. (2023) ensure MEWA's task distribution has no globally optimal non-adaptive policy (i.e., no policy can zero-shot solve all tasks). Therefore, the optimal agent must collect new data and adapt. This property makes MEWA ideal for our case, allowing us to evaluate LaSER's task exploration ability.

MEWA evaluates agents on their ability to take optimal actions in different types of "critical" states. These critical states provide agents with two options. Consider a critical state $s_x$ of type $x$. The agent's first option is to take a "risky" action. This may lead to a mistake of type $x$ happening, which in turn leads to a large delay in task completion. The probability of a type $x$ mistake happening depends on the transition function of the task. The second option is a "safe" action. This provides a guaranteed small delay and leads to a state $s_{x-1}$ of type $x$. An important feature shared by all MEWA tasks is that for any $y < x$, mistakes of type $y$ are less likely to happen than mistakes of type $x$. Therefore, depending on the task, taking several safe actions before a risky action could be optimal.

We meta-train all meta-RL agents on the narrow task distribution analyzed by Stoican et al. (2023). This corresponds to a distribution $p(\mathcal{M})$ in which any task $\mathcal{M}_i \sim p(\mathcal{M})$ has four different types of critical states and can be described by a vector $\boldsymbol{p}^{(i)} \in [0, 1]^4$. For each task $\mathcal{M}_i$, the probability of making a mistake of type $x \in [4]$ is $\boldsymbol{p}_x^{(i)} \sim \mathcal{N}(\boldsymbol{\mu}_x, 0.12)$, where $\mathcal{N}$ is a normal distribution and $\boldsymbol{\mu}^\mathsf{T} = [0.38, 0.28, 0.19, 0.09]$. Note that we clip all $\boldsymbol{p}_x^{(i)}$ to $[0, 1]$. See Appendix E.1 for a more formal definition of MEWA's tasks.

We meta-test on the fixed set of 12 tasks proposed by Stoican et al. (2023). Their corresponding $\boldsymbol{p}^{(i)}$ vectors are listed in Appendix E.1, Table 4. We assign a score to each task, given by the average probability of a mistake of any type happening, i.e., $1/4 \sum_{x=1}^4 \boldsymbol{p}_x^{(i)}$. We use this as a rough estimate of the similarity between tasks, such that tasks with a similar score require similar (but not identical) optimal policies. Additionally, note that half of the meta-testing tasks are out-of-distribution (OOD) tasks. OOD tasks lie outside the meta-training distribution $p(\mathcal{M})$, so they are more challenging to explore and solve during meta-testing.

Note that no globally optimal non-adaptive policy exists for these 12 meta-testing tasks. Additionally, the difference in performance between the optimal non-adaptive and optimal adaptive policies is high enough to evaluate the agent's ability to generalize and adapt.

MEWA offers 3 baselines to compare our agents to during meta-testing. The *random* baseline represents the expected performance of an agent that takes uniformly random actions. The *task-agnostic* baseline represents an agent that always takes optimal actions in MEWA's non-critical states. These are states in which the optimal action can be computed even if the policy has no task information, i.e., for the optimal value function $V^*$, if $V^*(s, \boldsymbol{c}) \leq V^*(s)$ for all $\boldsymbol{c}$, then $s$ is a non-critical state. For any critical state, the *task-agnostic* baseline considers all actions that could potentially be optimal in a valid task, then takes one at random. Finally, the *optimal* baseline gives the expected return of the optimal meta-RL agent, i.e., the agent that optimally adapts to and solves each task.

To assess the generalizability of our algorithm, we also conduct experiments on Meta-World V2 ML10 and MuJoCo HopperMass. Meta-World is a suite of 10 meta-training and 5 meta-testing robotic manipulation tasks where the goal is hidden and must be meta-learned. To avoid reproducibility issues, we follow McLean et al. (2025) and use the V2 reward function. HopperMass is the meta-RL extension of the popular MuJoCo Hopper environment (Todorov et al., 2012). The agent's goal is to rapidly move forward, taking into account

task variations given by multiplying the body parts' mass, the joints' damping, and the ground friction with weights uniformly sampled from $[1.5^{-1}, 1.5^1]$. To assess out-of-distribution (OOD) generalization, we follow Nakhaeinezhadfard et al. (2025) and extend HopperMass to OOD meta-testing tasks with weights in $[1.5^{-1.5}, 1.5^{-1}) \cup (1.5^1, 1.5^{1.5}]$.

In some environments, such as HopperMass, low exploitation returns during task exploration may lead to the LaSER agent only "surviving" for a small number of timesteps, hindering effective data collection. To avoid exploratory episodes terminating prematurely, we let LaSER's rewards for episode $k$ be

$$\tilde{r}'^k_t = \tilde{r}^k_t + \alpha_r r_t \tag{11}$$

where $\tilde{r}^k_t$ is the intrinsic exploration reward defined in Eq. 4 (zero everywhere except when $t$ is a terminal timestep), $r_t$ is the environment reward, and $\alpha_r$ is a constant weight. For MEWA and Meta-World, we set $\alpha_r = 0$. In HopperMass, we tune $\alpha_r$ such that $r_t$ does not dominate $\tilde{r}^k_t$.

### 5.1.2 Algorithms and Meta-Training Setup

We compare our method, LaSER, with five other meta-RL algorithms, MAML (Finn et al., 2017), PEARL (Rakelly et al., 2019), VariBAD (Zintgraf et al., 2021b), DREAM (Liu et al., 2021), and First-Explore (Norman & Clune, 2024). For each benchmark, we meta-train the algorithms on tasks sampled from the meta-training distribution. For each task $\mathcal{M}_i$, the agents are allowed to collect $K$ episodes during the task exploration phase. For both MEWA and HopperMass, we use $3,000$ tasks with $K = 4$, while for Meta-World, we use the 10 tasks provided and set $K = 2$. The agents' performance on $\mathcal{M}_i$ is then given by a single exploitation episode collected by the task policy. Besides this final return, we also analyze an agent's ability to improve its performance as more data is collected. MEWA tasks have a horizon of $H = 50$, Meta-World has $H = 200$, and HopperMass has $H = 400$. All of the results we present are averaged over 10 seeds.

We meta-train LaSER in two decoupled phases. Following Alg. 1, the exploration policy $\pi_\phi^{\text{explore}}$ and encoder $g_\omega$ are first trained for $N_{\text{explore}}$ iterations. The task policy $\pi_\theta$ is then trained separately for $N_{\text{task}}$ iterations. Note, however, that for the first $N_{\text{explore}}^{\text{initial}}$ out of $N_{\text{explore}}$ iterations, we only train $f_u$ (and thus $g_\omega$), without updating $\pi_\phi^{\text{explore}}$ since its training is conditioned on representations learned by $f_u$. For these initial updates, we use data collected by a uniformly random policy.

LaSER's hyperparameters are listed in Appendix H. We selected these based on performance on the meta-training task distribution of each environment. For implementation details and hyperparameters for MAML, PEARL, VariBAD, DREAM, and First-Explore, see Appendix F.

Note that we perform our primary in-depth analysis on MEWA, as its design ensures that task exploration and adaptability are strictly necessary. In contrast, while Meta-World and HopperMass are important benchmarks, their focus is not on exploration. Therefore, their primary role in our work is to demonstrate the general applicability of our algorithm to meta-RL problems.

For the latter two benchmarks, we select only VariBAD as a baseline for two primary reasons. First, prior literature has shown VariBAD consistently outperforming MAML and PEARL in continuous control environments (Zintgraf et al., 2021b). Demonstrating improvement over VariBAD is therefore sufficient. Second, as indicated by Liu et al. (2021), benchmarks such as Meta-World and HopperMass have not been explicitly designed to test complex task exploration policies, as naive exploration can often be sufficient. Because of this, Liu et al. (2021) highlight that DREAM is not engineered for such environments and is not necessarily expected to outperform simpler baselines. We follow a similar line of reasoning and omit First-Explore as well, not unlike Norman & Clune (2024), who also do not consider this type of benchmark.

### 5.1.3 Complexity and Trade-offs

We analyze the computational complexity of meta-training each algorithm on MEWA. Specifically, we measure the total wall clock time required to collect environmental transitions and optimize all components of an algorithm. We additionally report the total time divided by the number of timesteps collected, showcasing

the average cost of interacting with the environment once and updating based on that interaction. Note that all seeds of an algorithm are meta-trained for the same number of timesteps. This duration is set by the timestep at which the last seed's performance (i.e., return or loss) stops improving.

We provide detailed measurements, averaged over 10 seeds, in Fig. 15. Regarding total time, LaSER is the most expensive, being 1.34 times slower than the next slowest algorithm, DREAM. This is followed by PEARL, and finally by MAML, First-Explore, and VariBAD. Most of LaSER's time complexity comes from its transformer encoders, which are powerful but require a high amount of data. However, LaSER appears more efficient when collecting a single timestep and updating. Specifically, LaSER is 11.93 and 5.86 times faster than DREAM and PEARL, respectively. Furthermore, it is only 1.71, 2.31, and 2.64 times slower than MAML, First-Explore, and VariBAD, respectively.

To understand the architectural complexity of LaSER, Table 5 compares the number of trainable parameters to those of the baseline algorithms. While LaSER has notably more parameters, approximately 90% of these belong to the two pre-training heads, $h_{\text{rec}}$ and $h_{\text{t-rec}}$. Since these are not used beyond the pre-training phase, LaSER agents only require approximately 8 million parameters during meta-testing and task policy meta-training. LaSER's meta-testing parameter count therefore falls between First-Explore and DREAM, with roughly 2 million fewer than the former and eight times more than the latter.

LaSER's high number of parameters is a consequence of the transformer encoder $g$, whose additional complexity enables more effective in-context computations. Moreover, since the context vector $c$ outputted by $g$ is also higher-dimensional than the ones used by PEARL, VariBAD, and DREAM, LaSER's task policy requires a larger network for extracting features from $c$. Besides this feature extractor, LaSER's policy networks are comparable in size to those of PEARL and VariBAD.

Note that, while scaling up the encoders and the latent context spaces of the baselines could arguably lead to fairer comparisons, this is generally non-trivial and impractical. Unlike LaSER's transformers, architectures based on recurrent neural networks or variational auto-encoders (Kingma & Welling, 2014) suffer from instabilities and optimization difficulties when scaled to millions of parameters. First-Explore is the only baseline that uses a transformer encoder, so we set its size to match LaSER's meta-testing architecture.

While LaSER has a higher computational cost than the baselines during meta-training, this trade-off proves advantageous during meta-testing. By shifting the computational burden of learning general representations and exploration strategies for an entire task distribution to the meta-training phase, the meta-trained agent can reduce the amount of data required during online adaptation. While other meta-RL algorithms also attempt to offload computation to meta-training, their smaller or simpler architectures or their coupled approach to learning task exploration and solving may limit their ability to scale. In contrast, LaSER's decoupled paradigm can leverage large quantities of meta-training data to learn complex exploration priors.

Given the design of LaSER, we find it necessary to draw attention to the scope of our evaluation. Specifically, it is important to distinguish between LaSER's high meta-training compute/data requirements and its highly sample-efficient $K$-shot adaptation during meta-testing.

## 5.2 Results and Analysis

### 5.2.1 Meta-Training Convergence on MEWA

To ensure a fair evaluation, we first show that all algorithms converge during meta-training. For this, we evaluate agents on MEWA's meta-training task distribution. As shown in Fig. 5, LaSER's exploration policy, encoder, and task policy converge across 10 seeds. The encoder loss is computed by Eq. 10, with Fig. 10 showing results for each of its terms. The exploration return is computed using intrinsic rewards (see Eq. 4) collected by the exploration policy. Note that we first train the encoder and exploration policy in parallel for 220 and 170 million timesteps, respectively, then the task policy for 7.5 million timesteps.

Appendix G provides convergence results on MEWA's meta-training distribution for all the baseline algorithms. All their task policies converge (Fig. 11), together with PEARL's, VariBAD's, DREAM's, and First-Explore's encoders (Fig. 12), and DREAM's exploration policy (Fig. 13). Note that MAML lacks an encoder, and DREAM is the only baseline algorithm that uses a separate exploration policy with an explicit

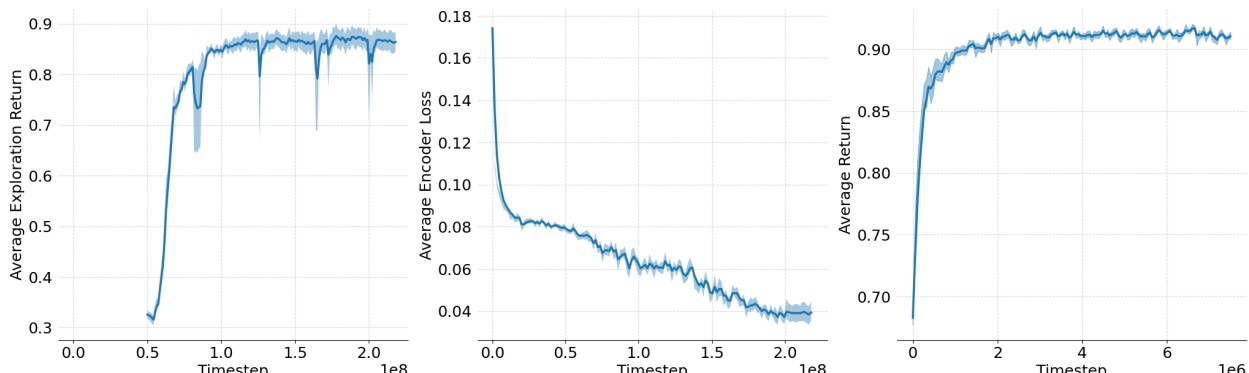

Figure 5: Performance on tasks from MEWA's meta-training task distribution. For each metric, the shaded areas report the standard error of the average return across 10 random seeds. We use exponential moving average (EMA) smoothing $y_j = \alpha_{\text{EMA}} x_j + (1 - \alpha_{\text{EMA}}) y_{j-1}$ for each point $x_j$, with $\alpha_{\text{EMA}} = 0.6$. We first meta-train the exploration policy and the encoders for $2.2e8$ timesteps, then the task policy for $7.5e6$. Additionally, we only start exploring tasks after 50 million timesteps. (left) Undiscounted exploration return of policy $\pi_\phi^{\text{explore}}$ for intrinsic exploration reward $\tilde{r}$. (middle) Encoder loss $\mathcal{L}_{\text{LaSER}}$. (right) Undiscounted return of policy $\pi_\theta$.

intrinsic reward. While First-Explore also has an exploration policy, its optimization is end-to-end via the encoder's loss.

Finally, to verify the validity of our linear task reconstruction assumption (Assumption 1), we train a variant of LaSER (on MEWA) that utilizes non-linear transformations to construct its latent exploration space, as formally introduced in Appendix C. We analyze the results of this ablation study in Appendix G.1. The non-linear variant achieves a lower task reconstruction loss, which is to be expected, as non-linear mappings are more expressive than linear ones. Crucially, however, this does not translate to an improvement in final task performance. This indicates that identifying a low-rank linear basis for a task's meta-episode space is sufficient, at least in the MEWA environment. Moreover, our competitive results on Meta-World (Sec. 5.2.3) suggest that the effectiveness of our linear assumption may also extend to environments that are highly non-linear and continuous.

### 5.2.2 Meta-Testing Performance on MEWA

We now evaluate LaSER's ability to solve MEWA's meta-testing tasks. For each algorithm, we meta-test the agent obtained at the end of meta-training. Note that before being evaluated on a task, each agent is allowed to explore for $K = 4$ episodes. We report average returns over 10 seeds in Fig. 6 and Table 1. Because of the inherent randomness in MEWA's tasks, we take an additional step to ensure the results are not due to random chance. For each seed, we repeat the entire meta-test (i.e., both task exploration and exploitation) 9 times per agent for each of the 12 meta-testing tasks. For each repetition, we collect 20 exploitation episodes, using the same context vector, and then average over their returns. In addition to the results averaged over all tasks, we report per-task performance in Fig. 14.

| LaSER (ours) | MAML | PEARL | VariBAD | DREAM | First-Explore |
|:---:|:---:|:---:|:---:|:---:|:---:|
| $\underline{0.8} \pm 0.021$ | $0.43 \pm 0.072$ | $0.67 \pm 0.32$ | $0.08 \pm 0.262$ | $0.6 \pm 0.191$ | $0.52 \pm \underline{0.018}$ |

Table 1: MEWA returns averaged across 10 seeds ($\pm$ std).

LaSER achieves the highest average returns among all meta-RL algorithms. It also proves to be the second most stable algorithm, with a post-adaptation standard deviation of just 0.021 across all seeds. LaSER is one of only two methods to outperform the *task-agnostic* baseline, while the other algorithms only outperform

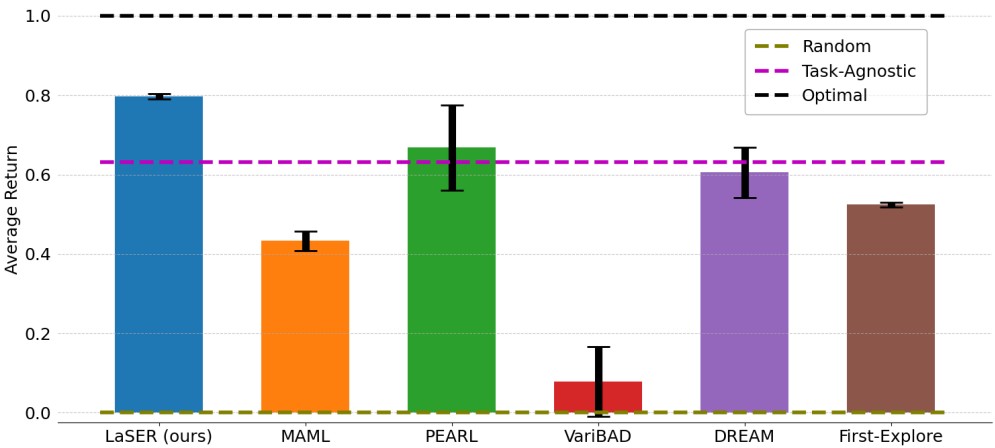

Figure 6: Average returns achieved on MEWA's meta-testing tasks. Each agent is meta-tested with an exploration budget of $K = 4$. Results are averaged over 10 seeds, with error bars indicating standard error. Note that we normalize results between the *random* and *optimal* baselines.

the *random* baseline. PEARL learns a strong global policy, but is less stable and underperforms compared to LaSER. While DREAM appears more stable, its performance is slightly below the *task-agnostic* baseline. First-Explore underperforms DREAM, but it has the lowest standard deviation, slightly below LaSER's. In contrast, MAML and VariBAD perform poorly, with VariBAD being both unstable and close in performance to the *random* baseline.

### 5.2.3 Meta-Testing Performance on Meta-World and HopperMass

In this section, we evaluate LaSER on Meta-World and HopperMass. As detailed in Sec. 5.1.2, we omit MAML, PEARL, DREAM, and First-Explore in this setting and compare only with VariBAD, the strongest baseline applicable to these environments.

Fig. 7 (left) shows that LaSER achieves competitive performance with VariBAD on Meta-World ML10. For a meta-training budget of 50 million timesteps, we present the average success rate on the ML10 meta-testing tasks, measured at intervals of $4 \cdot 10^5$ timesteps.

Note that for LaSER, the encoder and exploration policy have been pretrained for $N_{\text{explore}}$ iterations before starting the task policy training. Therefore, the learning curve of LaSER shows the post-pretraining optimization of the task policy, measuring how quickly the agent learns to solve tasks given a pretrained task exploration prior. On the contrary, VariBAD learns from scratch. We chose this type of comparison to explicitly measure adaptation efficiency and maximum asymptotic performance, rather than total sample complexity across the entire meta-training phase.

LaSER appears to have higher meta-training sample-efficiency initially. However, both algorithms end up performing similarly on Meta-World, as suggested by the error bars that largely overlap throughout meta-training. Overall, LaSER's adaptation efficiency and asymptotic performance during meta-training are comparable to VariBAD's, with both agents reaching a maximum performance of approximately 0.11.

As shown in Fig. 7 (right), LaSER performs slightly better than VariBAD in HopperMass in terms of OOD performance. We meta-train each agent for 10 million timesteps and measure OOD performance at intervals of $40,000$ timesteps. As before, LaSER's encoder and exploration policy are first pretrained for $N_{\text{explore}}$ iterations.

LaSER is more stable and adapts better to the HopperMass OOD tasks than VariBAD. Specifically, LaSER achieves a maximum average performance of 540, while VariBAD stops improving at 496. Additionally,

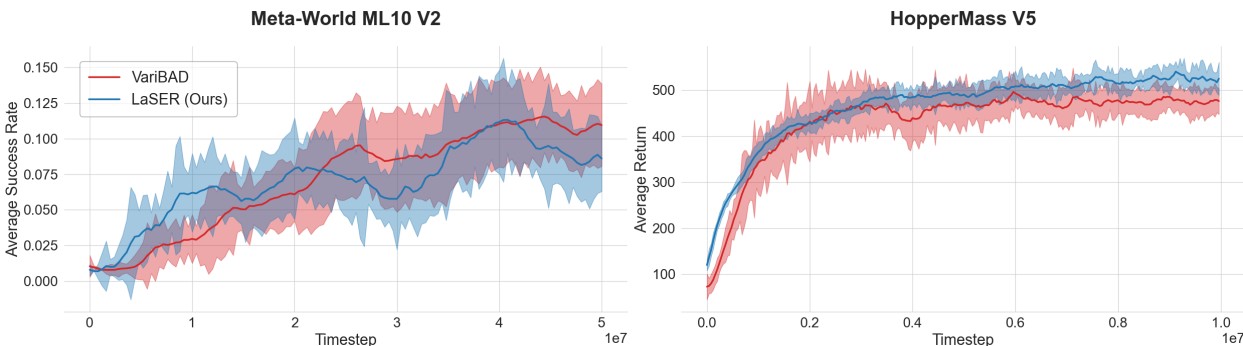

Figure 7: Meta-testing performance, averaged across 10 random seeds ($\pm$ standard error). We use EMA smoothing with $\alpha_{\text{EMA}} = 0.1$. (left) Average success rate on Meta-World ML10 meta-testing tasks for an exploration budget of $K = 2$. We measure performance every $4 \cdot 10^5$ timesteps. (right) Average return on HopperMass OOD tasks for $K = 4$, measured at intervals of $40,000$ thousand timesteps.

LaSER's superior stability is shown in the lower average standard error (26.3) compared to VariBAD (30.3) during the final million timesteps.

## 5.3 Ablation Study: Task Solving

As an ablation study, we evaluate our approach for meta-training in-context task policies, which was introduced in Sec. 3.1. The results in this section correspond solely to the task-solving phase of Fig. 1. We meta-test task policies by rolling them out before and after they receive a task context $c$. Our primary objective is to assess adaptability, i.e., a policy's ability to use $c$ to improve its performance. We quantify this as the difference between post-adaptation and pre-adaptation return.

We make two important simplifications to the standard MEWA benchmark.

- To ensure a fair comparison of task policies alone, we meta-train and meta-test without task exploration or task learning. Instead, for each task, we use a ground truth, oracle-provided context vector $c$, which is normally unavailable to the agent.

- To assess adaptability to non-stationary dynamics, we use a simpler, multi-task objective: we perform both meta-training and meta-testing on MEWA's 12 meta-testing tasks.

Despite these restrictions, MEWA's guarantee for the nonexistence of a globally optimal policy still holds. That is, only adaptive policies can achieve maximum return.

LaSER combines PPO and the proposed meta-reward $r_t^+$ (see Eq. 2) to optimize the in-context policy $\pi_\theta$. We compare it to a simpler context-aware PPO policy, where the state $s_t$ and context $c$ are first preprocessed separately, with their representations concatenated into a single vector, which is finally passed through a policy optimized through standard PPO. This corresponds to LaSER's $r_t^+$ with $\beta = 0$. We also compare to Decision Adapters (Beukman et al., 2024, DAs), which use hypernetworks to generate task-specific policies.[6] For a fair comparison, the context-aware PPO policy has the same architecture as LaSER's policy. We also use PPO to optimize DAs, so the main difference between these and the other two methods is the architectural change. Appendix F.6 provides implementation details and hyperparameters for DAs.

The meta-training results are shown in Fig. 8 and Table 2. Our method achieves the highest average return while also stabilizing meta-training. It is also the only method to find a policy that adapts optimally, which occurs in 5 out of 10 seeds. Context-aware PPO is slightly less stable, presumably because the agent seeks

---

[6]Since Beukman et al. (2024) build and evaluate their DAs with ground-truth contexts in mind, their algorithm is agnostic to the context-learning mechanism, and, as reported by the authors, sensitive to noisy contexts. We therefore only consider DAs as baselines for LaSER's task policy, and not for the entire LaSER algorithm.

a single policy that maximizes returns across all tasks. However, without task-specific adaptation, it fails to find such a globally optimal policy. Hypernetworks, on the other hand, are highly unstable. This is likely due to hypernetworks being inherently challenging to train (Ortiz et al., 2023; Chauhan et al., 2024; Beukman et al., 2024). They also achieve the lowest average return.

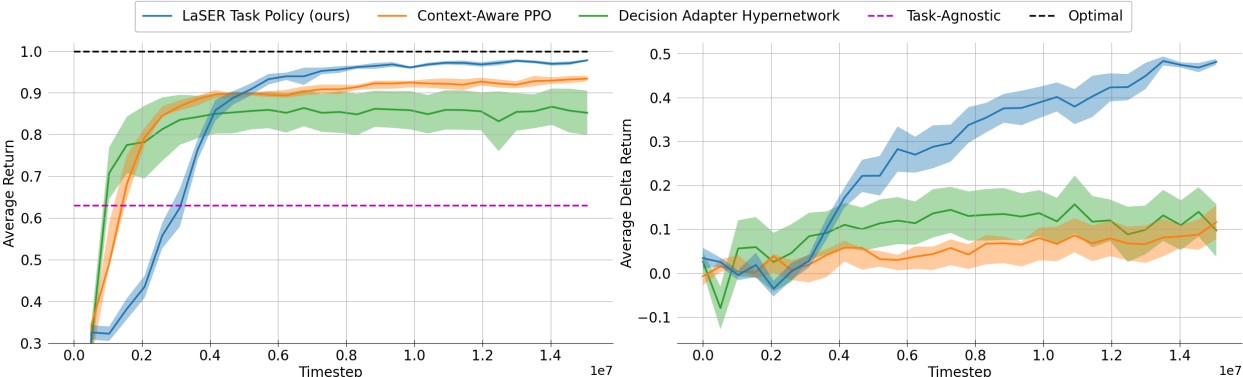

Figure 8: Task policy meta-training. Each agent uses oracle-provided task contexts $c$, and meta-trains and meta-tests on the same set of 12 MEWA tasks. The shaded areas show the standard error of the average return across 10 random seeds. (left) Post-adaptation return of task policy $\pi_\theta$, given task context $c$. All returns are normalized between MEWA's *random* and *optimal* baselines. To improve readability, we restrict the plot to the $[0.3, 1.0]$ range and exclude the *random* baseline (which is positioned at 0). (right) Adaptability, quantified as the difference in return achieved by $\pi_\theta$ with and without access to $c$.

|  | LaSER Task Policy (ours) | Context-Aware PPO | Decision Adapter Hypernetwork |
|---|---|---|---|
| Return | $\underline{0.97 \pm 0.006}$ | $0.93 \pm 0.014$ | $0.86 \pm 0.149$ |
| Delta Return | $\underline{0.47 \pm 0.019}$ | $0.07 \pm 0.067$ | $0.12 \pm 0.164$ |

Table 2: Average returns and delta returns over the last $1e6$ timesteps, averaged across 10 seeds ($\pm$ std). We meta-train and meta-test on the same 12 MEWA tasks, and use oracle-provided context vectors $c$.

The LaSER task policy also achieves greater task-adaptation success than the other two methods. After the meta-exploration phase, the agent stabilizes towards the end of meta-training. The task-conditioned policy then consistently outperforms the pre-adaptation policy. Hypernetwork-generated policies adapt better than context-aware PPO but are also more unstable. Additionally, their low average return limits the benefits of this adaptability. Context-aware PPO does not explicitly optimize for adaptation, so its adaptability is approximately five times lower than our method.

Note that, at the beginning, our method learns more slowly than context-aware PPO and the DA hypernetwork. That is, it requires more samples to pass the average return thresholds of 0.9 (for PPO) and 0.85 (for DA). We attribute this to additional meta-exploration in the state space, performed in the context of the task space. The meta-reward $r_t^+$ encourages the policy to learn how to take in-context actions that increase the gap between $V(s_t)$ and $V(s_t, c)$, for a state $s_t$ and context $c$. This is in addition to the meta-exploration required to only maximize the standard RL objective $V(s_t)$. We hypothesize that context-aware PPO and DA cannot surpass our method due to their lack of explicit task-space meta-exploration. The policies they produce optimize standard RL objectives, so they meta-explore accordingly.

We also observe that our method is nearly as computationally efficient as context-aware PPO and much faster than hypernetworks. We measure the average wall-clock time to roll out and optimize $\pi_\theta$, per iteration. On average, hypernetwork-based agents are approximately $3.3\times$ slower than ours, while our method is only $1.3\times$ slower than context-aware PPO. See Fig. 16 for a detailed comparison. We discuss how meta-rewards cause this additional overhead in Appendix A.1.2.

Finally, to verify that the benefit of our meta-reward arises specifically from the interaction between the context-usage weight $w$ and the entropy regularization $S$ (see Eq. 2), we conduct an additional ablation study. We compare our proposed objective with two simpler alternatives: a direct non-entropy-dependent penalty for not leveraging contexts effectively ($r_t - \beta\,w(s_t, \boldsymbol{c})$), and an "always-on" entropy bonus that is agnostic to the value function's context utilization ($r_t + \beta\,S[\pi](s_t, \boldsymbol{c})$). As discussed in Appendix G.2, we find that Eq. 2 outperforms both of these two simpler alternatives in terms of final returns, stability, and delta returns.

### 5.4 Ablation Study: Exploring and Learning Tasks

To better understand LaSER's ability to explore and encode tasks during meta-testing, we visualize its latent task space. We compare the task contexts computed by LaSER, PEARL, VariBAD, DREAM, and First-Explore for each of the 12 meta-testing tasks in MEWA. For each algorithm, we show results for only one of the meta-training seeds. However, we obtain similar representations for the other seeds.

We first roll out the corresponding agent's meta-trained policy, collecting 100 meta-episodes per task, with $K = 4$ episodes per meta-episode. Note that in the case of LaSER, we roll out the exploration policy $\pi_\phi^{\text{explore}}$. We then encode each meta-episode $\mathcal{D}^{(K)}$ into a latent task context vector $\boldsymbol{c}$ using the agent's meta-trained encoder. Similarly, DREAM and First-Explore collect data using their exploration policy. We visualize two-dimensional projections of task contexts, computed using t-SNE (Van der Maaten & Hinton, 2008), in Fig. 9. The 12 tasks are sorted by similarity, i.e., average mistake probability, as in Table 4.

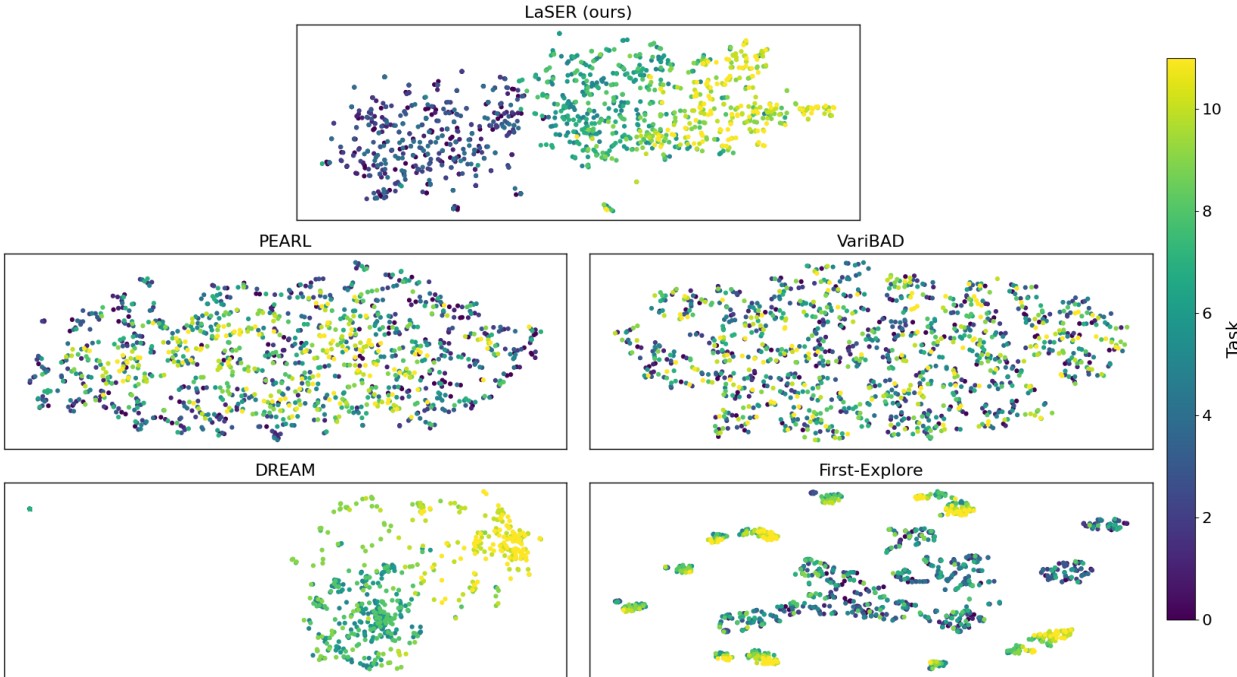

Figure 9: Latent representations of meta-episodes collected and encoded by meta-trained agents. Each of the 1200 points is a context vector belonging to one of the 12 meta-testing tasks in MEWA. The two-dimensional visualizations are computed using t-SNE. Different tasks have different colors.

For LaSER, we can find clusters of task contexts. Specifically, meta-episodes collected from the first five tasks, the next four tasks, and the final three tasks appear to be projected apart from each other. We note that the tasks in each cluster share common traits. In the first cluster, mistakes of type $x \in \{3, 4\}$ never occur. In the third cluster, $\boldsymbol{p}_x^{(i)} \geq 0.9$ for any $x \neq 4$. The second cluster's average mistake probabilities lie between those of the other two clusters. The separation appears to follow this trend. More details on the tasks' properties are shown in Table 4.

In contrast, PEARL's and VariBAD's latent contexts lack clear clustering that could distinguish data collected from different tasks. This low separation may make it more difficult to identify task-specific features. DREAM appears to follow a similar three-cluster separation as LaSER. A distinguishable feature is that data collected from any of the first five tasks collapses into a single context representation. Finally, the representations computed by First-Explore indicate the formation of only two cluster types, with contexts from tasks with mid-range mistake probabilities appearing in both. Additionally, the clusters are less densely packed than those generated by LaSER or DREAM, instead forming multiple sub-groups.

We argue that learning higher-quality clusters is not trivial. The difficulty may come from the high variance in episodes collected from the most difficult tasks. These tasks limit the exploration policy's control over the states it encounters. Additionally, the low adaptation budget of $K$ can make it difficult to separate contexts of similar tasks.

To gain an intuition about the behavior of LaSER's exploration policy, we provide a qualitative discussion of some example episodes. We choose to focus on MEWA due to its inherent interpretability and select two of its meta-testing tasks to discuss (see Appendix G.3 for visualizations and detailed discussions).

We find that, in in-distribution tasks where each type of mistake has a moderate likelihood of occurring, LaSER's policy reduces redundancy and increases diversity across $K$ exploration episodes. Specifically, LaSER appears inclined to generate action sequences that are distinct from those in earlier episodes or to experiment with the different types of risky actions available in the MEWA environment. In contrast, VariBAD tends to repeat action sequences and mistakes. In such tasks, the diversity in DREAM's exploration episodes is approximately on par with LaSER's.

However, we also find task setups where DREAM's exploration policy collapses, preventing full understanding of the task. For example, in an OOD task where mistakes of type 4 never occur but other types are highly likely, DREAM repeats the same action sequence for all $K$ episodes, as this sequence leads to high exploitation returns. In contrast, LaSER avoids this pitfall and attempts to sample distinct episodes that better capture the task context. We attribute LaSER's behavior in this setting to its meta-training objective, which explicitly encourages the agent to avoid redundancy and ignore environmental rewards during exploration.

# 6    Discussion

This section positions LaSER within the broader meta-RL domain. We discuss the implications of our design and results, the limitations of our assumptions, and the potential effectiveness of LaSER in various domains.

**Implications for Meta-RL.** An important strength of LaSER is that, in the setting where contexts are meta-learned, it outperforms other algorithms, despite none of the methods being optimally adaptive. We argue that this is a result of meta-training with higher-quality task contexts. Note that the main role of these contexts remains to provide task information for adaptation during meta-testing. However, our results suggest they may also enhance performance and sample efficiency during meta-training. We believe this secondary role should also be investigated in future work.

LaSER is built upon the unique properties and requirements of the meta-RL framework, which we approach through the lens of few-shot adaptation. As a result, we tackle challenges that are specific to meta-RL, i.e., challenges that might not exist in the broader fields of RL or meta-learning. For example, our exploration algorithm leverages a key assumption about the structure of data collected in few-shot RL environments. Similarly, LaSER's task policy is meta-trained with the explicit goal that in-context policies must outperform context-agnostic policies in a meta-RL setting. This idea, realized as a form of extended meta-exploration over the task context space, leads to almost-optimal adaptive policies in MEWA, when ground-truth contexts are available. We hope our findings inspire future research to leverage the meta-RL framework in novel ways.

While LaSER outperforms other meta-RL algorithms, it still struggles to adapt to new tasks when task contexts are meta-learned. We discuss this from the perspective of LaSER's three main components: exploration policy, encoder, and task policy. Empirical results suggest that the first two components are well-optimized for exploring and learning tasks effectively. For instance, the encoder learned to separate MEWA's meta-testing tasks into three groups. This implies that LaSER should be able to outperform its pre-adaptation

policy in these tasks. Furthermore, our results show that the task policy can become more adaptive when restricting the meta-training task distribution and using ground truth contexts. We therefore propose the hypothesis that effectively combining these three components is non-trivial.

**Scope and Limitations.** From a practical perspective, LaSER's previously discussed shift in computational burden from deployment to meta-training (Sec. 5.1.3) offers clues to the types of domains our work could be applied to. On one hand, this shift makes LaSER particularly suited to applications where task data is scarce, due to expensive or dangerous test-time interactions, while data from similar tasks can be obtained easily during meta-training. For example, sim-to-real settings can use simulations during meta-training to enable rapid adaptation during real-world deployment. On the other hand, environments with a moderate amount of data during both meta-training and meta-testing may see LaSER struggle. Its pre-training phase could fail to converge, while its quick-adaptation capabilities (if achieved) may not be necessary.

The set of applications in which LaSER is expected to be effective is also dependent on our framework's central assumption: low-rank linear task reconstruction (Assumption 1). This linear relationship has proven effective in environments such as MEWA and HopperMass, and even in the more complex robotic manipulation tasks of Meta-World. We therefore conclude that linear task reconstruction is sufficient whenever the information contained in a task can be reasonably approximated by a set of $K$ "basis" episodes. As previously mentioned, this simple low-rank approximation also acts as an information bottleneck, regularizing the model and preventing overfitting during task reconstruction.

If this linear assumption fails, then the task reconstruction loss remains high, producing a noisy target $\delta_j$. Ultimately, this means that the contrastive optimization is unable to generate a meaningful geometric structure within the latent exploration space. The meta-episode representation $\boldsymbol{\Gamma}$ will then guide the exploration policy to collect uninformative or redundant episodes, crippling $K$-shot adaptation. We suspect that such a chain of failures may occur in very complex environments with a high-dimensional state-action space, or where the task-specific dynamics cannot be captured by a linear basis.

For instance, in robotics domains with high-dimensional image observations, linear relationships between sequences of images may not reveal meaningful task information. Similarly, multi-objective environments, where each task acts as a unique combination of several goals, may exhibit exponential compositionality that cannot be approximated by a linear basis. Long-horizon tasks could also pose challenges, as small variations in early timesteps could lead to vastly different trajectories collected from the same task. Our linear assumption may then require a significantly larger exploration budget $K$, defeating the purpose of few-shot adaptation.

To evaluate the impact of our task reconstruction method on real-world application domains, a more exhaustive empirical analysis of LaSER would be required. In cases where the linear assumption proves insufficient, the non-linear variant in Appendix C may act as a more expressive alternative. We must, however, repeat our warning: an over-expressive non-linear task reconstruction head could trivially minimize the $\mathcal{L}_{\text{t-rec}}$ loss without creating a bottleneck to capture meaningful information about the data's $K$-shot potential.

# 7 Conclusion

**Summary of Contribution.** We introduced LaSER, a new approach for meta-learning RL exploration. Our results demonstrate that LaSER outperforms previous meta-RL algorithms on the MEWA benchmark and is competitive on the Meta-World and HopperMass benchmarks. They also show that LaSER can meta-learn better task clustering during exploration. Additionally, we propose a novel meta-exploration bonus for training highly adaptive task policies. This outperforms previous approaches of meta-training in-context policies in both accumulated rewards and adaptability. Since our task-solving method is agnostic to how tasks are explored or represented, it can be integrated into any in-context meta-RL algorithm and then optimized using standard RL.

**Open Challenges and Future Work.** A potential extension to LaSER, tied to the specific structure of the meta-RL framework, is dynamic exploration budgets. Although we use a fixed budget $K$ in this work, our task reconstruction objective implies that a quality metric (e.g., $\delta_j$, Eq. 7) could enable an autonomous

stopping criterion. This extension would have practical utility, allowing $K$ to be dynamically allocated on a task-by-task basis based on rank and complexity, instead of being selected by the algorithm designer.

As mentioned in our discussion, optimizing the individual components of the standard in-context meta-RL framework is not sufficient. We suggest that future research should seek to explore and understand the behavior of their joint execution.

Finally, we also note that LaSER ignores environmental reward maximization during task exploration.[7] On one hand, this keeps meta-RL agents general, discouraging them from overfitting to a single goal. On the other hand, this may cause issues when environmental rewards are necessary for the agent's survival. While we provide a simple method for combining our proposed exploration rewards with environmental rewards in Eq. 11, more complex approaches may be required to find the optimal trade-off between exploration and survival. We leave this interesting problem as future work.

### Acknowledgments

This project was supported by a grant from the Honda Research Institute to the University of Manchester. The authors would also like to acknowledge the assistance given by Research IT and the use of the Computational Shared Facility at The University of Manchester.

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

## Appendix Table of Contents

# A    Extended Methodology Details

This Appendix complements the discussion of our novel meta-RL method in Sec. 3. Here, we provide additional detail, as well as more rigorous formalisms. Additionally, for the reader's convenience, we summarize our main notation and definitions in Table 3.

## A.1    Policy Optimization in Meta-RL

LaSER uses the proximal policy optimization (Schulman et al., 2017, PPO) algorithm to optimize both its task-exploration and task-solving policies. Therefore, we provide a short technical description of PPO in Appendix A.1.1. Then, we show in Appendix A.1.2 a detailed overview of how PPO can be used in meta-RL, by implementing the method proposed in Sec. 3.1.

### A.1.1    PPO Background

For a policy $\pi_\theta$ with parameters $\theta$, Schulman et al. (2017) propose the objective

$$\mathcal{L}_t^{\mathrm{PPO}}(\theta) = \mathbb{E}_t\left[\mathcal{L}_t^{\mathrm{CLIP}}(\theta) - c_1\mathcal{L}_t^{\mathrm{VF}}(\theta) + c_2 S[\pi_\theta](s_t, \boldsymbol{c})\right], \tag{12}$$

where $c_1, c_2$ are constants, $\mathcal{L}_t^{\mathrm{CLIP}}$ is the main PPO objective, and $\mathcal{L}_t^{\mathrm{VF}}$ and $S[\pi_\theta]$ are additional objectives. $\mathcal{L}_t^{\mathrm{VF}}$ is a squared-error loss on the value function $V_\theta(s_t, \boldsymbol{c})$, while $S[\pi_\theta]$ is the policy entropy. Note that the sole change from the original description of PPO is that $\pi_\theta(a_t \mid s_t, \boldsymbol{c})$ and $V_\theta(s_t, \boldsymbol{c})$ depend not only on the state $s_t$, but also on the task context $\boldsymbol{c}$. The clipped surrogate objective $\mathcal{L}_t^{\mathrm{CLIP}}$ is defined as

$$\mathcal{L}_t^{\mathrm{CLIP}} = \mathbb{E}_t\left[\min\left(r_t(\theta)\hat{A}_t, \mathrm{clip}(r_t(\theta), 1-\epsilon, 1+\epsilon)\hat{A}_t\right)\right], \tag{13}$$

for a constant $\epsilon$, policy probability ratio $r_t(\theta)$, and estimated advantage $\hat{A}_t$. The probability ratio $r_t(\theta) = \frac{\pi_\theta(a_t|s_t,\boldsymbol{c})}{\pi_{\theta_{\mathrm{old}}}(a_t|s_t,\boldsymbol{c})}$ is computed using the parameters $\theta_{\mathrm{old}}$ from before an update. Eq. 13 encourages small, stable policy updates that keep $\pi_\theta$ close to $\pi_{\theta_{\mathrm{old}}}$, by constraining $r_t(\theta)$ to remain within $[1-\epsilon, 1+\epsilon]$.

A popular choice for the advantage estimator $\hat{A}_t$ is the generalized advantage estimator (GAE) (Schulman et al., 2015). In the meta-RL setting, the GAE can be defined as $\hat{A}_t^{\mathrm{GAE}} = \sum_{l=0}^{H-t}(\gamma\lambda)^l\delta_{t+l}^V$ for an episode $\tau$ with horizon $H$. Here, $\gamma \in [0,1]$ is the MDP's discount factor, $\lambda \in [0,1]$ is an additional discount, and $\delta_t^V = r_t + \gamma V(s_{t+1}, \boldsymbol{c}) - V(s_t, \boldsymbol{c})$ is the temporal-difference (TD) error at timestep $t$ for a reward $r_t$.

Furthermore, as mentioned in Sec. 3.1, we stabilize PPO (Wang et al., 2020; Sun et al., 2022; 2023; Moalla et al., 2024) using proximal feature optimization (Moalla et al., 2024, PFO).

### A.1.2    PPO for Meta-RL

In Sec. 3.1, we propose a simple change to the policy optimization objective. By replacing the environment reward $r_t$ with our proposed meta-reward $r_t^+$, PPO can be used to optimize $\pi_\theta$ to solve meta-RL tasks. That is, we compute the TD error

$$\delta_t^V(\theta) = r_t + \beta\, w(s_t, \boldsymbol{c})\, S[\pi_\theta](s_t, \boldsymbol{c}) + \gamma V(s_{t+1}, \boldsymbol{c}) - V(s_t, \boldsymbol{c}), \tag{14}$$

where $w(s_t, \boldsymbol{c}) = \max(0, \tanh(V(s_t) - V(s_t, \boldsymbol{c}) - \zeta))$. The estimated advantages can then be computed as $\hat{A}_t^{\mathrm{GAE}}(\theta) = \sum_{l=0}^{H-t}(\gamma\lambda)^l\delta_{t+l}^V(\theta)$. Note that, because $\hat{A}_t^{\mathrm{GAE}}(\theta)$ is now a function of the parameters $\theta$, the advantages must be recomputed every time $\theta$ is updated, i.e., after each PPO minibatch update. This is in contrast to standard PPO, where $\hat{A}_t^{\mathrm{GAE}}$ is computed only once, before an update, and then kept fixed until new data is collected.

Intuitively, $w(s_t, \boldsymbol{c})$ measures how effective the policy $\pi_\theta$ is at using the context $\boldsymbol{c}$, for each timestep $t$. The assumption is that, while PPO-optimized policies in single-MDP settings might explore the state space sufficiently, there is no guarantee that the task context space is meta-explored enough in a meta-RL setting. By introducing $w(s_t, \boldsymbol{c})$, whenever $\boldsymbol{c}$ does not lead to an improvement above a threshold $\zeta$, the policy is urged to meta-explore from state $s_t$, thus learning more about $\boldsymbol{c}$.

| Notation | Description | Definition |
|---|---|---|
| **Meta-RL Framework** | | |
| $\mathcal{M}_i \sim p(\mathcal{M})$ | Task (MDP) from distribution $p(\mathcal{M})$ | |
| $s_t^k, a_t^k, r_t^k$ | Timestep $t$ of episode $k$ | $s_t^k \in \mathbb{R}^{d_s}, a_t^k \in \mathbb{R}^{d_a}, r_t^k \in \mathbb{R}$ |
| $d$ | Sum of dimensions in $(s_t^k, a_t^k, r_t^k)$ tuple | $d = d_s + d_a + 1$ |
| $H$ | Horizon | $H \in \mathbb{Z}^+$ |
| $H'$ | Total dimension of $H$ steps | $H' = Hd$ |
| $K$ | Few-shot adaptation budget | $K \in \mathbb{Z}^+$ |
| $\tau$ | Episode vector | $\tau \in \mathbb{R}^{H'}$ |
| $P_{\mathcal{M}_i}^{\pi}(\tau)$ | Episode distribution in $\mathcal{M}_i$ under $\pi$ | |
| $\mathcal{D}^{(K)}$ | Meta-episode matrix ($K$ episodes) | $\mathcal{D}^{(K)} \in \mathbb{R}^{H' \times K}$ |
| **B** | Tensor of $Q$ meta-episodes from $\mathcal{M}_i$ | $\mathbf{B} \in \mathbb{R}^{Q \times H' \times K}$ |
| $\mathbf{B}_j$ | $j$-th meta-episode in **B** | $\mathbf{B}_j \in \mathbb{R}^{H' \times K}$ |
| $\mathcal{B}$ | Batch of meta-episodes | $\mathcal{B} = \left\{ \mathbf{B} \sim P_{\mathcal{M}_i}^{\pi^{\mathrm{explore}}} \middle| \mathcal{M}_i \sim p(\mathcal{M}) \right\}$ |
| **Latent Representations** | | |
| $\boldsymbol{c}$ | Latent task context | $\boldsymbol{c} \in \mathbb{R}^{(HKd_{\mathrm{model}})}$ |
| $\boldsymbol{\Gamma}$ | Task exploration representation | $\boldsymbol{\Gamma} \in \mathbb{R}^{(Hd_{\mathrm{model}}) \times K}$ |
| **Z** | Task-specific representation | $\mathbf{Z} \in \mathbb{R}^{Q \times (HK) \times d_{\mathrm{model}}}$ |
| $\boldsymbol{z}$ | Global representation | $\boldsymbol{z} \in \mathbb{R}^{d_{\mathrm{model}}}$ |
| **Optimization** | | |
| $w(s_t, \boldsymbol{c})$ | Dynamic in-context weight | $w(s_t, \boldsymbol{c}) = \max(0, \tanh(V(s_t) - V(s_t, \boldsymbol{c}) - \zeta))$ |
| $r_t^+$ | Augmented meta-reward | $r_t^+ = r_t + \beta\, w(s_t, \boldsymbol{c})\, S[\pi](s_t, \boldsymbol{c})$ |
| $\boldsymbol{d}_k$ | Causal similarity between episode $k$ and previous $(k-1)$ episodes | $\boldsymbol{d}_k = \frac{1}{K} \sum_{k'=1}^{k-1} S_C(\boldsymbol{\Gamma}_{:,k}, \boldsymbol{\Gamma}_{:,k'})$ |
| $\tilde{r}^k$ | Exploration reward in $k$-th episode | $\tilde{r}^k = \exp\left(-\frac{1}{\sigma}\boldsymbol{d}_k^2\right)$ |
| **C** | Tensor of coefficients | $\mathbf{C} \in \mathbb{R}^{Q \times K \times K}$ |
| $\mathbf{B}_j\mathbf{C} \approx \mathbf{B}$ | Linear task reconstruction | |
| $\delta_j$ | Linear reconstruction target | $\delta_j = 1 - \exp\left(-\xi \overline{(\mathbf{B}_j\mathbf{C} - \mathbf{B})^2}\right)$ |
| $\mathcal{L}_{\mathrm{rec}}$ | Reconstruction loss | |
| $\mathcal{L}_{\mathrm{t\text{-}rec}}$ | Task-reconstruction loss | |
| $\mathcal{L}_{\mathrm{contr}}$ | Contrastive loss | |
| $\mathcal{R}$ | Regularizer | |
| $\mathcal{L}_{\mathrm{LaSER}}$ | Unified LaSER loss | |
| **Architecture** | | |
| $\pi_\theta$ | Task-solving policy | |
| $\pi_\phi^{\mathrm{explore}}$ | Task exploration policy | |
| $f_u$ | Online unidirectional encoder | |
| $f_b$ | Offline bidirectional encoder | |
| $g_\omega$ | Hybrid dual-encoder combining $f_u$ and $f_b$ | |
| $h_{\mathrm{rec}}$ | Self-supervised reconstruction head | |
| $h_{\mathrm{t\text{-}rec}}$ | Task-reconstruction head | |

Table 3: Summary of main notation and definitions.

As the agent improves at using $\boldsymbol{c}$ to maximize return, the exploration bonus $w(s_t, \boldsymbol{c})S[\pi_\theta](s_t, \boldsymbol{c})$ at state $s_t$ decreases. The standard PPO objective is only recovered when $V(s_t) \leq V(s_t, \boldsymbol{c}) + \zeta$. This signals that the agent has learned how to use $\boldsymbol{c}$ in state $s_t$, and no additional meta-exploration is required.

## A.2 Matrix Formulation for Task Exploration

This section provides the formal matrix operations used to compute the causal similarity scores discussed in Sec. 3.2. As previously established, the goal of task exploration is to collect a $K$-shot meta-episode $\mathcal{D}^{(K)}$ that avoids redundancy and captures the essential dynamics of a task $\mathcal{M}_i$. Ultimately, this data must be sufficient to construct a context vector $\boldsymbol{c}$ that enables the task policy $\pi$ to become optimal in $\mathcal{M}_i$.[8]

The latent representation generated by $f_u$ is a matrix $\boldsymbol{\Gamma} \in \mathbb{R}^{(Hd_{\mathrm{model}}) \times K}$, while the pairwise cosine similarity is a matrix $S_C(\boldsymbol{\Gamma}^{\mathsf{T}}, \boldsymbol{\Gamma}) \in \mathbb{R}^{K \times K}$. The average column-wise similarity in $\mathcal{D}^{(K)}$ is then a vector $\hat{\boldsymbol{d}} \in \mathbb{R}^{K}$ computed as $\hat{\boldsymbol{d}} = \frac{1}{K} S_C(\boldsymbol{\Gamma}^{\mathsf{T}}, \boldsymbol{\Gamma}) \mathbf{1}$, where $\mathbf{1}$ is the $K$-dimensional all-ones vector.

Consequently, $\hat{\boldsymbol{d}}_k$ represents the average similarity between the $k$-th episode and all other episodes. However, since $\pi^{\mathrm{explore}}$ must collect episodes sequentially during online adaptation, it cannot access the similarity scores between its current episode $k$ and future, uncollected episodes. To resolve this, we apply a causal mask $\boldsymbol{M} \in \mathbb{R}^{K \times K}$, defined as a strictly lower triangular matrix (i.e., $\boldsymbol{M}_{i,j} = 1$ if $j < i$, and 0 otherwise), by taking the Hadamard product with $S_C(\boldsymbol{\Gamma}^{\mathsf{T}}, \boldsymbol{\Gamma})$. This masked similarity matrix is then used to compute the causal similarity vector $\boldsymbol{d} = \frac{1}{K}(S_C(\boldsymbol{\Gamma}^{\mathsf{T}}, \boldsymbol{\Gamma}) \odot \boldsymbol{M}) \mathbf{1}$ required in Eq. 4.

## A.3 Formal Matricization of Linear Task Reconstruction

In Sec. 3.3, we introduced Assumption 1 (Linear Task Reconstruction) using batched tensor notation for clarity. Here, we reiterate this assumption in a more detailed and formal manner.

Recall that our motivation is to sidestep the challenge of having to directly find $K$ episodes that allow $K$-shot adaptation. Consider rearranging the tensor $\mathbf{B} \in \mathbb{R}^{Q \times H' \times K}$ of $Q$ meta-episodes into a matrix $\boldsymbol{B}_{[2]} \in \mathbb{R}^{H' \times (QK)}$ of $QK$ episodes via mode-2 matricization (Vasilescu, 2009). That is, $(\boldsymbol{B}_{[2]})_{:,k}$ represents the $k$-th episode, for $k \in [QK]$. Then, assuming that $\boldsymbol{B}_{[2]}$ is sufficient for $K$-shot adaptation, Definition 1 implies that $\boldsymbol{B}_{[2]}$ is composed of the $K$ episodes required for adaptation, together with the $(Q-1)K$ episodes containing redundant or irrelevant information. More precisely, there exists a meta-episode $\boldsymbol{B}_{[2]}^*$ composed of these $K$ episodes. We can now formally introduce the matrix form of our work's core assumption:

**Assumption 1** (Matrix Formulation of Linear Task Reconstruction)**.** *Let $\boldsymbol{B}_{[2]} \in \mathbb{R}^{H' \times (QK)}$ contain $QK$ episodes collected from an MDP $\mathcal{M}_i$ such that $\boldsymbol{B}_{[2]}$ is sufficient for $K$-shot adaptation. Let $\boldsymbol{B}_{[2]}^* \in \mathbb{R}^{H' \times K}$ be the submatrix of $\boldsymbol{B}_{[2]}$ that contains these $K$ necessary episodes. Then, we assume there exists a matrix $\boldsymbol{C}_{[2]} \in \mathbb{R}^{K \times (QK)}$ such that*

$$\boldsymbol{B}_{[2]}^* \boldsymbol{C}_{[2]} \approx \boldsymbol{B}_{[2]} \,. \tag{15}$$

It directly follows from our assumption that $\mathrm{rank}(\boldsymbol{B}_{[2]}) \approx K$. Consequently, the submatrix $\boldsymbol{B}_{[2]}^*$ is an optimal full-rank approximation of $\boldsymbol{B}_{[2]}$, containing the $K$ linearly independent columns in $\boldsymbol{B}_{[2]}$, while $\boldsymbol{C}_{[2]}$ is its coefficient matrix.

To make our framework practical to implement and simpler to describe, we can easily switch back to the tensor representations used in the main text. Specifically, we use the tensor $\mathbf{B}$ instead of $\boldsymbol{B}_{[2]}$, represent $\boldsymbol{B}_{[2]}^*$ as the meta-episode slice $\mathbf{B}_j \in \mathbb{R}^{H' \times K}$ for a given $j \in [Q]$, and rearrange the coefficients $\boldsymbol{C}_{[2]}$ into a tensor $\mathbf{C} \in \mathbb{R}^{Q \times K \times K}$. Eq. 15 is then equivalent to the original batch matrix multiplication formulation $\mathbf{B}_j \mathbf{C} \approx \mathbf{B}$.

## A.4 Extended Architecture and Optimization Details

Here, we extend the discussion in Sec. 3.4 on our architecture and optimization approach. The implementation details are then given in Appendix D.

---

[8]Here, we assume that $\pi$ has already been meta-trained and has enough prior knowledge of the task distribution $p(\mathcal{M})$ to use $\boldsymbol{c}$ effectively.

### A.4.1 Architecture

While encoders $f_u$ and $f_b$ share the Eq. 10 loss $\mathcal{L}_{\text{LaSER}}$, their specific architectures require different optimization approaches. The unidirectional transformer $f_u$ is trained using autoregressive attention masks (Vaswani et al., 2017; Radford et al., 2018). This masks future timesteps when computing the attention matrix, ensuring that the agent cannot "cheat" during training by using information that should not be available. This ultimately enables $f_u$ to operate in online settings, where encodings are computed timestep by timestep as data is collected.

In contrast, $f_b$ has no online constraints. Specifically, $f_b$ always operates on complete meta-episodes, where a masked bidirectional model may offer richer representation. We follow the standard approach of training bidirectional transformers through masked self-supervision, but defer the details of this process to Appendix A.4.3. While we simplified notation in the main text when discussing the computation of the global and task-specific representations $(\boldsymbol{z}, \mathbf{Z}) = f_b(\mathbf{B}; \omega_b)$, we note that in practice $f_b$ actually takes a batch

$$\mathcal{B} = \left\{ \mathbf{B} \sim P_{\mathcal{M}_i}^{\pi_\phi^{\text{explore}}} \middle| \mathcal{M}_i \sim p(\mathcal{M}) \right\} \tag{16}$$

as input, where each $\mathbf{B}^{(i)} \in \mathbb{R}^{Q \times H' \times K}$ is a collection of $Q$ meta-episodes collected by $\pi_\phi^{\text{explore}}$ from a unique task $\mathcal{M}_i$ (as shown in Alg. 1). This accelerates training in a practical implementation. More importantly, it allows $\boldsymbol{z} \in \mathbb{R}^{d_{\text{model}}}$ to capture the shared structure across multiple elements of $p(\mathcal{M})$ by encoding the full batch. Simultaneously, we compute a set $\{\mathbf{Z}^{(i)}\}_{i=1}^{|\mathcal{B}|}$ of task-specific representations, where each $\mathbf{Z}^{(i)} \in \mathbb{R}^{Q \times (HK) \times d_{\text{model}}}$ is independently encoded from $\mathbf{B}^{(i)}$ to capture the properties of $\mathcal{M}_i$.

### A.4.2 Optimization

The dual encoder $g$ employs two pre-training heads during meta-training, with each having different roles for the optimization of the LaSER model. The task reconstruction head $h_{\text{t-rec}}$ is optimized to minimize loss $\mathcal{L}_{\text{t-rec}}$ (Eq. 6), learning to compute the coefficient tensor $\mathbf{C} = h_{\text{t-rec}}(g_\omega(\mathbf{B}), g_\omega(\mathbf{B}_j); \omega_{\text{t-rec}})$. That is, $h_{\text{t-rec}}$ combines $g_\omega(\mathbf{B})$ (the representations of $Q$ meta-episodes) with $g_\omega(\mathbf{B}_j)$ (the representation of the $j$-th meta-episode). To increase stability, the tensor $\mathbf{C}$ used to compute $\delta_j$ in Eq. 7 is encoded by target parameters $\hat{\omega}_{\text{t-rec}}$, which are only updated to $\omega_{\text{t-rec}}$ every $\nu$ iterations (Mnih et al., 2015).

We note, however, that we do not backpropagate $\mathcal{L}_{\text{t-rec}}$ through the parameters $\omega$ of $g$. The intuition is that the role of $g$ is to create a contrastive-optimized latent space by following loss $\mathcal{L}_{\text{contr}}$ (Eq. 8), not to learn to compute $\mathbf{C}$, which is infeasible in practice. The sole role of $h_{\text{t-rec}}$ is then to project the output of $g$ into a collection of coefficients that satisfy the linear equation in Assumption 1.

Conversely, the role of the reconstruction head $h_{\text{rec}}$ is to optimize parameters $\omega$ by learning to reconstruct corrupted inputs. Specifically, $h_{\text{rec}}$ outputs an attempt at reconstructing the masked input given to $g$, necessary for computing the loss $\mathcal{L}_{\text{rec}}$.

The final piece left to discuss in our optimization process is the regularization term $\mathcal{R}$ in the composite loss $\mathcal{L}_{\text{LaSER}}$. We take inspiration from Piratla et al. (2020) and use $\mathcal{R}$ to regularize the latent spaces generated by $f_b$ and $f_u$. Specifically, we enforce a soft orthogonality constraint between the vector representations $\boldsymbol{z}$ and $\mathbf{Z}_{j,t,:}$, for each $j \in [Q]$ and timestep $t \in [HK]$, by minimizing $(\boldsymbol{z}^\mathsf{T} \mathbf{Z}_{j,t,:})^2$. As discussed in the main text, this aims to encourage $\mathbf{Z}$ to avoid capturing the global information already contained in $\boldsymbol{z}$. Finally, $\mathcal{R}$ also normalizes the vector representations $\boldsymbol{z}$, $\mathbf{Z}_{j,t,:}$, and $\mathbf{\Gamma}_{:,k}$, for all $j \in [Q], t \in [HK], k \in [K]$.

### A.4.3 Masked Self-Supervised Training

The bidirectional transformer encoder $f_b$, introduced in Sec. 3, learns useful data representations by learning to reconstruct its input. Since reconstructing a meta-episode $\mathbf{B}_j$ is trivial when the entire $\mathbf{B}_j$ is given as input, we use masked self-supervised training. We create a masked meta-episode $\mathbf{B}_j^{\text{masked}}$ by applying a stochastic masking function, similarly to Devlin et al. (2019) and Lewis et al. (2019). More precisely, $\mathbf{B}_j^{\text{masked}}$ is identical to $\mathbf{B}_j$, with the exception that each timestep $(s, a, r)$ in $\mathbf{B}_j^{\text{masked}}$ has a 15% chance of being corrupted. The loss $\mathcal{L}_{\text{rec}}$ measures the ability of $f_b$ to predict the true value of each corrupted timestep. The encoder must

learn to compute these values from the rest, non-corrupted timesteps in the input. We consider two types of corruption.

- **Masking**: with a probability of 80%, the selected timestep is replaced by a special $\langle MASK \rangle$ timestep, which carries no information.

- **Replacing**: with a probability of 10%, the selected timestep is replaced by another timestep from the same meta-episode.

The rest (i.e., 10%) of the selected timesteps are not corrupted but are still used when computing $\mathcal{L}_{\text{rec}}$. To optimize this loss, $f_b$ must learn general temporal relationships between the timesteps in a meta-episode. We compute $\mathcal{L}_{\text{rec}}$ as the MSE between the unmasked tokens in $\mathbf{B}_j$ and the reconstructed meta-episode $\mathbf{B}'_j = h_{\text{rec}}(\boldsymbol{c}; \omega_{\text{rec}})$, where $\boldsymbol{c} = g_\omega(\mathbf{B}_j^{\text{masked}})$. Note that masked meta-episodes are only used to train the encoder. When training the task policy $\pi_\theta$ or during meta-testing, we compute $\boldsymbol{c}$ using unmasked meta-episodes.

# B    Algorithms

For the sake of completeness, but also to enable reproducibility and further analysis (Phuong & Hutter, 2022), we provide pseudocode for our proposed algorithm LaSER. Alg. 1 shows our meta-training process. It is composed of three main parts. Each part optimizes one of the three LaSER components: the encoder $g_\omega$ (Alg. 3), the exploration policy $\pi_\phi^{\text{explore}}$ (Alg. 4), and the task policy $\pi_\theta$ (Alg. 5). Finally, we provide details on how we meta-test a fully trained LaSER agent in Alg. 2.

# C    Non-Linear Task Reconstruction

In this section, we show a straightforward method of extending the linear task reconstruction formulation from Sec. 3.3 to a non-linear variant. Specifically, we relax the linear Assumption 1, which assumes the existence of a coefficient tensor $\mathbf{C}$ such that $\mathbf{B}_j \mathbf{C} \approx \mathbf{B}$.

For the non-linear extension, we replace the linear mapping with a non-linear function $\Psi_\psi$, modeled as an MLP parameterized by $\psi$. To obtain these parameters, we repurpose the task reconstruction head $h_{\text{t-rec}}$ (defined in Sec. 3.4) to output the vector $\psi$ instead of a tensor $\mathbf{C}$. This effectively turns $h_{\text{t-rec}}$ into a hypernetwork that computes the weights and biases $\psi = h_{\text{t-rec}}(g_\omega(\mathbf{B}), g_\omega(\mathbf{B}_j); \omega_{\text{t-rec}})$. Assumption 1 can then be replaced with the non-linear variant $\Psi_\psi(\mathbf{B}_j) \approx \mathbf{B}$.

The task-reconstruction loss (Eq. 6) is adapted to minimize the error of this non-linear reconstruction:

$$\mathcal{L}_{\text{t-rec}} = \mathbb{E}_{\mathcal{M}_i \sim p(\mathcal{M}), \, \mathbf{B} \sim P_{\mathcal{M}_i}^{\pi^{\text{explore}}}, \, j \sim U(Q)} \left[ \text{MSE}\left( \Psi_\psi(\mathbf{B}_j), \mathbf{B} \right) \right]. \tag{17}$$

Similarly, the target $\delta_j$ (Eq. 7) used to shape the latent exploration space becomes:

$$\delta_j = 1 - \exp\left( -\xi \overline{(\Psi_\psi(\mathbf{B}_j) - \mathbf{B})^2} \right). \tag{18}$$

Apart from these changes, the LaSER variant with non-linear task reconstruction follows the same formulation as linear LaSER (see Sec. 3).

In practice, we take the same approach as in the linear variant and process a batch $\mathbf{B}$ of $Q$ meta-episodes by independently generating a non-linear transformation for each meta-episode. Specifically, $h_{\text{t-rec}}$ outputs a set of $Q$ distinct parameter vectors $\psi_q$, one for each target meta-episode $\mathbf{B}_q \in \mathbf{B}$, for $q \in [Q]$. An element-wise concatenation of these $Q$ reconstructions would then give the desired approximation: $\{\text{MLP}_{\psi_q}(\mathbf{B}_j)\}_{q=1}^Q \approx \mathbf{B}$.

# D    Implementation Details

We provide additional details on our implementation of the LaSER algorithm and our architecture. This section is complemented by the hyperparameters given in Appendix H.

---

**Algorithm 3** train_encoder()

---

**Input** $p(\mathcal{M})$, task distribution;
   $\pi_\phi^{\text{explore}}$, exploration policy; $g_\omega$, encoder;
   $\omega_{\text{rec}}, \omega_{\text{t-rec}}$, head parameters
**Output** $\omega, \omega_{\text{rec}}, \omega_{\text{t-rec}}$, updated parameters

1: $\mathcal{B} \leftarrow \left\{ \mathbf{B} \sim P_{\mathcal{M}_i}^{\pi_\phi^{\text{explore}}} \middle| \mathcal{M}_i \sim p(\mathcal{M}) \right\}$   $\triangleright$ Collect an ordered batch of data from multiple tasks

2: $\mathcal{B}^{\text{masked}} \leftarrow \{\text{mask}(\mathbf{B}) | \mathbf{B} \in \mathcal{B}\}$

3: **for** $u \in [N_{\text{encoder}}]$ **do**

4:    $\left( z, \left\{ \mathbf{Z}^{(1)}, \ldots, \mathbf{Z}^{(|\mathcal{B}|)} \right\} \right) \leftarrow f_b(\mathcal{B}^{\text{masked}}; \omega_b)$   $\triangleright$ Get global and task-specific representations

5:    $\mathcal{L}_{\text{LaSER}} \leftarrow 0$

6:    **for** $i \in \left[ |\mathcal{B}| \right]$ **do**

7:       $\mathbf{B} \leftarrow \mathcal{B}_i$;       $\mathbf{B}^{\text{masked}} \leftarrow \mathcal{B}_i^{\text{masked}}$

8:       **for** $j \in [Q]$ **do**

9:          $\mathbf{\Gamma} \leftarrow f_u(\mathbf{B}_j^{\text{masked}}; \omega_u)$

10:          $c \leftarrow h_{\omega_h}(z, \mathbf{Z}_j^{(i)}, \mathbf{\Gamma})$   $\triangleright$ Eq. 9

11:          $\mathbf{B}'_j \leftarrow h_{\text{rec}}(c; \omega_{\text{rec}})$   $\triangleright$ Compute reconstructed input

12:       **end for**

13:       $j \sim U(Q)$

14:       $\mathbf{C} \leftarrow h_{\text{t-rec}}(g_\omega(\mathbf{B}), g_\omega(\mathbf{B}_j); \omega_{\text{t-rec}})$

15:       $\delta_j \leftarrow 1 - \exp\left( -\xi \overline{(\mathbf{B}_j\mathbf{C} - \mathbf{B})^2} \right)$   $\triangleright$ Eq. 7

16:       $\omega \leftarrow \omega_u \oplus \omega_b \oplus \omega_h$

17:       $\mathcal{L}_{\text{LaSER}}^{(i)}(\omega, \omega_{\text{rec}}, \omega_{\text{t-rec}}) \leftarrow$ compute using $\mathbf{B}$, $\mathbf{B}'$, $\mathbf{C}$, and $\delta_j$   $\triangleright$ Eq. 6, 8, 10

18:       $\mathcal{L}_{\text{LaSER}} \leftarrow \mathcal{L}_{\text{LaSER}} + \frac{1}{|\mathcal{B}|}\mathcal{L}_{\text{LaSER}}^{(i)}(\omega, \omega_{\text{rec}}, \omega_{\text{t-rec}})$

19:    **end for**

20:    $\omega, \omega_{\text{rec}}, \omega_{\text{t-rec}} \leftarrow$ update using $\nabla\mathcal{L}_{\text{LaSER}}$

21: **end for**

22: **return** $\omega, \omega_{\text{rec}}, \omega_{\text{t-rec}}$

---

---

**Algorithm 4** train_exploration_policy()

---

**Input** $p(\mathcal{M})$, task distribution; $\pi_\phi^{\text{explore}}$, exploration policy; $g_\omega$, encoder

**Output** $\phi$, updated parameters

1: $\mathcal{D} \leftarrow []$
2: **for** $\mathcal{M}_i \sim p(\mathcal{M})$ **do**
3:      $\mathcal{D}^{(K)} \leftarrow []$
4:      **for** $k \in [K]$ **do**
5:          **for** $t \in [H]$ **do**
6:              $a_t^k \sim \pi_\phi^{\text{explore}}\big(\cdot \mid s_t^k, \mathbf{\Gamma}^{(:k,:t-1)}\big)$
7:              $\mathcal{D}^{(k,t)} \leftarrow (s_t^k, a_t^k, r_t^k)$
8:              $\mathbf{\Gamma}^{(:k,:t)} \leftarrow f_u\big(\mathcal{D}^{(:k,:t)}; \omega_u\big)$
9:              Collect $s_{t+1}^k, r_{t+1}^k$ by taking action $a_t^k$ in $\mathcal{M}_i$
10:          **end for**
11:      **end for**
12:      $\mathbf{\Gamma} \leftarrow f_u(\mathcal{D}^{(K)}; \omega_u)$;        $\boldsymbol{d} \leftarrow \text{mask}\big(\frac{1}{K} S_C(\mathbf{\Gamma}^{\mathsf{T}}, \mathbf{\Gamma})\mathbf{1}\big)$
13:      $\tilde{r}^k \leftarrow$ compute for each $k \in [K]$                                     ▷ Eq. 4
14:      Replace environment rewards in $\mathcal{D}^{(K)}$ with $\tilde{r}^k$
15:      $\mathcal{D} \leftarrow [\mathcal{D}, \mathcal{D}^{(K)}]$
16: **end for**
17: **for** $u \in [N_{\text{PPO}}]$ **do**
18:      Optimize $\phi$ on transitions from $\mathcal{D}$ using PPO
19: **end for**
20: **return** $\phi$

---

**Algorithm 5** train_task_policy()

---

**Input** $p(\mathcal{M})$, task distribution;
         $\pi_\theta$, task policy; $\pi_\phi^{\text{explore}}$, exploration policy; $g_\omega$, encoder;
         $\hat{\boldsymbol{z}}$, shared component

**Output** $\theta$, updated parameters

1: $\mathcal{B} \leftarrow []$
2: **for** $\mathcal{M}_i \sim p(\mathcal{M})$ **do**
3:      $\mathcal{D}^{(K)} \sim P_{\mathcal{M}_i}^{\pi_\phi^{\text{explore}}}$                                    ▷ Sample exploration meta-episode
4:      $\boldsymbol{c} \leftarrow g_\omega(\mathcal{D}^{(K)}, \hat{\boldsymbol{z}})$
5:      $\mathcal{B} \leftarrow [\mathcal{B}, \tau \sim P_{\mathcal{M}_i}^{\pi_\theta(a|s,\boldsymbol{c})}]$                   ▷ Sample exploitation episode
6: **end for**
7: **for** $u \in [N_{\text{PPO}}]$ **do**
8:      $\hat{A}_\theta \leftarrow []$
9:      **for** $\tau \in \mathcal{B}$ **do**
10:          **for** $t \in [H]$ **do**
11:              Compute $V(s_t, \boldsymbol{c})$, $V(s_t)$, and $S[\pi_\theta](s_t, \boldsymbol{c})$
12:              $w(s_t, \boldsymbol{c}) \leftarrow \max\big(0, \tanh\big(V(s_t) - V(s_t, \boldsymbol{c}) - \zeta\big)\big)$     ▷ Eq. 3
13:              $r_t^+ \leftarrow r_t + \beta w(s_t, \boldsymbol{c}) S[\pi_\theta](s_t, \boldsymbol{c})$                    ▷ Eq. 2
14:              $\hat{A}_t^{\text{GAE}}(\theta) \leftarrow$ compute using $r_t^+$
15:              $\hat{A}_\theta \leftarrow \big[\hat{A}_\theta, \hat{A}_t^{\text{GAE}}(\theta)\big]$
16:          **end for**
17:      **end for**
18:      Optimize $\theta$ on transitions from $\mathcal{B}$ with advantages $\hat{A}_\theta$ using PPO
19: **end for**
20: **return** $\theta$

---

### D.1 Encoder

Both $f_u$ and $f_b$ use the same transformer architecture (Vaswani et al., 2017), except that $f_u$ is unidirectional, while $f_b$ is bidirectional. Each transformer has size $d_{\mathrm{model}} = 128$, 8 layers, 16 attention heads, and feed-forward networks of size 512, with `GELU` activation functions (Hendrycks & Gimpel, 2016). All input timesteps are linearly mapped to $d_{\mathrm{model}}$-dimensional embeddings, with positional encodings added before they are passed through the transformer. For stable training, we apply the T-Fixup initialization scheme (Huang et al., 2020), as recommended by Melo (2022).

The output of the transformer in $f_u$ is passed through a single-layer feed-forward network that outputs $\boldsymbol{\Gamma}$. Similarly, the transformer output of $f_b$ is independently processed by two separate single-layer feed-forward networks, outputting $\boldsymbol{z}$ and $\mathbf{Z}$. Each of these three networks has size 64 and uses `tanh` activation functions.

When using data from multiple tasks, $\boldsymbol{\Gamma}$ and $\mathbf{Z}$ are computed separately for each task. In contrast, $\boldsymbol{z}$ is computed with data from multiple tasks, leading to a general representation of the entire distribution.

Before computing $\boldsymbol{z}$, we apply average pooling to reduce the dimensionality of the output of $f_b$. We treat $\mathbf{Z}$ as a batch of $Q$ meta-episode representations, such that each slice $\mathbf{Z}_j$, for $j \in [Q]$, is computed independently by the feed-forward network. Similarly, each of the $Q$ meta-episodes has its own representation $\boldsymbol{\Gamma}$, computed independently by the feed-forward network in $f_u$.

The encoder $g_\omega$ has two final transformer layers, $h_{\omega_h}$, similar to those in $f_b$. The first layer combines $\mathbf{Z}$ and $\boldsymbol{\Gamma}$. The second computes $\boldsymbol{c}$ by combining the output of the first with $\boldsymbol{z}$. The reconstruction head $h_{\mathrm{rec}}$ is a single linear layer that transforms each of the $HK$ vectors in $\boldsymbol{c}$ from $d_{\mathrm{model}}$ to $d$, thus computing the reconstructed input. The task-reconstruction head $h_{\mathrm{t\text{-}rec}}$ is a $(256, 128, 128)$ feed-forward network with `GELU` activations, used for computing the tensor of coefficients $\mathbf{C}$. We have found, empirically, that using only the task-specific representations $\mathbf{Z}$ as input for $h_{\mathrm{t\text{-}rec}}$ is enough. That is, for a given $j \in [Q]$, we compute

$$\mathbf{C} = h_{\mathrm{t\text{-}rec}}(\mathbf{Z}, \mathbf{Z}_j; \omega_{\mathrm{t\text{-}rec}}), \tag{19}$$

$$\text{where } (\cdot, \mathbf{Z}) = f_b(\mathbf{B}; \omega_b), \tag{20}$$

and optimize $\omega_{\mathrm{t\text{-}rec}}$ such that $\mathbf{C}$ approximates the linear transformation that maps $\mathbf{B}_j$ to $\mathbf{B}$. To train $g_\omega$, $h_{\mathrm{rec}}$, and $h_{\mathrm{t\text{-}rec}}$, we use the Adam optimizer (Kingma & Ba, 2015).

### D.2 Policies

We represent $\pi_\phi^{\mathrm{explore}}$, $V_\phi$, $\pi_\theta$, and $V_\theta$ using feed-forward networks. For task exploration, we use two separate $(128, 128, 128)$ networks with `tanh` activations to represent policy $\pi_\phi^{\mathrm{explore}}$ and value function $V_\phi$. These take 64-dimensional embeddings of $s_t$ and $\boldsymbol{\Gamma}^{(:k,:t)}$ as input, which are computed by a shared, single-layer, `tanh`-activated network.

We use similar architectures for $\pi_\theta$ and $V_\theta$. However, $\boldsymbol{c}$ is first mapped to a lower-dimensional representation by a feed-forward context encoder with `tanh` activations. The same encoder is used for both $\pi_\theta$ and $V_\theta$. The hyperparameters of $\pi_\theta$ and $V_\theta$ are given in Table 9.

To optimize both policies, each PPO update is run for $N_{\mathrm{PPO}}$ epochs, using minibatches. The exploration policy is always meta-trained using $N_{\mathrm{PPO}} = 4$ with 2 minibatches. For the task policy, these two hyperparameters are environment-dependent and shown in Table 9. Moreover, we normalize the state $s$, the reward $r$, and the context $\boldsymbol{c}$ before passing them to the task policy. We use the Adam optimizer, and clip the norm of the gradients to 0.5 before updating parameters $\phi$ and $\theta$. Additionally, we set the coefficients in Eq. 12 to $c_1 = 0.5$ and $c_2 = 0.01$. The GAE $\hat{A}_t^{\mathrm{GAE}}$ is computed using discounts $\gamma = 0.99$ and $\lambda = 0.9$.

Finally, as mentioned in Sec. 3.1, we stabilize the optimization of $\pi_\theta$ by adding the PFO term suggested by Moalla et al. (2024) to the PPO objective, weighted by $c_{\mathrm{PFO}}$, which is set based on the environment. The PFO term is omitted when optimizing $\pi_\phi^{\mathrm{explore}}$. We found that including it made training more difficult, whereas $\pi_\phi^{\mathrm{explore}}$ was already sufficiently stable without it.

### D.3 Meta-Training

LaSER agents are meta-trained in two phases. In the first phase, we alternate between updating the encoder $g_\omega$ and the exploration policy $\pi_\phi^{\text{explore}}$ for $N_{\text{explore}}$ iterations. Before each encoder update, we collect a dataset $\mathcal{B}$ by following $\pi_\phi^{\text{explore}}$ in different tasks. Each element $\mathbf{B} \in \mathcal{B}$ is a tensor that contains $Q$ meta-episodes from a task $\mathcal{M}_i \sim p(\mathcal{M})$.

We update $g_\omega$ on the masked dataset $\mathcal{B}^{\text{masked}}$ for $N_{\text{encoder}}$ epochs (see Alg. 3). To improve meta-training sample efficiency, we use all $Q$ meta-episodes in each tensor $\mathbf{B}$ to compute $\mathcal{L}_{\text{rec}}$ and $\mathcal{R}$. In practice, each $\mathcal{B}$ is stored in a buffer and reused in future updates. To update $\pi_\phi^{\text{explore}}$, exploration data is collected from multiple tasks as described in Sec. 3.2. In the second phase, the task policy $\pi_\theta$ is updated for $N_{\text{task}}$ iterations.

For the losses $\mathcal{L}_{\text{rec}}$ and $\mathcal{L}_{\text{t-rec}}$ in Eq. 10, we exclude the reconstructed actions from the loss computation. This ensures that the encoder focuses on learning patterns in states and rewards, rather than the exploration policy. Additionally, we scale the loss on states by a factor of 0.05.

Note that the encoder $g_\omega$ is optimized purely through self-supervised methods. This constraint can be lifted. The RL loss used to update the task policy $\pi_\theta$ can also be used to fine-tune $g_\omega$. This fine-tuning could adapt the general pretrained encoder to align more closely with the task-solving objective. However, similar to Zintgraf et al. (2021b), we observe no empirical benefits from fine-tuning, so we omit it in our experiments.

A closely related idea is to continue improving the context vector $\boldsymbol{c}$ by encoding data from the exploitation episode as it is being collected by the task policy $\pi_\theta$. This effectively turns our setting into "$(K + 1)$-shot adaptation", where the final episode comes from a different source. We have considered this alternative variant, but observed no significant changes in terms of empirical performance. However, a more interleaved interaction between the exploration and task policies may lead to more effective data collection.

In practice, LaSER's encoder $g_\omega$ and exploration policy $\pi_\phi^{\text{explore}}$ are meta-trained using an Nvidia A100 80GB GPU. In the second phase, we switch to an Nvidia RTX 4070 Ti SUPER 16GB GPU to meta-train the task policy $\pi_\theta$. All environment data is collected using CPU computations only.

## E Benchmarks Details

This section provides additional implementation details for the meta-RL benchmarks used in this work.

### E.1 MEWA

We continue the discussion from Sec. 5.1.1 and provide more details on the tasks available in the MEWA benchmark. In our implementation, we use the publicly available official code for MEWA, at https://github.com/RStoican/MEWA.

In a critical state $s_x$ of type $x$, a risky action $a_{\text{risk}}$ can either lead to a state $s'$ or $s'_x$. We use $s'$ to denote that a mistake has been avoided and the agent has progressed in the task. In contrast, $s'_x$ denotes that a mistake has happened, resulting in a large delay in task completion and thus, lower returns. The probability of $a_{\text{risk}}$ leading to a mistake depends on both the mistake's type $x$ and the dynamics of task $\mathcal{M}_i \sim p(\mathcal{M})$. Formally, we define $T_i(s'_x \mid s_x, a_{\text{risk}}, \mathcal{M}_i) = \boldsymbol{p}_x^{(i)}$ to be the task-specific transition leading to a mistake. Here, the vector $\boldsymbol{p}^{(i)}$, with $\boldsymbol{p}_y^{(i)} \leq \boldsymbol{p}_x^{(i)}$ for all $y < x$, denotes the probabilities of making a mistake of each type. The probability of avoiding a mistake is then simply $T_i(s' \mid s_x, a_{\text{risk}}, \mathcal{M}_i) = 1 - \boldsymbol{p}_x^{(i)}$.

An agent's second option is to take a safe action $a_{\text{safe}}$ in a critical state $s_x$. This comes at the cost of a small delay, and leads to a critical state $s_{x-1}$, with $\boldsymbol{p}_{x-1}^{(i)} \leq \boldsymbol{p}_x^{(i)}$. Formally, the dynamics are given by the task-agnostic transition $T_i(s_{x-1} \mid s_x, a_{\text{safe}}, \mathcal{M}_i) = 1$.

Table 4 provides the 12 meta-testing tasks used for the results in Sec. 5. We describe each task by its corresponding vector of mistake probabilities $\boldsymbol{p}^{(i)} \in [0, 1]^4$. Additionally, for each task $\mathcal{M}_i$, we provide the average probability of a mistake of any type happening, computed as $1/4 \sum_{x=1}^{4} \boldsymbol{p}_x^{(i)}$. As previously

mentioned, we use this as a rough measure of similarity between tasks. Finally, we mark tasks that are outside of MEWA's meta-training distribution (i.e., cannot be sampled during meta-training) as OOD tasks.

| Task Index | Mistake Probability | | | | Average Mistake Probability | OOD |
|---|---|---|---|---|---|---|
| | Type I | Type II | Type III | Type IV | | |
| 0 | 0 | 0 | 0 | 0 | 0.000 | No |
| 1 | 0.05 | 0 | 0 | 0 | 0.013 | No |
| 2 | 0.1 | 0 | 0 | 0 | 0.025 | No |
| 3 | 0.772 | 0 | 0 | 0 | 0.193 | No |
| 4 | 0.672 | 0.618 | 0 | 0 | 0.322 | No |
| 5 | 0.622 | 0.618 | 0.577 | 0.434 | 0.563 | No |
| 6 | 0.622 | 0.618 | 0.577 | 0.534 | 0.588 | Yes |
| 7 | 0.872 | 0.818 | 0.777 | 0 | 0.617 | Yes |
| 8 | 0.672 | 0.668 | 0.627 | 0.584 | 0.638 | Yes |
| 9 | 0.972 | 0.968 | 0.927 | 0.384 | 0.813 | Yes |
| 10 | 0.972 | 0.968 | 0.927 | 0.784 | 0.913 | Yes |
| 11 | 0.972 | 0.968 | 0.927 | 0.884 | 0.938 | Yes |

Table 4: The configuration of the 12 MEWA tasks used for meta-testing agents. A task is described by a 4-dimensional vector. Each dimension denotes the probability of a mistake of the corresponding type happening. We additionally compute the average mistake probability across all types. This can be seen as a way of comparing tasks, i.e., tasks with similar average probabilities have similar optimal policies. Finally, tasks marked as OOD are outside MEWA's meta-training task distribution.

### E.2 Meta-World

Meta-World was proposed by Yu et al. (2020). Its latest version, which was introduced by McLean et al. (2025), offers a unified and consistent framework for easily evaluating and comparing meta-RL algorithms. The most important change in this version is the option of selecting between Meta-World's initial set of task-specific reward functions (V1) and the newer set (V2). To be more consistent with the recent literature, we choose to follow V2 in our work. All Meta-World experiments are performed on the ML10 evaluation protocol. Ten tasks, with randomized positions for objects and goals, are selected for meta-training. Five structurally similar novel tasks are then used for meta-testing (for more details, see Yu et al. (2020)). To implement Meta-World, we use the official open codebase provided at https://github.com/Farama-Foundation/Metaworld.

### E.3 HopperMass

Our implementation of MuJoCo HopperMass is similar to the one provided at https://github.com/MohammadrezaNakhaei/ER-TRL by Nakhaeinezhadfard et al. (2025). The main difference from other implementations used in the literature is the focus on generating OOD tasks during meta-testing. The hopper model is characterized by four vectors: body mass, body inertia, friction, and degree of freedom damping. When generating a new task, each element from each of these vectors is scaled by a different weight. Weights are sampled from $[1.5^{-1}, 1.5^1]$ for meta-training tasks and from $[1.5^{-1.5}, 1.5^{-1}) \cup (1.5^1, 1.5^{1.5}]$ for OOD meta-testing tasks.

## F   Baseline Algorithms Details

We provide details for the architectures, hyperparameters, and the meta-training process for the baseline algorithms used in Sec. 5. We tune these algorithms and use the best-performing hyperparameters. Where possible, and if performance is not negatively affected, the task policy's architecture and meta-training are similar to LaSER's. Otherwise, the task policy is tuned with the rest of the model. All baseline algorithms are meta-trained using an Nvidia RTX 4070 Ti SUPER 16GB GPU.

### F.1 MAML

For MAML (Finn et al., 2017), we use the publicly available code provided for meta-RL by Deleu (2018) at https://github.com/tristandeleu/pytorch-maml-rl. We train for 12000 meta-iterations, with batches of 16 tasks. For each task, we sample 5 episodes. MAML is meta-trained to maximize returns after one policy gradient update. However, during meta-testing, it performs $K = 4$ gradient updates. We use a discount factor of $\gamma = 0.99$. We also use $\lambda = 0.9$ to compute the generalized advantage estimator (GAE). To ensure optimal results, we did not use the first-order approximation proposed by Finn et al. (2017), but instead computed the second derivatives and backpropagated. The policy is a $(64, 64)$ feed-forward network with `tanh` activations. Due to the computational cost of MAML, we had to use a smaller policy network than in the other algorithms. All other hyperparameters follow the ones used by Finn et al. (2017) in their RL experiments.

### F.2 PEARL

We use the publicly available code at https://github.com/katerakelly/oyster for our implementation of PEARL (Rakelly et al., 2019). Each agent is meta-trained for 350 iterations on a total of 3000 training tasks, with 16 tasks per batch. At each iteration, both the encoder and the task policy are optimized for 2000 gradient steps, with batches of 64 transitions for the encoder and 256 for the policy.

PEARL uses a variational approach for computing task contexts $c$. When computing PEARL's loss, the KL divergence of the encoder is weighted by 0.1. Additionally, both the task policy and the encoder have a learning rate of $3 \times 10^{-4}$ and use the Adam optimizer.

PEARL uses Soft-Actor Critic (SAC) (Haarnoja et al., 2018a;b), an off-policy RL algorithm. Note that, since SAC is designed for continuous action spaces, we instead use a SAC version for discrete action spaces (Christodoulou, 2019). We tune SAC's temperature hyperparameter automatically, using the approach introduced by Haarnoja et al. (2018b), since manual tuning can be difficult. We use a discount factor of $\gamma = 0.99$ and a target smoothing coefficient for SAC of 0.005. At each iteration, PEARL adds data collected from 5 randomly selected tasks to a replay buffer of size 1000000 timesteps. For each of these tasks, it collects 400 timesteps by conditioning its policy on a context $c$ sampled from a prior distribution over tasks. It additionally collects 600 more timesteps using a $c$ sampled from its meta-trained posterior over tasks. However, this latter data is only used to update the task policy, not the encoder. Before training starts, the replay buffer is populated with 2000 timesteps per task, collected by following a uniformly random policy.

PEARL's encoder is a $(200, 200, 200)$ feed-forward network with `ReLU` activations that computes 5-dimensional latent task context vectors $c$. The task policy is a $(128, 128, 128)$ feed-forward network with `tanh` activations.

### F.3 VariBAD

We use the code at https://github.com/lmzintgraf/varibad for VariBAD (Zintgraf et al., 2021b). We meta-train for 2350 (MEWA), 18750 (Meta-World), or 3125 (HopperMass) iterations. In MEWA, we meta-train on 3000 tasks. For any benchmark, at each iteration, we collect 200 timesteps per task from 16 different tasks. These timesteps are stored in a buffer of 10000 episodes. Additionally, before training, a uniformly random policy adds 5000 timesteps to the buffer. This buffer is later used to update the encoder.

VariBAD uses a variational auto-encoder (Kingma & Welling, 2014, VAE) to encode data collected from tasks into a latent context $c$. In MEWA, this encoder is optimized using Adam with a learning rate of 0.001. At each iteration, the encoder is updated for 3 steps, using batches of 15 episodes, sampled from the buffer. The encoder is a recurrent neural network of size 128 that computes 5-dimensional task contexts $c$. The decoder takes $c$ as input, together with the current transition $s, a, s'$, and reconstructs the reward $r$. We use a $(64, 32)$ feed-forward network to represent the encoder. The state, action, and reward inputs are preprocessed into representations of size 32, 16, and 16, respectively, by separate single-layer networks with `ReLU` activations. Both the encoder and decoder have such pre-processing layers.

Since VariBAD uses PPO to optimize its policy, we use the same network architectures and hyperparameters for MEWA as LaSER's task policy (see Appendix D.2 and Table 9). However, we do not use the additional PFO objective. Finally, before passing $s$ and $c$ to the policy, we embed them into 64-dimensional representations using separate single-layer networks with `tanh` activations. Similarly to LaSER, all experiments use the discount factor $\gamma = 0.99$.

For Meta-World and HopperMass, we use the hyperparameters proposed by Beck et al. (2023a) for ML10 as a starting point for our tuning. In Meta-World, we use 2 PPO epochs with 8 minibatches with a learning rate of $3 \cdot 10^{-4}$. In HopperMass, we use batches of 2 episodes to update the VAE. For the task policy, we have 4 PPO updates with 8 minibatches and a learning rate of $3 \cdot 10^{-4}$. The VAE for both Meta-World and HopperMass computes 10-dimensional latent context vectors $c$, which are used by a $(256, 256, 128)$ task policy. Moreover, we normalize the state $s$, the reward $r$, and the context $c$ before passing them to the task policy. The rest of the hyperparameters are the same as for the VariBAD models meta-trained on MEWA.

## F.4   DREAM

We use the public code provided by Liu et al. (2021) at https://github.com/ezliu/dream to implement DREAM. We meta-train for 100000 iterations. At each iteration, the model first collects $K = 4$ exploration episodes and updates the exploration policy. Afterwards, the task policy, conditioned on the exploration meta-episode, is used to collect an exploitation episode, and then updated.

The decoder used by DREAM takes a history of transitions of type $(s, a, r, s')$ as input. Each transition is individually embedded, using layers of size 64 for states, 16 for rewards, and 16 for actions. For each transition, we then concatenate these embeddings and apply a linear layer of size 64. Then, an LSTM with hidden dimension 128 encodes the history of embedded transitions. Finally, the output of this LSTM is postprocessed by a $(128, 64)$ feed-forward network with `ReLU` activations.

Both the task and exploration policies are trained with dueling double Q-learning (Wang et al., 2016b; Van Hasselt et al., 2016), parameterised as deep recurrent Q-networks. The deep Q-network (DQN) used to train the task policy takes the current state, history, and task context as inputs. The state and task context are separately embedded using matrices of size 64. The history is encoded using a process similar to the one used by the encoder. However, following Liu et al. (2021), the history is composed of timesteps instead of transitions (i.e., no next state $s'$), rewards are not embedded, actions use a 32-dimensional linear layer, and the hidden size of the LSTM is decreased to 64. A feed-forward network of size $(256, 64)$ with `ReLU` activations is then applied to the concatenation of these output embeddings. The task DQN has two final linear heads for computing the value and advantage functions. The exploration policy's DQN has a similar architecture.

Each of the two DQNs has its own replay buffer, each storing up to 16000 sequences of 50 timesteps. For each sequence, the timesteps are stored in consecutive order. We train both DQNs using the Adam optimizer with a learning rate of $1 \times 10^{-4}$. At each iteration, we update the DQNs 4 times using batches of size 32, clipping the norm of the gradients to 10. The DQNs are then synchronized after every 5000-th update. Finally, in all experiments, we use the discount factor $\gamma = 0.99$.

## F.5   First-Explore

Our implementation of First-Explore uses the publicly available code at https://github.com/btnorman/First-Explore provided by the authors (Norman & Clune, 2024). During meta-training, the agent is optimized for 93 iterations on tasks sampled from a pool of 3000 training tasks. At each iteration, First-Explore collects 80000 timesteps, then performs 50 updates. Each update contains a batch of 4 distinct tasks, with two meta-episodes (one from the exploration and one from the exploitation policy) formed of $K = 4$ episodes of length $H = 50$ per task, for a total of 1600 timesteps per update.

We follow the implementation of Norman & Clune (2024) and use a unidirectional causal transformer encoder, similar to the GPT-2 architecture (Radford et al., 2019). This acts as a Decision Transformer (Chen et al., 2021). To match the size of LaSER's meta-testing architecture, the encoder is a 12-layer transformer of size $d_{\text{model}} = 128$ with 16 heads. The feed-forward networks have a size of 3072, while the embeddings are

768-dimensional. The transformer uses `GELU` activations. This encoder then acts as both the explore and exploit policies, which share parameters everywhere except in the final linear layer.

This architecture is optimized using the authors' novel sequence-modeling-based approach. Similarly to them, the exploration and exploitation policies both have successor policy variants. The main policies are then updated to match their successor every $T = 10$ iterations. During meta-training, we set the environment-specific baseline reward hyperparameter to $b = -20$, which is lower than the minimum return of a MEWA episode of length $H = 50$. Finally, we use the Adam optimizer with a linear learning schedule. The learning rate linearly increases from zero to $1 \times 10^{-4}$ over the first 10% of timesteps collected during meta-training, then linearly decreases back to zero over the course of the optimization process. Additionally, we clip gradients to the $[-0.25, 0.25]$ range.

### F.6 Decision Adapters

We implement the hypernetwork-based task policy using the code provided by Beukman et al. (2024) at https://github.com/Michael-Beukman/DecisionAdapter. The policy is a $(128, 128)$ feed-forward network, similar to LaSER's task policy (see Appendix D.2). It is also meta-trained with the same hyperparameters (see Table 9). However, this policy only takes the state $s$ as input. Moreover, between the policy's last hidden layer and output layer, we introduce an additional $(32, 32)$ network with `tanh` activations. The weights of this final network are generated by a hypernetwork, which takes the task context $c$ as input. Following Beukman et al. (2024), we also use a skip connection between the input and output of this network with hypernetwork-generated weights.

The hypernetwork itself is a $(64, 64)$ feed-forward network with `tanh` activations. The output weights of the hypernetwork are computed in chunks of size 16. Before passing the input $c$, we pre-process it into a 4-dimensional representation using a single `tanh`-activated layer. Finally, the actor and the critic have separate hypernetworks.

## G  Additional Results

We provide results that complement those in Sec. 5. Note that for Fig. 11, LaSER uses a pretrained encoder and exploration policy, while the baseline algorithms are trained from scratch. The choice to display adaptation efficiency measurements in this manner follows a similar argument to the one given in Sec. 5.2.3.

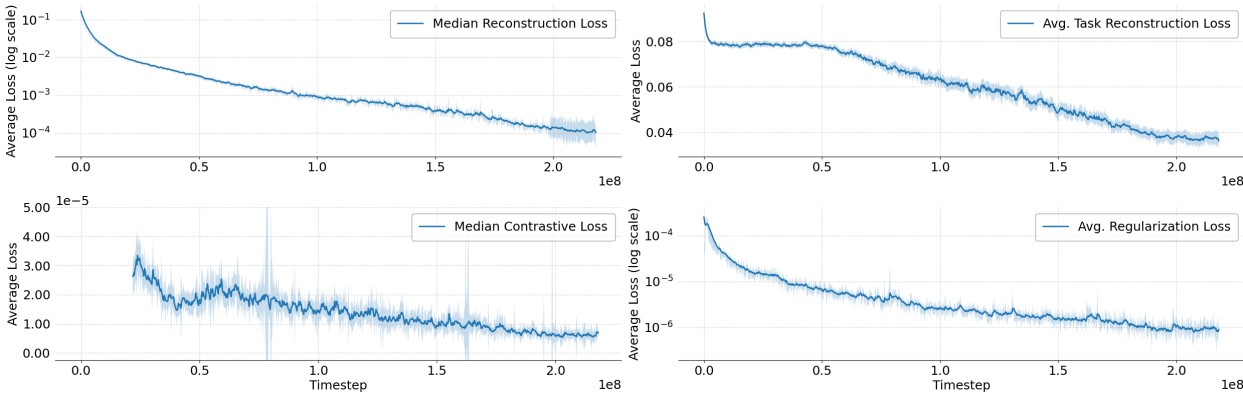

Figure 10: Encoder loss for LaSER, computed on tasks from MEWA's meta-training distribution. We show separate plots for each term in the encoder loss function. For 10 random seeds, we provide the standard error as the shaded areas, average the second and fourth metrics, and report the median for the other two. We found the median to be more appropriate, as one of the seeds was unstable towards the end of meta-training, for approximately $0.2e8$ timesteps. For this same reason, we restrict the plot of the contrastive loss to the range $[-2.5e{-}6, 5e{-}5]$. To aid readability for exponential losses, the reconstruction and regularization losses use a logarithmic scale. All metrics are smoothed using EMA with $\alpha_{\text{EMA}} = 0.2$.

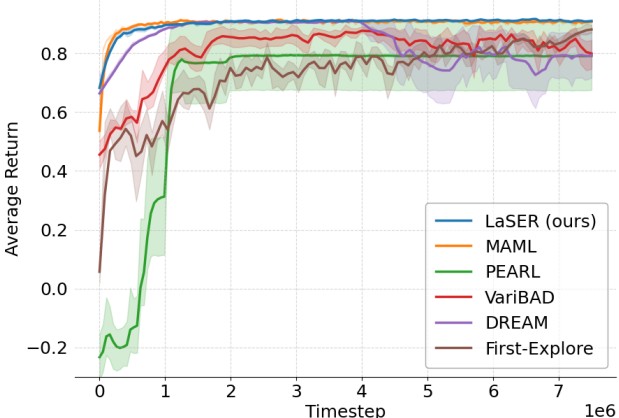

Figure 11: Undiscounted returns of task-solving policies on tasks from MEWA's meta-training task distribution. The shaded areas report the standard error of the average return across 10 seeds. We use EMA smoothing with $\alpha_{\text{EMA}} = 0.6$.

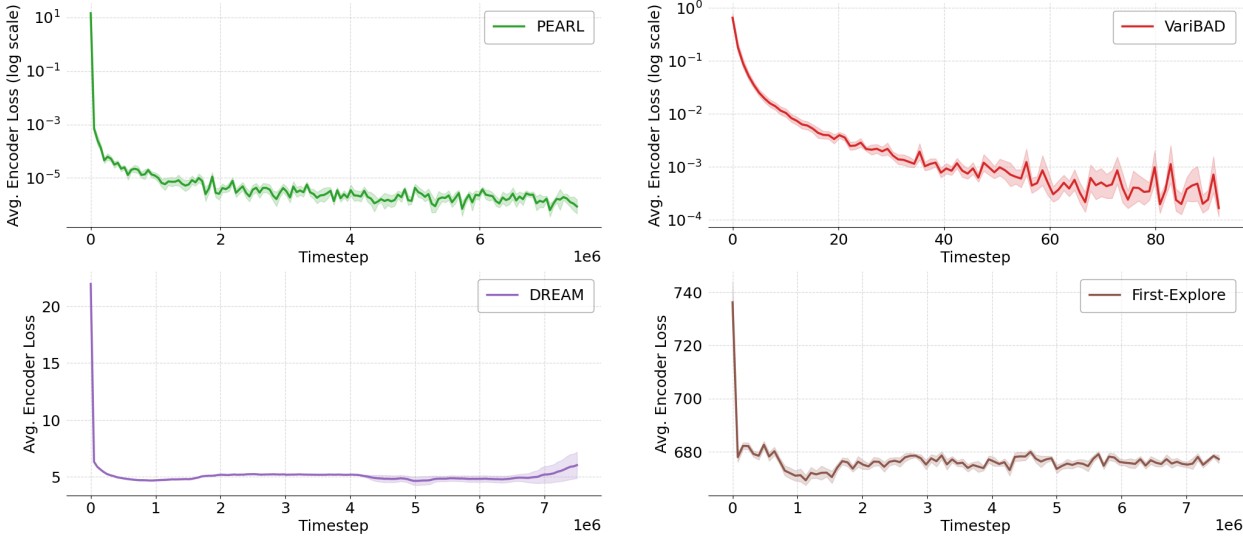

Figure 12: Encoder loss for PEARL, VariBAD, DREAM, and First-Explore on tasks from MEWA's meta-training task distribution. The shaded areas report the standard error of the average loss across 10 seeds. We use logarithmic scales for PEARL and VariBAD, as their losses decay exponentially.

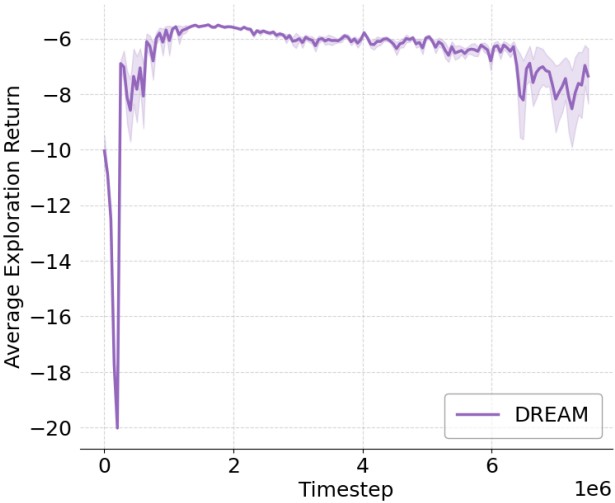

Figure 13: Intrinsic exploration return for DREAM's exploration policy on MEWA's meta-training task distribution. Shaded areas are the standard error of the average over 10 random seeds.

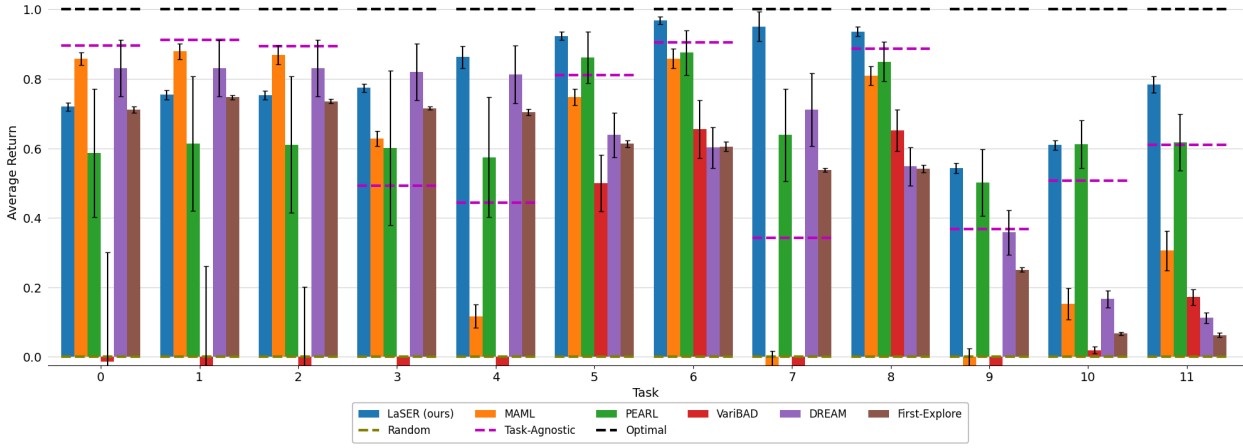

Figure 14: Average per-task returns for each of the 12 meta-testing tasks in MEWA. Each agent is meta-tested with an exploration budget of $K = 4$. Results are averaged over 10 seeds, with error bars indicating standard error across seeds. The three baselines are computed on a per-task basis. We normalize returns between the *random* and *optimal* baselines. For better visualization, we also restrict the plot to the $[-0.02, 1.0]$ range.

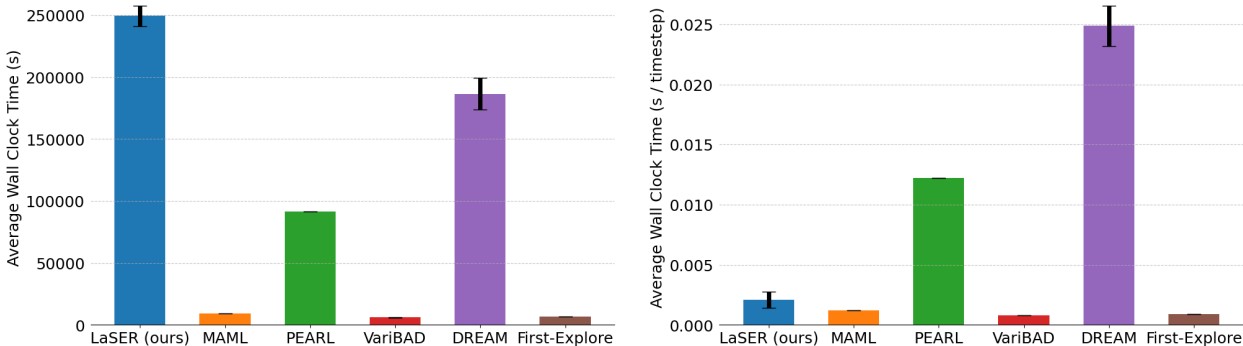

Figure 15: Wall clock time required to meta-train each method. We report the standard error of the mean over 10 seeds. (left) Total meta-training time. (right) Meta-training time averaged over all timesteps.

| Algorithm | Parameters | | Component Breakdown | |
|---|---|---|---|---|
| | Meta-Training | Meta-Testing | Component | Parameters |
| LaSER (ours) | $78,394,281$ | $8,111,314$ | Encoder $g_\omega$ | $4,484,672$ |
| | | | Encoder Heads $h_{\text{rec}}, h_{\text{t-rec}}$ | $70,282,967$ |
| | | | Exploration Policy $\pi_\phi^{\text{explore}}$ | $109,193$ |
| | | | Task Policy (No Context Extractor) $\pi_\theta$ | $141,961$ |
| | | | Task Policy Context Extractor | $3,375,488$ |
| MAML | $5,676$ | $5,676$ | Task Policy | $5,676$ |
| PEARL | $233,271$ | $233,271$ | Encoder | $87,210$ |
| | | | Task Policy | $146,061$ |
| VariBAD | $186,580$ | $178,243$ | Encoder | $76,538$ |
| | | | Decoder | $8,337$ |
| | | | Task Policy (No Context Extractor) | $101,001$ |
| | | | Task Policy Context Extractor | $704$ |
| DREAM | $1,080,546$ | $1,080,546$ | Encoder | $426,880$ |
| | | | Exploration Policy | $214,809$ |
| | | | Task Policy | $438,857$ |
| First-Explore | $10,541,192$ | $10,541,192$ | Encoder | $10,541,192$ |

Table 5: Number of parameters for each algorithm. All architectures are for the MEWA benchmark. "Meta-Training" refers to the total number of parameters required to meta-train the model, which can differ from the final number of parameters during "Meta-Testing". We additionally offer a component-wise breakdown.

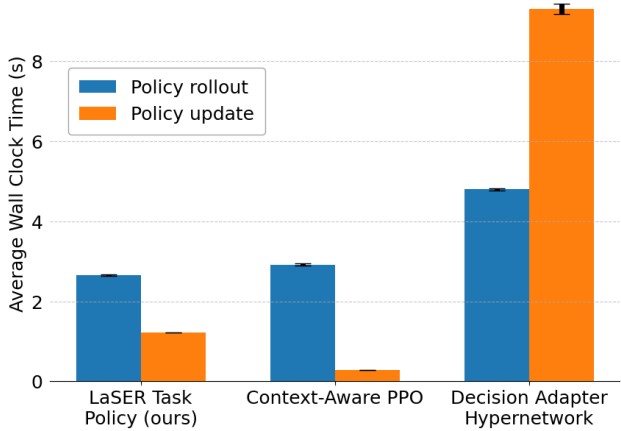

Figure 16: Wall clock time results for task policy meta-training. We measure the average time per iteration (in seconds) required to roll out episodes and optimize the policy. The results are averaged over 10 seeds. The bars represent standard error across all seeds.

### G.1 Ablation Study: Linear vs. Non-Linear Task Reconstruction

Fig. 17 compares LaSER with a variant that adopts the non-linear approach to task reconstruction, as introduced in Appendix C. Since the head $h_{\text{t-rec}}$ now acts as a hypernetwork, we increase its size from a $(256, 128, 128)$ to a $(256, 128, 128, 320, 128)$ MLP, to compensate for the increase in complexity. Apart from these changes, the linear and non-linear variants share identical hyperparameters. We model $\Psi_\psi$, the output of $h_{\text{t-rec}}$, as $Q$ distinct $(16, 16)$ MLPs with `tanh` activations.

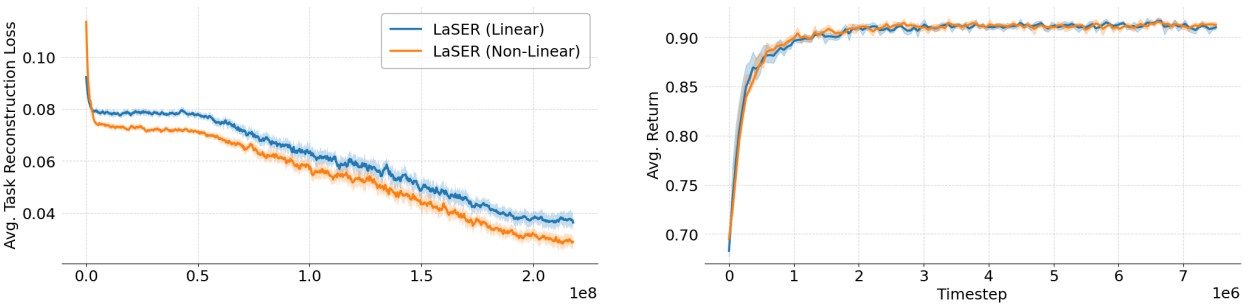

Figure 17: Performance comparison between LaSER with linear and non-linear task reconstruction on MEWA's meta-training task distribution. The meta-training setup and metrics are identical to those in Fig. 5. (left) Task reconstruction loss $\mathcal{L}_{\text{t-rec}}$ for linear (Eq. 6) and non-linear (Eq. 17) variants. We use EMA smoothing with $\alpha_{\text{EMA}} = 0.2$. (right) Undiscounted exploitation return of policy $\pi_\theta$. We use EMA smoothing with $\alpha_{\text{EMA}} = 0.6$.

The non-linear variant achieves lower task reconstruction loss compared to linear LaSER, indicating that its latent context vector $\boldsymbol{c}$ may capture more complex task structures. However, this increase in expressivity does not translate to improved performance during the task-solving phase, as the task policy achieves returns similar to those of linear LaSER. This supports our linear assumption, at least in environments similar to MEWA. Consequently, the additional complexity in terms of implementation and computation of the non-linear approach is unnecessary in this setting.

## G.2   Ablation Study: Task Solving Alternatives

Fig. 18 compares LaSER's task policy from Sec. 5.3, optimized using meta-rewards from Eq. 2, with two simpler alternatives to those meta-rewards. All models are trained with ground-truth contexts on a fixed set of tasks. The penalty-only baseline, defined as $r_t - \beta\, w(s_t, \boldsymbol{c})$, removes the entropy bonus. It penalizes the policy directly when the context $\boldsymbol{c}$ is underutilized (according to $w$), but lacks an explicit mechanism to encourage exploration that corrects this behavior. The entropy-only baseline, defined as $r_t + \beta\, S[\pi](s_t, \boldsymbol{c})$, encourages constant exploration, regardless of how effectively the context is being used. While the policy still receives $\boldsymbol{c}$ as input, the meta-reward is agnostic to how effectively this context is used. Note that both baselines in this ablation study use the same architecture and hyperparameters as the LaSER task policy, except $\beta$ is reduced by a factor of 50 in the entropy-only setting to improve stability.

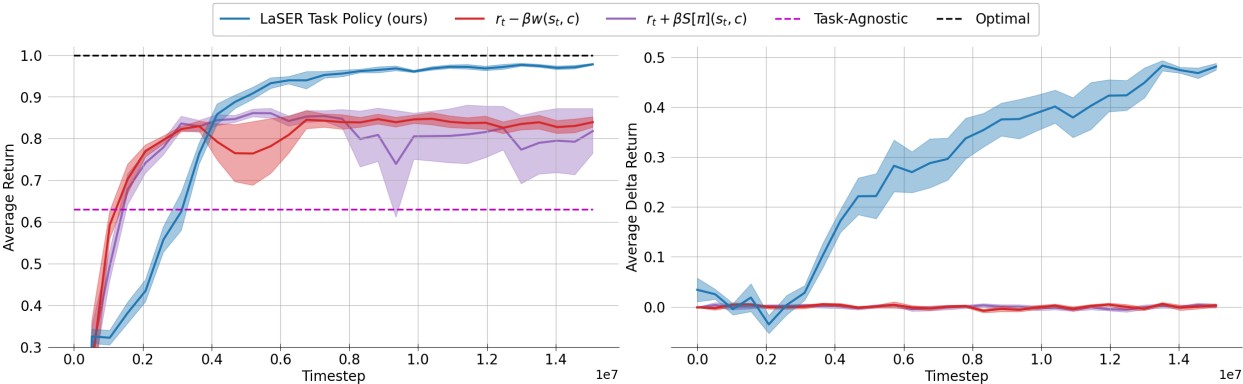

Figure 18: Ablation study on the meta-reward formulation. The experimental setup and metrics follow those in Fig. 8 (ground-truth contexts, 12 MEWA tasks). (left) Average post-adaptation return. (right) Average delta return.

Our results demonstrate that the proposed LaSER meta-reward outperforms both baselines. The penalty-only agent eventually stabilizes but converges to a suboptimal return. We hypothesize that, while this baseline is able to identify context misuse, it cannot correct it due to the lack of targeted exploration. The entropy-only baseline learns well initially, but loses stability in the second half of training. This is likely due to the non-decaying entropy bonus, which is beneficial at the start, but eventually leads to excessive exploration after the agent has learned to use the context. Critically, both baselines have average delta returns close to zero. This indicates that, compared to a policy trained with the meta-rewards in Eq. 2, the ablated agents ignore or misuse the provided context, leading to no post-adaptation improvements.

## G.3   Qualitative Results for Exploration Policy

In this section, we provide a qualitative discussion with step-by-step visualizations (see Fig. 19) of the meta-episodes collected during task exploration by three fully meta-trained agents. We use the MEWA meta-testing tasks with indexes 5 and 7 in our evaluation (see Table 4).

Task 5 is in-distribution and has an average mistake probability of 0.563. It acts as a "middle ground" task in the MEWA curated distribution: complex enough to allow frequent mistakes, without having almost unavoidable mistakes.

In Fig. 19a, VariBAD adopts a simple exploration strategy for task 5, with episodes that follow similar patterns. Specifically, it takes only red and blue actions in the first half of each episode, then green and orange in the second half.

The visualized episodes suggest that DREAM can achieve more diverse exploration than VariBAD, as it experiments with three out of four mistake types. DREAM only appears to struggle at the beginning of episodes, where the sequence of the first 5 actions is always identical, while the action at timestep 5 appears to depend on whether a mistake has happened at timestep 4. This may suggest that the diversity seen in

this meta-episode may partially be a consequence of the environment, rather than being purely driven by DREAM's ability to collect distinct data.

LaSER demonstrates a structured approach to task exploration that appears on par with DREAM. While it chooses to focus mostly on mistakes of types 3 and 4, it experiments with three distinct mistake types in the very first episode, and eventually probes the final type in the third episode. Therefore, as opposed to DREAM, it explores all four mistake types. It additionally avoids taking the same sequence of actions in the first five timesteps of each episode. This is likely due to LaSER being explicitly optimized to avoid redundancy by learning to collect episodes that are orthogonal in the latent exploration space, while DREAM is only exploring to maximize future exploitation gains.

Task 7 is an out-of-distribution task with an average mistake probability of 0.617. A special property of this particular task is that mistakes of types 1, 2, and 3 have a relatively high chance of occurring, while mistakes of type 4 never occur.

In Fig. 19b, VariBAD utilizes the same exploration policy in the first three episodes. Specifically, the agent continuously attempts to avoid a mistake of type 2, then one of type 1, and finally one of type 3. This results in three episodes that can only be distinguished based on the environment's reactions, rather than VariBAD's active choices. The agent adapts its exploration strategy in the final episode. However, it never attempts a risky action that could lead to a type 4 mistake, and thus fails to learn that these never occur.

DREAM collects a meta-episode composed of four identical episodes. Since the DREAM agent has been meta-trained to collect episodes that maximize subsequent exploitation returns, it fails to diversify its behavior in this task. Instead, it chooses to follow the simplest approach of exploiting the task's type four mistake, leading to a collapse into a deterministic policy. Consequently, the lack of experimentation with mistakes of other types implies that this specific meta-episode is not sufficient to learn the true specifications of this task.

LaSER avoids falling into the same pitfall as DREAM, and successfully collects a diverse meta-episode. The first episode is similar to those of DREAM's, quickly highlighting the low likelihood of a type 4 mistake occurring. However, the agent then decides to also experiment with mistakes of types 2 and 3 in subsequent episodes. This exploration strategy is likely a result of LaSER's meta-training, where diverse task exploration is explicitly encouraged, regardless of the task's structure or of the optimal exploitation policy. Its exploration policy is thus capable of ignoring distracting environmental rewards, instead collecting a set of $K$ episodes that, according to the agent, best approximates a large dataset constructed from this task.

## H   Hyperparameters

Tables 6, 7, 8, 9 show the hyperparameters used by LaSER in each of the three benchmarks.

| General Hyperparameters | | | | |
|---|---|---|---|---|
| | **MEWA** | **Meta-World** | **HopperMass** | **Notes** |
| $N_{explore}$ | $10,000$ | $700$ | $600$ | |
| $N_{explore}^{initial}$ | $2,000$ | $20$ | $30$ | |
| $H$ | $50$ | $200$ | $400$ | Horizon |
| $K$ | $4$ | $2$ | $4$ | Episodes per meta-episode |

Table 6: LaSER hyperparameters used throughout meta-training and meta-testing.

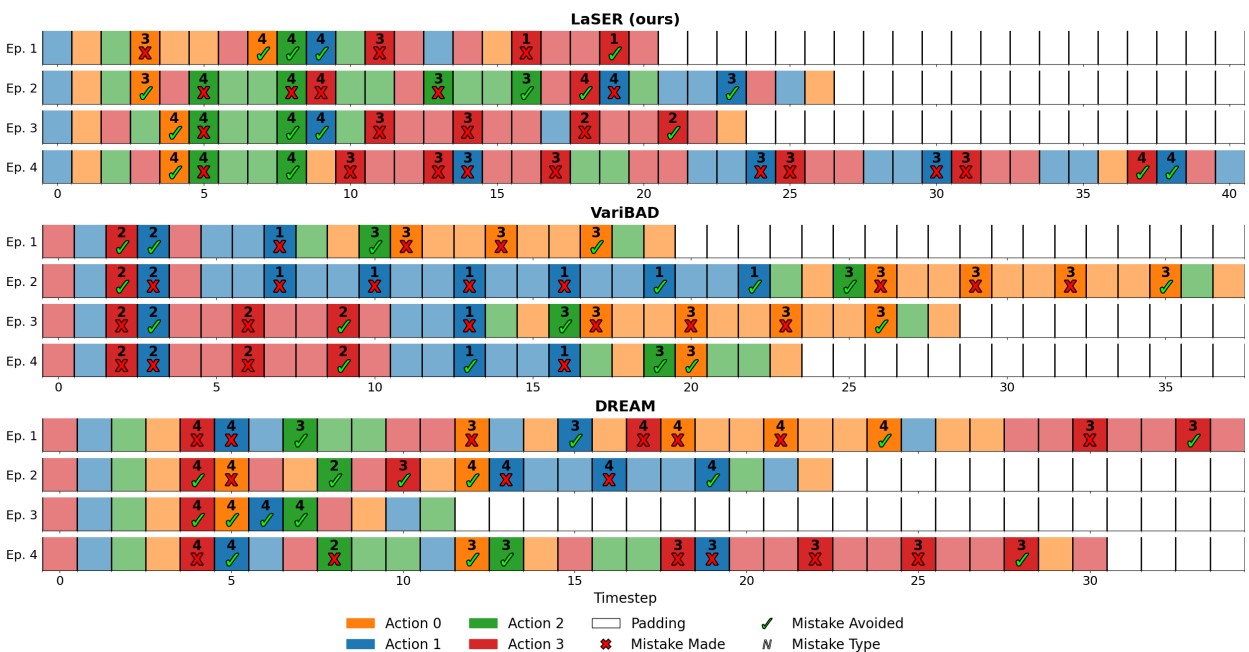

(a) MEWA's 5-th meta-testing task; 0.563 average mistake probability.

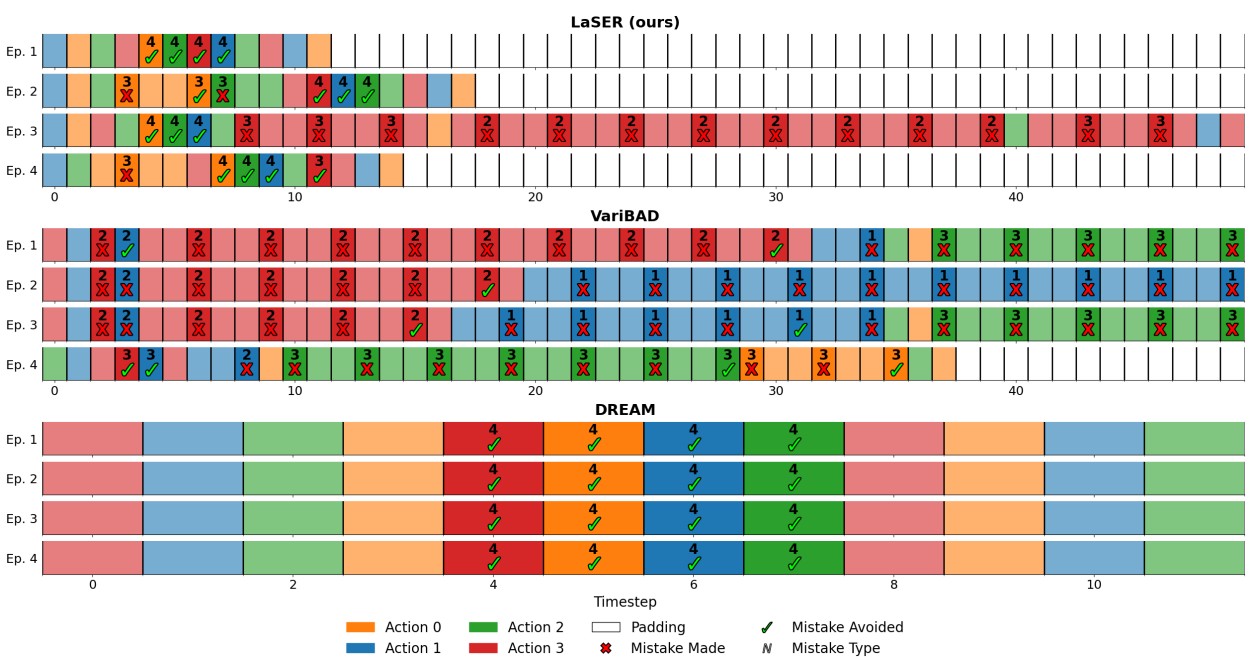

(b) MEWA's 7-th meta-testing task; 0.617 average mistake probability.

Figure 19: Qualitative comparison of exploration behaviors on two MEWA tasks. We plot color-coded sequences of actions for exploration meta-episodes of $K = 4$ episodes. Solid colors highlight risky actions (i.e., those that may lead to mistakes). The symbols indicate whether a mistake occurred or was avoided after a risky action, while numbers indicate the type of mistake. Each episode is padded to the maximum length of its respective meta-episode, and each episode terminates automatically after $H = 50$ timesteps.

| Encoder $g_\omega$ | | | | |
|---|---|---|---|---|
| | **MEWA** | **Meta-World** | **HopperMass** | **Notes** |
| Learning rate | $1 \times 10^{-4}$ | | $5 \times 10^{-4}$ | |
| $N_{\text{encoder}}$ | 50 | 2 | | Encoder updates per iteration |
| $Q$ | 20 | | | Meta-episodes per task |
| $\|\mathcal{B}\|$ | 5 | 10 | | # of tasks per batch |
| Buffer size | $100,000$ | $1,000$ | $2,000$ | Maximum # of tasks in buffer |
| $c_{\text{rec}}, c_{\text{t-rec}}, c_{\text{contr}}, c_{\mathcal{R}}$ | 0.5, 1, 1, 0.125 | | | Coefficients for loss $\mathcal{L}_{\text{LaSER}}$; Eq. 10 |
| $\xi$ | 0.05 | | | Constant in Eq. 7 |
| $\nu$ | 250 | 25 | | Update delay for parameters $\hat{\omega}$ |

Table 7: LaSER hyperparameters used for meta-training the encoder.

| Exploration Policy $\pi_\phi^{\text{explore}}$ | | | | |
|---|---|---|---|---|
| | **MEWA** | **Meta-World** | **HopperMass** | **Notes** |
| Learning rate | $1 \times 10^{-5}$ | | $8 \times 10^{-5}$ | |
| $\sigma$ | 0.025 | 0.01 | 0.003 | Constant in Eq. 4 |
| $\epsilon$ | 0.1 | | | PPO clipping (exploration); Eq. 13 |
| $\alpha_r$ | 0 | | 0.01 | Environmental reward weight; Eq. 11 |

Table 8: LaSER hyperparameters used for meta-training the exploration policy.

| Task Policy $\pi_\theta$ | | | | |
|---|---|---|---|---|
| | **MEWA** | **Meta-World** | **HopperMass** | **Notes** |
| Learning rate | $1 \times 10^{-4}$ | | $6 \times 10^{-5}$ | |
| $N_{\text{task}}$ | $10,000$ | $9,000$ | $2,500$ | |
| $\beta$ | 1.5 | 0 | | Coefficient in Eq. 2 |
| $\zeta$ | $-0.13$ | 0 | | Threshold in Eq. 3 |
| $\epsilon$ | 0.2 | | | PPO clipping (exploitation); Eq. 13 |
| $c_{\text{PFO}}$ | 0.1 | 0 | | PFO coefficient |
| $N_{\text{PPO}}$ | 4 | 2 | 4 | |
| # minibatches | 2 | 8 | | |
| Layers | $(128, 128, 128)$ | $(256, 256, 128)$ | | |
| Context encoder layers | $(128, 128, 128)$ | $(256, 256, 128, 10)$ | | Encoder for context vector $\boldsymbol{c}$ |

Table 9: LaSER hyperparameters used for meta-training the task policy.

