# OpenReview forum: "Task-Specific Exploration in Meta-Reinforcement Learning via Task Reconstruction"
_TMLR — Accepted by TMLR_

### Review · Reviewer_EqNm · 2025-12-15

**Summary Of Contributions:**

This paper addresses an important limitation in existing meta-reinforcement learning methods: even when informative task context is available, learned policies often fail to actually exploit it. The authors argue that this problem arises from two coupled issues—inefficient task exploration and weak optimization on context-conditioned policies to meaningfully use the inferred task information.

The paper proposes LaSER, a meta-RL framework that explicitly separates task exploration and task exploitation at inference time. During exploration, a dedicated policy $ \pi_{\text{explore}} $ is trained to collect a fixed number $K$ of informative episodes. Informativeness is defined through a task reconstruction objective: a small set of episodes is considered useful if it can approximately reconstruct a larger dataset from the same task. This yields a principled, reconstruction-based notion of exploration quality that goes beyond diversity or novelty measures.

To address failures in context exploitation, the authors introduce a meta-reward that penalizes the task policy for ignoring the inferred task context. Concretely, the method selectively encourages entropy when conditioning on the context does not improve value, pushing the policy to further explore the context–action relationship instead of collapsing to context-agnostic behavior.

Empirically, the paper shows that LaSER outperforms or matches strong meta-RL baselines across several benchmarks. The experimental section includes  oracle-context evaluations and ablations, which support the claim that both improved exploration and improved context exploitation are necessary for effective few-shot adaptation.

**Strengths**

- Introduces a principled definition of informative exploration grounded in task reconstruction, rather than heuristic notions of diversity or information gain.
- Clearly identifies and empirically validates an underexplored failure mode in meta-RL: policies ignoring task context even when high-quality context is available.
- Integrates exploration learning, representation learning, and task policy optimization into a coherent framework.
- Includes strong diagnostic experiments (e.g., oracle-context evaluations) and ablations that justify key design choices.

**Weaknesses**

- The method relies on a strong inductive assumption that task data lies in an approximately low-rank *linear* subspace with fix rank $K$. While this assumption is central to the reconstruction objective, the paper provides limited evidence that it holds beyond the considered benchmarks. It remains unclear how robust the approach would be in task families with inherently nonlinear or compositional structure, or whether reconstruction quality correlates with downstream adaptation performance in such settings.
- The exploration budget $K$ is fixed and specified during meta-training, which limits flexibility at test time. In practice, different tasks may require different amounts of exploration, and the framework does not currently allow the agent to adaptively decide when sufficient task information has been collected.
- The exploration policy and encoder are pretrained before task policy learning, effectively shifting data and computation to earlier stages. This makes comparisons of early-stage sample efficiency with baselines difficult to interpret and would benefit from alternative evaluation protocols (see Requested Changes).
- Baseline coverage becomes narrower in some experimental sections without explicit justification, particularly when earlier baselines are omitted despite being evaluated elsewhere in the paper (see Requested Changes).

**Additional Comments:**

Although the framework is built around a fixed exploration budget $K$ which is common in many meta-RL baselines, the overall structure of LaSER appears well suited to more flexible formulations. In particular, the reconstruction-based objective naturally suggests the possibility of adapting or selecting $K$ dynamically, for example based on marginal gains in reconstruction quality. Allowing the model to determine when sufficient task information has been collected could significantly strengthen the approach and broaden its applicability.

Finally, while the paper deliberately separates exploration and exploitation into two distinct phases with two policies, it may be interesting to consider formulations closer to standard reinforcement learning, where exploration and exploitation are interleaved rather than strictly phased. For instance, allowing the exploration policy to continue contributing data during task execution, with the task encoding updated online, could enable more progressive adaptation and reduce reliance on a hard phase boundary.

Overall, these points represent promising directions for future work and do not detract from the paper’s core contributions. However, addressing them, even at the level of discussion, would further strengthen the paper.

**Audience:**

Yes

**Audience Explanation:**

Although the method introduces strong inductive biases like low-rank linear reconstruction assumption and a fixed exploration budget (which is common among many existing meta-RL approaches), the overall framework is well motivated and technically insightful. In particular, the explicit decoupling of the exploration policy and the exploitation policy represents a clear conceptual contribution and helps expose important failure modes in context-conditioned policy learning that are often obscured in end-to-end approaches.

Moreover, the results reported in MEWA show a non-trivial performance gap over the selected baselines, which supports the practical effectiveness of the proposed method. While the assumptions underlying the approach may limit its applicability in some settings, the ideas introduced in this paper are likely to be of interest to the TMLR audience and to stimulate further work on structured exploration and context utilization in meta-RL.

**Broader Impact Concerns:**

I do not identify any significant ethical or societal concerns arising directly from this work.

**Claims And Evidence:**

Yes

**Claims Explanation:**

The experimental results largely support the paper’s main claims. The authors report performance improvements across multiple benchmarks and include useful diagnostic experiments, such as oracle-context evaluations, that help isolate key failure modes. The presented ablation studies also support the importance reconstruction-based exploration objective and the proposed meta-reward for encouraging context usage. Overall, the empirical evidence is mostly consistent with the paper’s stated motivations and conclusions.

However, several aspects of the empirical evaluation would benefit from clearer justification and more careful framing. In particular, comparisons of sample efficiency are difficult to interpret without explicitly accounting for the pretraining of the encoder and exploration policy, which provides the task policy with a significantly more informative initialization during meta-training. This effectively shifts data and computation to an earlier stage and should be discussed more explicitly when making efficiency claims.

Additionally, in Section 5.2.3, results are reported against only a single baseline, which was not consistently the strongest performer in earlier experiment. Without a clear justification for this choice, it is difficult to assess the strength of the reported gains in this setting. Including additional baselines or explaining their omission would substantially improve transparency.

Overall, these issues do not fundamentally undermine the paper’s claims, but addressing them would make the empirical evaluation more convincing and the overall narrative more complete.

**Requested Changes:**

1. **Improve clarity and narrative structure.**
The paper would benefit from substantial rewriting to improve clarity and overall narrative flow. While the core ideas are interesting, the current presentation is dense and does not consistently convey the main intuitions. I recommend shortening the main paper and moving secondary technical details to the appendix, with greater emphasis on motivating the problem, explaining key design choices, and clarifying how the different components of the framework fit together.

2. **Clarify baseline selection in Section 5.2.3.**
In Section 5.2.3, only VariBAD is included as a baseline. It is unclear why other baselines evaluated earlier in the paper are omitted in this setting. Providing an explicit justification for this choice, or including additional baselines, would improve transparency and strengthen the empirical comparison.

3. **Clarify sample efficiency trade-offs.**
The paper would benefit from a clearer discussion of sample efficiency. Since LaSER pretrains the encoder and the exploration policy, it effectively shifts computation and data usage to the training phase. Even if this leads to lower overall sample efficiency, such a trade-off can still be valuable in practice. The authors should clarify whether LaSER is particularly helpful in *data-scarce adaptation settings*, where test-time interaction is limited, and explicitly discuss the benefits and drawbacks of this design choice.

4. **Additional ablation studies on the meta-reward formulation.**
The task-policy meta-reward is defined as
$r_t^{+} = r_t + \beta\, w(s_t, c)\, \mathcal{S}[\pi](s_t, c),$
where $ w(s_t, c) $ measures context underutilization and $ \mathcal{S}[\pi] $ denotes policy entropy.
To better isolate the contribution of this specific design, it would be helpful to include ablation studies comparing against simpler alternatives, such as
$r_t^{+} = r_t + \beta\ \mathcal{S}[\pi](s_t, c),$
(always-on entropy bonus), and
$r_t^{+} = r_t - \beta\ w(s_t, c),$
(direct penalty for context underutilization).
These comparisons would help clarify whether the primary benefit arises from entropy regularization, the context-usage signal $w$, or their interaction.

5. **Discussion of structural assumptions.**
The paper would benefit from a more explicit discussion comparing LaSER’s *explicit low-rank linear reconstruction assumption* with the *implicit structural assumptions* made by baseline methods. Such a discussion would help contextualize the inductive biases introduced by LaSER relative to existing meta-RL approaches.

6. An important open question concerns the generalizability of the proposed framework beyond linear reconstruction. While the current formulation relies explicitly on a low-rank linear structure to define the informativeness of exploration data, it would be helpful for the authors to discuss how the approach might extend to settings where task structure is inherently nonlinear, or to clarify the regimes in which the linear assumption is most appropriate.

---

> ### Author Response · Authors · 2026-01-24
> **Reply to Reviewer EqNm's comments**
>
> We thank the reviewer for the detailed feedback and insights, and for bringing attention to some of the issues in our work. We also especially appreciate the comment regarding our work's focus on exploration through task reconstruction rather than the more common heuristic notions.
>
> We have updated the manuscript to address your requested changes. We provide details below.
>
> 1. **Improve clarity and narrative structure.**
>    We have revised the text in Sec. 3 to improve flow and clarity. We focused specifically on providing intuition before the dense math notation. We hope these improvements address the clarity concerns.
>    Regarding the placement of technical details: we carefully reviewed Sec. 3. We respectfully believe that maintaining the formal definitions of the meta-reward, exploration reward, task reconstruction, and contrastive objective in the main body is crucial for the method description to be complete and self-contained. The additional detail in Sec. 3.4 is also necessary to introduce the algorithm built on these ideas. However, if you still feel specific details regarding our ideas, algorithm, or architecture still interrupt the flow, we are open to moving them to the appendix.
> 2. **Clarify baseline selection in Section 5.2.3.**
>    We have added a more detailed justification for this choice in Sec. 5.1.2 and referenced it in Sec. 5.2.3. As detailed in the text, VariBAD is known to outperform MAML and PEARL in complex, continuous environments, such as Meta-World. Regarding DREAM, Liu et al. (2021) explicitly note that their algorithm has not been designed for environments where naive exploration may be sufficient. For this specific reason, they choose not to evaluate DREAM on Meta-World, so it is unclear whether it can outperform simpler baselines in such environments.
> 3. **Clarify sample efficiency trade-offs.**
>    We have added additional discussions in Sec. 5.1.3 and Sec. 6. In the former, we acknowledge the computation shift and discuss the trade-offs encountered during meta-training and meta-testing. In the latter, we give a more practical example of environments that are data-scarce during deployment/meta-testing and how LaSER may be beneficial in such situations.
> 4. **Additional ablation studies on the meta-reward formulation.**
>    We implemented your suggested variations to our meta-reward and provided a new ablation study. We appreciate this suggestion, as it made our overall analysis more complete and introduced two interesting alternatives to our idea. The discussion of the results is in Sec. 5.3, with visualizations and a detailed analysis in Appendix H.2. We find that our original formulation outperforms both alternatives in terms of stability, returns, and delta returns.
> 5. **Discussion of structural assumptions.**
>    We expanded Sec. 4 to include a discussion on structural assumptions. We contrast LaSER's explicit low-linear reconstruction assumption with the implicit assumptions made by other methods (e.g., we discuss specific architectures focused on state or reward reconstruction). We argue that an explicit assumption allows more directed task exploration, while implicit methods may lack clear objectives effective in the meta-RL setting.
> 6. **Beyond linear reconstruction.**
>    We discuss the implications of adapting our algorithm to non-linear reconstruction in Sec. 3.3 (also see footnote 3). We also discuss the risk of having non-linear encoders be too expressive, leading to trivial, input-agnostic reconstructions of the tensor $B$. We then provide a formal extension of LaSER to non-linear task reconstruction (Appendix E), by updating Assumption 1 and turning our $h_\text{t-rec}$ head into a hypernetwork. In Sec. 5.2.1 and Appendix H.1, we perform an ablation study on MEWA to compare this non-linear variant with standard LaSER. While we find the task reconstruction loss to be slightly lower, the exploitation returns are similar for both models. This new study, together with our results on Meta-World and HopperMass, suggests that the linear assumption is sufficient for the benchmarks considered, although the non-linear variant remains viable for more complex distributions.
>
> We would also like to thank you for the additional comments, which provided interesting ideas and insights. We discuss a potential future extension of LaSER to a dynamic exploration budget $K$ in Sec. 6.
>
> Your idea for more interleaved exploration and exploitation is similar to our brief discussion in Appendix D.3. We extend this discussion to mention variants of LaSER that allow the encoder to be updated even after the pre-training phase or allow other data to contribute to the encoding of context $c$, e.g., episodes collected by the task policy.
>
>
> References
> - Liu, E.Z., Raghunathan, A., Liang, P. and Finn, C., 2021, July. Decoupling exploration and exploitation for meta-reinforcement learning without sacrifices. In International conference on machine learning (pp. 6925-6935). PMLR.

---

> > ### Comment · Reviewer_EqNm · 2026-02-08
> >
> > Thanks for the response and for addressing most of my comments. The added ablations, baseline justification, and new discussion meaningfully strengthen the paper, and I appreciate the effort to engage with the feedback seriously.
> > But still, the paper is dense. The technical contribution is solid, but the presentation still needs work. Some parts are unnecessarily hard to follow. Overall, my main concerns are largely addressed, but I still believe the paper’s impact would improve with more attention to presentation and narrative polish.

---

### Review · Reviewer_X7vr · 2025-12-31

**Summary Of Contributions:**

This paper proposes LaSER, a meta-RL framework aimed at improving few-shot adaptation through task-specific, sample-efficient exploration. The key idea is to formalize informative exploration as identifying a small set of episodes that can linearly reconstruct a larger dataset collected from the same task, under a low-rank assumption. Based on this perspective, the authors introduce a task reconstruction objective to supervise representation learning for exploration, an intrinsic exploration reward that encourages episode-level diversity in a learned latent space, and a novel meta-reward to address the issue that in-context policies may fail to exploit task contexts even when they are available. The method is evaluated primarily on the MEWA benchmark, where exploration is essential, and additionally on Meta-World and HopperMass to demonstrate generality. Empirical results show that LaSER significantly outperforms established meta-RL baselines on exploration-centric tasks, while providing competitive performance on standard benchmarks.

**Audience:**

No

**Audience Explanation:**

At least a substantial subset of TMLR’s audience would be interested in this paper’s findings, particularly researchers working on meta-reinforcement learning, exploration, representation learning, and in-context adaptation. The paper addresses a well-known but under-explored failure mode in meta-RL: inefficient task exploration and the inability of in-context policies to effectively exploit learned task contexts, and proposes a principled solution grounded in a clear modeling assumption.

**Broader Impact Concerns:**

This paper does not raise broader impact concerns. It focuses on algorithmic contributions to meta-reinforcement learning, specifically task exploration, representation learning, and policy optimization in simulated environments. The proposed methods are evaluated exclusively on standard benchmark tasks and do not involve human subjects, personal data, deployment in real-world decision-making systems, or applications with foreseeable societal, ethical, or safety implications.

**Claims And Evidence:**

Yes

**Claims Explanation:**

Pros.
- Clear and novel conceptual framing of exploration in meta-RL. The idea of low-rank task encoder is clean and technically sound. This is a novel perspective of task modeling in meta-RL.
- Well-motivated optimization contribution addressing context underutilization. The proposed meta-reward explicitly tackles the empirically observed but under-discussed issue that in-context policies may ignore task context even when it is informative.
- The implementation is provided to improve reproducibility.

Cons.
- Limited qualitative analysis of exploration behavior. While quantitative results are strong, the paper lacks trajectory-level or qualitative examples illustrating how the learned exploration policy behaves differently from baselines. Given the interpretability of MEWA, concrete behavioral visualizations or walkthroughs would substantially strengthen the claims about task-specific exploration.
- Assumption of linear task reconstruction is not empirically stress-tested. Considering the complexity of tasks in the wild, the linear task reconstruction could be unrealistic if task identity is not representable as a low-rank combination of episode-level information, such as in environments with highly continuous, nonlinear, or noise-dominated task variations.
- Missing discussion of recent meta-RL works. Zhou, Hongtu, et al. "CERTAIN: Context Uncertainty-aware One-Shot Adaptation for Context-based Offline Meta Reinforcement Learning." Forty-second International Conference on Machine Learning. Chu, Zhendong, et al. "Meta-reinforcement learning via exploratory task clustering." Proceedings of the AAAI Conference on Artificial Intelligence. Vol. 38. No. 10. 2024.

**Requested Changes:**

Please see the cons.

---

> ### Author Response · Authors · 2026-01-24
> **Reply to Reviewer X7vr's comments**
>
> We thank the reviewer for the thoughtful feedback and constructive criticism. We would also like to thank the reviewer for the highlights regarding the clarity and novelty of our take on meta-RL.
>
> We hope that the changes highlighted below address your concerns.
>
> 1. **Qualitative analysis of exploration behavior.**
>    We agree that qualitative visualizations will strengthen our claims. We therefore add Fig. 19, where we examine meta-episodes collected from two MEWA tasks during the exploration phase. The first is an in-distribution MEWA task with a moderate chance of mistakes occurring, while the second is an OOD task with the special property that mistakes of type 4 never occur, while the other types have a high likelihood. We compare meta-episodes collected by LaSER to those collected by VariBAD and DREAM. We provide a discussion of these qualitative results in Sec. 5.4 and Appendix H.3. Our conclusions regarding this study are that LaSER appears to avoid redundancy, especially compared to VariBAD, which tends to repeat similar action sequences. DREAM also appears to achieve diverse exploration in the first task, but the agent gets confused by the structure of the second task, collapsing into a deterministic exploration policy. On the contrary, LaSER remains diverse in that same task, likely due to its explicit redundancy avoidance and its exploration approach being agnostic to environmental rewards.
> 2. **Stress-testing the linear task reconstruction assumption**
>    To empirically test our assumption, we have implemented a non-linear extension of LaSER (briefly discussed in Sec. 3.3 and formally defined in Appendix E) and performed a comparative ablation study (Sec. 5.2.1 and Appendix H.1). Our non-linear variant simply introduces a non-linear function to replace the linear transformation in Assumption 1. We also turn our task reconstruction head $h_\text{t-rec}$ into the hypernetwork that computes the parameters of this new function. Our results show that the non-linear variant achieves a lower task reconstruction loss on MEWA, which is expected, given the increase in expressivity. This, however, does not translate to increased gains during exploitation, as both linear and non-linear LaSER achieve similar returns. This suggests that the linear assumption is sufficient for MEWA. Moreover, our competitive results on Meta-World and HopperMass indicate that the linear assumption can also handle more continuous and non-linear task distributions. However, the non-linear variant may still prove useful in other, more complex environments.
> 3. **Discussion of recent meta-RL works**
>    We thank the reviewer for pointing out an important missing reference. We discuss Zhou et al. (2025) in the "Meta-Learning Contexts" part of Sec. 4. We specifically highlight CERTAIN's aim of measuring context reliability/quality to avoid ambiguous or OOD contexts, then discuss similarities with LaSER's goal of measuring the reconstruction potential of meta-episodes.
>    We extend our discussion of Chu et al. (2024) in Sec. 4. We especially highlight their exploration method based on identifying tasks through a divide-and-conquer approach.
>
>
> References
> - Zhou, H., Yang, R., Zhu, Y., Zhao, H., Zhang, H., Zhang, D., Zhao, J., Ye, C. and Jiang, C., CERTAIN: Context Uncertainty-aware One-Shot Adaptation for Context-based Offline Meta Reinforcement Learning. In Forty-second International Conference on Machine Learning.
> - Chu, Z., Cai, R. and Wang, H., 2024, March. Meta-reinforcement learning via exploratory task clustering. In _Proceedings of the AAAI Conference on Artificial Intelligence_ (Vol. 38, No. 10, pp. 11633-11641).

---

> > ### Comment · Reviewer_X7vr · 2026-02-08
> >
> > I appreciate the added experiments and discussions. However, I still concern about the generalization of the linear task reconstruction assumption in more complicated real-world environments. I suggest the authors to extend the assumption in future works and I think the current version make it a solid contribution to the meta-RL community.

---

### Review · Reviewer_soxu · 2026-01-12

**Summary Of Contributions:**

This paper proposes a full Meta RL approach, that learns an exploration policy that aims to "meta-explore", in particular, to find informative episodes considering the episodes it has already collected. Beyond this they develop a context extraction approach based on a linear task reconstruction assumption, and use this to train a task solving policy that effectively uses the context. This is done by incentivising it to have higher entropy when it is not fully using the contextual information.

##### Strengths
- I like how the work explicitly separates the task exploration and the task solving parts of the meta RL process, and uses quite different techniques for both.
- The results in MEWA seem reasonable, in that Laser is one of the only algorithms that beats the task agnostic baseline.
- The linear subspace idea is neat.
- Using the difference between a context-conditioned value function and an unconditional one to determine if the context is being used is also a clever idea, and I suspect that can be used on its own in several settings.
- The idea of incentivising the different episodes to be informative, and different from each other, is a good idea.
- The ablation results in section 5.3 seem more convincing. particularly how large the delta return is when given access to the context
- The visualisations in fig 9 are also quite useful


##### Weaknesses

- It would be great to compare against the much simpler algorithm from this paper, or explain why it wouldn't be a fair comparison. In particular, because this paper also has this separation between exploration and exploitation policies.
		- "First-Explore, then Exploit: Meta-Learning to Solve Hard Exploration-Exploitation Trade-Offs"
- The method is quite complex and has so many moving parts, that I wonder if it will hinder adoptability and other research building on it. The open source code is useful though
- The results in meta-world do not seem much different from varibad, although this perhaps just means that the benchmark is not challenging enough.
- The meta training number of parameters is significantly more than all other baselines (e.g. more than 300x the number of parameters in PEARL). I wonder if those baselines were given more parameters or more compute if they would do better. While the authors do mention that there are fewer meta test parameters, the difference is quite substantial.
- It took me a few passes to properly understand what was going on in section 3. I read it as very dense, but also verbose at the same time, and I found it hard to follow all of the different parts. I don't know if I have a great suggestion for how to address this though. I think perhaps one part of this is that there are so many components
	- In particular, the last part of section 3.3 makes sense but is quite dense and quite hard to understand on a first pass.  Also the jump from how to find $g$ to how to find $f_u$ is quite sudden

**Additional Comments:**

##### Questions:
- Do you train each method for a different number of timesteps?

**Audience:**

Yes

**Audience Explanation:**

Yes, this paper does present a new meta learning approach that seems to effectively explore, and make use of contextual information, which some other approaches don't do as well. It has a number of interesting ideas that I think are worth sharing.

**Claims And Evidence:**

Yes

**Claims Explanation:**

I think this is partly true.

In particular, the exposition is somewhat unclear, and the paper is quite verbose / long, which dilutes it a bit in my opinion.
However, the claims are reasonably supported, although the positive empirical results rely mostly on one benchmark, MEWA. I don't think that is a massive problem though.

Their new algorithm, laser, does learn the context using a small dataset, and the objective of the exploration policy seems to incentivise it to collect a diverse set of experience.

Their architecture also makes use of the context more effectively when solving task problems.

**Requested Changes:**

##### Major changes
- I think section 3 is quite unclear and it  make the whole paper hard to follow. I am unsure how to improve it, but perhaps making it feel like less of a jump between subsections could help.
- Figure 8: Having a baseline that is just state and context concatenated, using normal PPO, would be great.

###### From the weaknesses:
- It would be great to compare against the much simpler algorithm from this paper, or explain why it wouldn't be a fair comparison.
		- "First-Explore, then Exploit: Meta-Learning to Solve Hard Exploration-Exploitation Trade-Offs"
- The meta training number of parameters is significantly more than all other baselines (e.g. more than 300x the number of parameters in PEARL). I wonder if those baselines were given more parameters or more compute if they would do better.
	- A discussion or experiment for this would be useful.
##### Small Points that are not critical but I think would improve the work
- Switching the notation from $D^{(K)}$ to $B_j$ is a bit confusing in section 3.2 -> 3.3
- This can be made clearer I think:
	- "We therefore optimize $f_u$ to learn a latent space that encodes this information."
	- The paragraph after equation (8) does a good job of explaining this, and I wonder if it makes sense to have that, or something similar, before the dense maths and notation
-  I think it would be clearer to replace "The latter head" on page 8 with "$h_\text{rec}$", and specifying in more depth how the reconstruction works, like figure 4(a) would be beneficial here too.
-  I wonder if it would be better to move the meta training complexity results of 5.1.3 to after the main results, but I am not very tied to this idea.
-  I think the discussion of figure 7 (left) reads too much into the small spikes. The curves of variabad and laser overlap with the error bars, so I would argue that they perform the same on metaworld. On hopper mass I would say laser performs slightly better, and is a bit more stable.
- In section 3.2, it is mentioned that $d_k$ is the similarity between the $k$-th episode and all the others, but then you mention that you use causal masking to compute this for only the episodes $i < k$; I would perhaps suggest using different notation in equation (4) then to indicate this.
- One other experiment that would be nice is showing that the exploration policy qualitatively or quantitatively explores differently to other algorithms / that it does some intelligent exploration based on the existing episodes it has already explored. This would nicely complement the results showing that the laser task solving policy does actually use the context.

---

> ### Author Response · Authors · 2026-01-24
> **Reply to Reviewer soxu's comments**
>
> We thank the reviewer for the detailed comments and for the suggested improvements. We also appreciated the highlights of our strengths, especially the linear subspace and the context-conditioned meta-reward.
>
> We hope our updated work addresses your main concerns. See below for a highlight of our changes.
> 1. **Clarity in Section 3**
>    We agree that Sec. 3 is dense and make several improvements. To reduce the jump between subsections, we added new introductory paragraphs to each subsection, tying them to the previous discussion. We also provide a smoother transition from our discussion of $g$ to that of $f_{u}$. Related to the clarity in this section, we also address your smaller concerns:
> 	- We clarified the notation change from $\mathcal{D}^{(K)}$ to $B_{j}$ in Sec. 3.3, and discussed our reasoning of emphasizing $B_{j}$ being part of a batch of meta-episodes
> 	- We clarified the discussion on the optimization of $f_{u}$ in Sec. 3.3. Specifically, we follow your advice of adapting the paragraph after Eq. 8, moving most of the high-level intuition before the detailed formal definitions. We believe readers will now have a better understanding of what we aim to achieve in this subsection
> 	- We explicitly named the $h_\text{rec}$ head in Sec. 3.4 and discussed in more detail what its role is and how it works
> 	- Thank you for highlighting the inconsistency in our notation regarding vector $d$. We updated Sec. 3.2 and introduced the $\hat{d}$ vector as the version of $d$ with a causal mask applied. We additionally updated Eq. 4 to reflect this change
> 2. **Baseline for Figure 8**
>    We believe your suggested baseline is almost identical to our In-Context PPO baseline. Specifically, the main difference is that In-Context PPO first processes the state and context separately, and only then concatenates their representations and optimizes via standard PPO. We have updated Sec. 5.3 to better explain In-Context PPO
> 3. **Comparison to "First-Explore, then Exploit"**
>    We have expanded Sec. 4 to include a detailed conceptual comparison with Norman & Clune (2024), as their work shares some objectives with ours. However, we clarify that while both methods decouple exploration, First-Explore is not explicitly designed to explore in a manner that enhances subsequent exploration. That is, in a K-shot adaptation setting with $K > 1$, First-Explore is likely to not fully leverage exploration beyond the first episode. On the contrary, LaSER is specifically designed to work in settings with a $K > 1$ exploration budget. Specifically, LaSER aims to minimize the redundancy between exploration episodes and to leverage early collected episodes to enhance future exploration. Given these differences in the scope of the algorithms, we focused on discussing the conceptual distinctions rather than performing empirical evaluations.
> 4. **Parameter count and computational complexity**
>    We extended Sec. 5.1.3 to address your concerns. We especially highlight that, due to the architectures employed by the baselines, scaling them to the size of LaSER is very difficult in practice, as it often leads to instabilities or overfitting.
> 5. Regarding the rest of your smaller points:
> 	- We appreciate and agree with your comments on our discussion on Fig. 7. We have updated our discussion in Sec. 5.2.3 to better reflect the results, and we have explicitly acknowledged the overlap of the error bars
> 	- We have introduced a new qualitative analysis of LaSER's behavior during exploration. In Fig. 19, we present meta-episodes composed of $K=4$ episodes each and collected from two MEWA tasks (in-distribution task with moderate mistake chance, and OOD task where type 4 mistakes never occur, while the rest have high likelihood). Episodes are collected during exploration using the fully meta-trained LaSER, VariBAD, and DREAM. The discussion of these results is in Sec. 5.4 and Appendix H.3. This study indicates that LaSER successfully avoids redundancy, especially compared to VariBAD. LaSER's diverse behavior can be seen in the different sequences of actions it takes or types of MEWA mistakes it experiments with. DREAM is diverse in the first task, but eventually collapses into a deterministic exploration policy in the other. LaSER appears to avoid this pitfall, which is likely due to it being explicitly encouraged to avoid redundancy during meta-training and due to its tendency to ignore environmental rewards during exploration.
>
> Finally, to answer your final question, all algorithms are trained for the same amount of timesteps during the task-solving phase, but LaSER also has some pre-training for its transformer encoder. Fig. 5 shows this, while we also discuss this difference in training throughout Sections 5.1 and 5.2.
>
>
> References
> - Norman, B. and Clune, J., 2024. First-Explore, then Exploit: Meta-Learning to Solve Hard Exploration-Exploitation Trade-Offs. Advances in Neural Information Processing Systems, 37, pp.27490-27528.

---

### Comment · Reviewer_X7vr · 2025-12-25

This paper proposes LaSER, a meta-RL framework aimed at improving few-shot adaptation through task-specific, sample-efficient exploration. The key idea is to formalize informative exploration as identifying a small set of episodes that can linearly reconstruct a larger dataset collected from the same task, under a low-rank assumption. Based on this perspective, the authors introduce a task reconstruction objective to supervise representation learning for exploration, an intrinsic exploration reward that encourages episode-level diversity in a learned latent space, and a novel meta-reward to address the issue that in-context policies may fail to exploit task contexts even when they are available. The method is evaluated primarily on the MEWA benchmark, where exploration is essential, and additionally on Meta-World and HopperMass to demonstrate generality. Empirical results show that LaSER significantly outperforms established meta-RL baselines on exploration-centric tasks, while providing competitive performance on standard benchmarks.

Pros.
1. **Clear and novel conceptual framing of exploration in meta-RL.** The idea of low-rank task encoder is clean and technically sound. This is a novel perspective of task modeling in meta-RL.
2. **Well-motivated optimization contribution addressing context underutilization.** The proposed meta-reward explicitly tackles the empirically observed but under-discussed issue that in-context policies may ignore task context even when it is informative.
3. **The implementation is provided to improve reproducibility.**

Cons.
1. **Limited qualitative analysis of exploration behavior.** While quantitative results are strong, the paper lacks trajectory-level or qualitative examples illustrating how the learned exploration policy behaves differently from baselines. Given the interpretability of MEWA, concrete behavioral visualizations or walkthroughs would substantially strengthen the claims about task-specific exploration.
2. **Assumption of linear task reconstruction is not empirically stress-tested.** Considering the complexity of tasks in the wild, the linear task reconstruction could be unrealistic if task identity is not representable as a low-rank combination of episode-level information, such as in environments with highly continuous, nonlinear, or noise-dominated task variations.
3. **Missing discussion of recent meta-RL works.**
Zhou, Hongtu, et al. "CERTAIN: Context Uncertainty-aware One-Shot Adaptation for Context-based Offline Meta Reinforcement Learning." Forty-second International Conference on Machine Learning.
Chu, Zhendong, et al. "Meta-reinforcement learning via exploratory task clustering." Proceedings of the AAAI Conference on Artificial Intelligence. Vol. 38. No. 10. 2024.

---

### Decision · Action_Editor_aiqC · 2026-03-01

**Recommendation:** Accept with minor revision

**Additional Comments:**

I support acceptance, and I recommend the following revisions to improve clarity and impact (many are partially addressed in the rebuttal, but should be reflected cleanly in the final manuscript):
•	Reduce perceived density in Section 3: further streamline notation, add more “why this design” explanation, and consider moving secondary derivations/architectural details to the appendix where possible, while keeping the method definition self-contained. Multiple reviewers consistently flagged presentation density as the main remaining weakness.
•	Clarify efficiency accounting: explicitly separate (i) meta-training compute/data (including encoder/exploration pretraining) from (ii) meta-test interaction budget, and avoid ambiguous “sample efficiency” claims.
•	Strengthen the discussion of structural assumptions: the paper should more directly articulate when the linear/low-rank reconstruction assumption is expected to be adequate, when the nonlinear variant may be necessary, and what failure modes arise as tasks become more nonlinear/compositional. The rebuttal adds helpful material here; it should be integrated into a crisp “scope and limitations” narrative.

**Audience:**

Yes

**Audience Explanation:**

Yes. The paper tackles two practically important and under-discussed issues in meta-RL: task-specific exploration under few-shot budgets and context underutilization by in-context policies. Reviewers highlight that the reconstruction-based framing of “informativeness” and the explicit exploration/exploitation decoupling provide useful conceptual tools that could influence follow-on work, even if some assumptions (e.g., linear/low-rank reconstruction and fixed exploration budget) may limit applicability in certain environments.

**Claims And Evidence:**

Yes

**Claims Explanation:**

Overall, "yes". All reviewers agree the paper’s central claims are supported by experiments (including oracle-context evaluations and ablations) showing that (i) better task exploration and (ii) better context exploitation are both necessary for strong few-shot adaptation performance.

That said, reviewers raised valid caveats about how some claims should be framed: (a) the method’s complexity and density initially made it hard to follow, (b) some comparisons and baseline coverage were not consistently justified in earlier drafts, and (c) the “sample-efficiency” narrative is nuanced because LaSER includes pretraining for the encoder/exploration components, effectively shifting compute/data to earlier stages.

The authors’ rebuttal materially strengthens the evidence and clarity by adding (i) qualitative analyses of exploration behavior, (ii) additional ablations, especially on the meta-reward alternatives, and (iii) clearer justification of baseline selection and a more explicit discussion of the training-time vs test-time efficiency trade-off. Post-rebuttal, reviewers’ remaining concerns are primarily about presentation density and scope/assumptions rather than evidentiary validity, and the final reviewer recommendations converge to accept / leaning accept.

---

> ### Author Response · Authors · 2026-03-27
>
> We wish to thank the Action Editor for supporting acceptance and for suggesting the final improvements for our camera-ready version. We have addressed all suggestions:
>
> - We moved the more rigorous or technical parts of Section 3 to the Appendix. Section 3 now provides the main intuition of our approach and the most important mathematical and architectural details, and remains self-contained. Besides adding detailed explanations in the Appendix, we also add a new table summarising the main notation and definitions of our work. We believe this will help readers follow along.
> - We agree that many of the sample-efficiency claims we made were ambiguous and could have led to confusion. We fixed this by explicitly mentioning when we refer to meta-training sample efficiency (i.e., the number of samples required to train the whole model) and few-shot "sample efficiency" (i.e., the number of "shots" required to adapt).
> - We provided new discussions regarding our linear task reconstruction assumption. We explicitly tied this assumption to the type of domains in which LaSER is expected to underperform (including examples), described our hypothesis of what failures may occur if the low-rank assumption does not hold, and proposed our non-linear task reconstruction extension as a starting point for scaling LaSER to these environments.
> - We followed the suggestion of Reviewer soxu and added the First-Explore (Norman & Clune, 2024) baseline, comparing with LaSER on the MEWA benchmark. Moreover, since this is a transformer-based algorithm, we were able to scale the First-Explore agent to the same size as LaSER's meta-testing architecture.
>
> References
> - Norman, B. and Clune, J., 2024. First-Explore, then Exploit: Meta-Learning to Solve Hard Exploration-Exploitation Trade-Offs. Advances in Neural Information Processing Systems, 37, pp.27490-27528.